# Mirror and Preconditioned Gradient Descent in Wasserstein Space

**Clément Bonet**
CREST, ENSAE, IP Paris
clement.bonet@ensae.fr

**Théo Uscidda**
CREST, ENSAE, IP Paris
theo.uscidda@ensae.fr

**Adam David**
Institute of Mathematics
Technische Universität Berlin
david@math.tu-berlin.de

**Pierre-Cyril Aubin-Frankowski**
TU Wien
pierre-cyril.aubin@tuwien.ac.at

**Anna Korba**
CREST, ENSAE, IP Paris
anna.korba@ensae.fr

## Abstract

As the problem of minimizing functionals on the Wasserstein space encompasses many applications in machine learning, different optimization algorithms on $\mathbb{R}^d$ have received their counterpart analog on the Wasserstein space. We focus here on lifting two explicit algorithms: mirror descent and preconditioned gradient descent. These algorithms have been introduced to better capture the geometry of the function to minimize and are provably convergent under appropriate (namely relative) smoothness and convexity conditions. Adapting these notions to the Wasserstein space, we prove guarantees of convergence of some Wasserstein-gradient-based discrete-time schemes for new pairings of objective functionals and regularizers. The difficulty here is to carefully select along which curves the functionals should be smooth and convex. We illustrate the advantages of adapting the geometry induced by the regularizer on ill-conditioned optimization tasks, and showcase the improvement of choosing different discrepancies and geometries in a computational biology task of aligning single-cells.

## 1 Introduction

Minimizing functionals on the space of probability distributions has become ubiquitous in machine learning for *e.g.* sampling [13, 129], generative modeling [53, 86], learning neural networks [36, 90, 107], dataset transformation [4, 63], or modeling population dynamics [23, 120]. It is a challenging task as it is an infinite-dimensional problem. Wasserstein gradient flows [5] provide an elegant way to solve such problems on the Wasserstein space, *i.e.*, the space of probability distributions with bounded second moment, equipped with the Wasserstein-2 distance from optimal transport (OT). These flows provide continuous paths of distributions decreasing the objective functional and can be seen as analog to Euclidean gradient flows [111]. Their implicit time discretization, referred to as the JKO scheme [66], has been studied in depth [1, 26, 95, 111]. In contrast, explicit schemes, despite being easier to implement, have been less investigated. Most previous works focus on the optimization of a specific objective functional with a time-discretation of its gradient flow with the Wasserstein-2 metrics. For instance, the forward Euler discretization leads to the Wasserstein gradient descent. The latter takes the form of gradient descent (GD) on the position of particles for functionals with a closed-form over discrete measures, *e.g.* Maximum Mean Discrepancy (MMD), which can be of interest to train neural networks [7, 30]. For objectives involving absolutely continuous measures, such as the Kullback-Leibler (KL) divergence for sampling, other discretizations can be easily computed such as the Unadjusted Langevin Algorithm (ULA) [106]. This leaves the question open of assessing

38th Conference on Neural Information Processing Systems (NeurIPS 2024).

the theoretical and empirical performance of other optimization algorithms relying on alternative geometries and time-discretizations.

In the optimization community, a recent line of works has focused on extending the methods and convergence theory beyond the Euclidean setting by using more general costs for the gradient descent scheme [77]. For instance, mirror descent (MD), originally introduced by Nemirovskij and Yudin [92] to solve constrained convex problems, uses a cost that is a divergence defined by a Bregman potential [12]. Mirror descent benefits from convergence guarantees for objective functions that are relatively smooth in the geometry induced by the (Bregman) divergence [88], even if they do not have a Lipschitz gradient, *i.e.*, are not smooth in the Euclidean sense. More recently, a closely related scheme, namely preconditioned gradient descent, was introduced in [89]. It can be seen as a dual version of the mirror descent algorithm, where the role of the objective function and Bregman potential are exchanged. In particular, its convergence guarantees can be obtained under relative smoothness and convexity of the Fenchel transform of the potential, with respect to the objective. This algorithm appears more efficient to minimize the gradient magnitude than mirror descent [68]. The flexible choice of the Bregman divergence used by these two schemes enables to design or discover geometries that are potentially more efficient.

Mirror descent has already attracted attention in the sampling community, and some popular algorithms have been extended in this direction. For instance, ULA was adapted into the Mirror Langevin algorithm [3, 32, 62, 64, 79, 121, 132]. Other sampling algorithms have received their counterpart mirror versions such as the Metropolis Adjusted Langevin Algorithm [116], diffusion models [82], Stein Variational Gradient Descent (SVGD) [114], or even Wasserstein gradient descent [113]. Preconditioned Wasserstein gradient descent has been also recently proposed for specific geometries in [31, 44] to minimize the KL in a more efficient way, but without an analysis in discrete time. All the previous references focus on optimizing the KL as an objective, while Wasserstein gradient flows have been studied in machine learning for different functionals such as more general $f$-divergences [6, 93], interaction energies [19, 78], MMDs [7, 30, 59, 60, 72] or Sliced-Wasserstein (SW) distances [15, 18, 45, 86]. In this work, we propose to bridge this gap by providing a general convergence theory of both mirror and preconditioned gradient descent schemes for general target functionals, and investigate as well empirical benefits of alternative transport geometries for optimizing functionals on the Wasserstein space. We emphasize that the latter is different from [9, 67], wherein mirror descent is defined in the Radon space of probability distributions, using the flat geometry defined by TV or $L^2$ norms on measures, see Appendix A for more details.

**Contributions.** We are interested in minimizing a functional $\mathcal{F} : \mathcal{P}_2(\mathbb{R}^d) \to \mathbb{R} \cup \{+\infty\}$ over probability distributions, through schemes of the form, for $\tau > 0$, $k \geq 0$,

$$\mathrm{T}_{k+1} = \operatorname*{argmin}_{\mathrm{T} \in L^2(\mu_k)} \langle \nabla_{\mathrm{W}_2}\mathcal{F}(\mu_k), \mathrm{T} - \mathrm{Id} \rangle_{L^2(\mu_k)} + \frac{1}{\tau}\mathrm{d}(\mathrm{T}, \mathrm{Id}), \quad \mu_{k+1} = (\mathrm{T}_{k+1})_{\#}\mu_k, \quad (1)$$

with different costs $\mathrm{d} : L^2(\mu_k) \times L^2(\mu_k) \to \mathbb{R}_+$, and in providing convergence conditions. While we can recover a map $\bar{\mathrm{T}} = \mathrm{T}_k \circ \mathrm{T}_{k-1} \cdots \circ \mathrm{T}_1$ such that $\mu_k = \bar{\mathrm{T}}_{\#}\mu_0$, the scheme (1) proceeds by successive regularized linearizations retaining the Wasserstein structure, since the tangent space to $\mathcal{P}_2(\mathbb{R}^d)$ at $\mu$ is a subset of $L^2(\mu)$ [96]. This paper is organized as follows. In Section 2, we provide some background on Bregman divergences, as well as on differentiability and convexity over the Wasserstein space. In Section 3, we consider Bregman divergences on $L^2(\mu)$ for the cost in (1), generalizing the mirror descent scheme to the Wasserstein space. We study this new scheme by discussing its implementation, and proving its convergence under relative smoothness and convexity assumptions. In Section 4, we consider alternative costs in (1), that are analogous to OT distances with translation-invariant cost, extending the dual space preconditioning scheme to the latter space. Finally, in Section 5, we apply the two schemes to different objective functionals, including standard free energy functionals such as interaction energies and KL divergence, but also to Sinkhorn divergences [50] or SW [16, 102] with polynomial preconditioners on single-cell datasets.

**Notations.** Consider the set $\mathcal{P}_2(\mathbb{R}^d)$ of probability measures $\mu$ on $\mathbb{R}^d$ with finite second moment and $\mathcal{P}_{2,\mathrm{ac}}(\mathbb{R}^d) \subset \mathcal{P}_2(\mathbb{R}^d)$ its subset of absolutely continuous probability measures with respect to the Lebesgue measure. For any $\mu \in \mathcal{P}_2(\mathbb{R}^d)$, we denote by $L^2(\mu)$ the Hilbert space of functions $f : \mathbb{R}^d \to \mathbb{R}^d$ such that $\int \|f\|^2 \mathrm{d}\mu < \infty$ equipped with the norm $\|\cdot\|_{L^2(\mu)}$ and inner product $\langle \cdot, \cdot \rangle_{L^2(\mu)}$. For a Hilbert space $X$, the Fenchel transform of $f : X \to \mathbb{R}$ is $f^*(y) = \sup_{x \in X} \langle x, y \rangle - f(x)$. Given a measurable map $\mathrm{T} : \mathbb{R}^d \to \mathbb{R}^d$ and $\mu \in \mathcal{P}_2(\mathbb{R}^d)$, $\mathrm{T}_{\#}\mu$ is the pushforward measure of $\mu$ by

T; and $T \star \mu = \int T(\cdot - x) d\mu(x)$. For $\mu, \nu \in \mathcal{P}_2(\mathbb{R}^d)$, the Wasserstein-2 distance is $W_2^2(\mu, \nu) = \inf_{\gamma \in \Pi(\mu, \nu)} \int \|x - y\|^2 \, d\gamma(x, y)$, where $\Pi(\mu, \nu) = \{\gamma \in \mathcal{P}(\mathbb{R}^d \times \mathbb{R}^d), \ \pi_\#^1 \gamma = \mu, \ \pi_\#^2 \gamma = \nu\}$ with $\pi^i(x_1, x_2) = x_i$, is the set of couplings between $\mu$ and $\nu$, and we denote by $\Pi_o(\mu, \nu)$ the set of optimal couplings. When the optimal coupling is of the form $\gamma = (\mathrm{Id}, T_\mu^\nu)_\# \mu$ with $\mathrm{Id} : x \mapsto x$ and $T_\mu^\nu \in L^2(\mu)$ satisfying $(T_\mu^\nu)_\# \mu = \nu$, we call $T_\mu^\nu$ the OT map. We refer to the metric space $(\mathcal{P}_2(\mathbb{R}^d), W_2)$ as the Wasserstein space. We note $S_d^{++}(\mathbb{R})$ the space of symmetric positive definite matrices, and for $x \in \mathbb{R}^d, \Sigma \in S_d^{++}(\mathbb{R}), \|x\|_\Sigma^2 = x^T \Sigma x$.

## 2 Background

In this section, we fix $\mu \in \mathcal{P}_2(\mathbb{R}^d)$ and introduce first the Bregman divergence on $L^2(\mu)$ along with the notions of relative convexity and smoothness that will be crucial in the analysis of the optimization schemes. Then, we introduce the differential structure and computation rules for differentiating a functional $\mathcal{F} : \mathcal{P}_2(\mathbb{R}^d) \to \mathbb{R}$ along curves and discuss notions of convexity on $\mathcal{P}_2(\mathbb{R}^d)$. We refer the reader to Appendix B and Appendix C for more details on $L^2(\mu)$ and the Wasserstein space respectively. Finally, we introduce the mirror descent and preconditioned gradient descent on $\mathbb{R}^d$.

**Bregman divergence on $L^2(\mu)$.** Frigyik et al. [54, Definition 2.1] defined the Bregman divergence of Fréchet differentiable functionals. In our case, we only need Gâteaux differentiability. In this paper, $\nabla$ refers to the Gâteaux differential, which coincides with the Fréchet derivative if the latter exists.

**Definition 1.** *Let $\phi_\mu : L^2(\mu) \to \mathbb{R}$ be convex and continuously Gâteaux differentiable. The Bregman divergence is defined for all $T, S \in L^2(\mu)$ as $d_{\phi_\mu}(T, S) = \phi_\mu(T) - \phi_\mu(S) - \langle \nabla \phi_\mu(S), T - S \rangle_{L^2(\mu)}$.*

We use the same definition on $\mathbb{R}^d$. The map $\phi_\mu$ (respectively $\nabla \phi_\mu$) in the definition of $d_{\phi_\mu}$ above is referred to as the Bregman potential (respectively mirror map). If $\phi_\mu$ is strictly convex, then $d_{\phi_\mu}$ is a valid Bregman divergence, *i.e.* it is positive and separates maps $\mu$-almost everywhere (a.e.). In particular, for $\phi_\mu(T) = \frac{1}{2}\|T\|_{L^2(\mu)}^2$, we recover the $L^2$ norm as a divergence $d_{\phi_\mu}(T, S) = \frac{1}{2}\|T - S\|_{L^2(\mu)}^2$. Bregman divergences have received a lot of attention as they allow to define provably convergent schemes for functions which are not smooth in the standard (*e.g.* Euclidean) sense [11, 88], and thus for which gradient descent is not appropriate. These guarantees rely on the notion of relative smoothness and relative convexity [88, 89], which we introduce now on $L^2(\mu)$.

**Definition 2** (Relative smoothness and convexity). *Let $\psi_\mu, \phi_\mu : L^2(\mu) \to \mathbb{R}$ be convex and continuously Gâteaux differentiable. We say that $\psi_\mu$ is $\beta$-smooth (respectively $\alpha$-convex) relative to $\phi_\mu$ if and only if for all $T, S \in L^2(\mu)$, $d_{\psi_\mu}(T, S) \leq \beta d_{\phi_\mu}(T, S)$ (respectively $d_{\psi_\mu}(T, S) \geq \alpha d_{\phi_\mu}(T, S)$).*

Similarly to the Euclidean case [88], relative smoothness and convexity are equivalent to respectively $\beta \phi_\mu - \psi_\mu$ and $\psi_\mu - \alpha \phi_\mu$ being convex (see Appendix B.2). Yet, proving the convergence of (1) requires only that these properties hold at specific functions (directions), a fact we will soon exploit.

In some situations, we need the $L^2$ Fenchel transform $\phi_\mu^*$ of $\phi_\mu$ to be differentiable, *e.g.* to compute its Bregman divergence $d_{\phi_\mu^*}$. We show in Lemma 18 that a sufficient condition to satisfy this property is for $\phi_\mu$ to be strictly convex, lower semicontinuous and superlinear, *i.e.* $\lim_{\|T\| \to \infty} \phi_\mu(T)/\|T\|_{L^2(\mu)} = +\infty$. Moreover, in this case, $(\nabla \phi_\mu)^{-1} = \nabla \phi_\mu^*$. When needed, we will suppose that $\phi_\mu$ satisfies this assumption.

**Differentiability on $(\mathcal{P}_2(\mathbb{R}^d), W_2)$.** Let $\mathcal{F} : \mathcal{P}_2(\mathbb{R}^d) \to \mathbb{R} \cup \{+\infty\}$, and denote $D(\mathcal{F}) = \{\mu \in \mathcal{P}_2(\mathbb{R}^d), \ \mathcal{F}(\mu) < +\infty\}$ the domain of $\mathcal{F}$ and $D(\tilde{\mathcal{F}}_\mu) = \{T \in L^2(\mu), \ T_\# \mu \in D(\mathcal{F})\}$ the domain of $\tilde{\mathcal{F}}_\mu$ defined as $\tilde{\mathcal{F}}_\mu(T) := \mathcal{F}(T_\# \mu)$ for all $T \in L^2(\mu)$. In the following, we use the differential structure of $(\mathcal{P}_2(\mathbb{R}^d), W_2)$ introduced in [17, Definition 2.8], and we say that $\nabla_{W_2} \mathcal{F}(\mu)$ is a Wasserstein gradient of $\mathcal{F}$ at $\mu \in D(\mathcal{F})$ if for any $\nu \in \mathcal{P}_2(\mathbb{R}^d)$ and any optimal coupling $\gamma \in \Pi_o(\mu, \nu)$,

$$\mathcal{F}(\nu) = \mathcal{F}(\mu) + \int \langle \nabla_{W_2} \mathcal{F}(\mu)(x), y - x \rangle \, d\gamma(x, y) + o(W_2(\mu, \nu)). \tag{2}$$

If such a gradient exists, then we say that $\mathcal{F}$ is Wasserstein differentiable at $\mu$ [17, 74]. Moreover there is a unique gradient belonging to the tangent space of $\mathcal{P}_2(\mathbb{R}^d)$ verifying (2) [74, Proposition

2.5], and we will always restrict ourselves without loss of generality to this particular gradient, see Appendix C.1. The differentiability of $\mathcal{F}$ and $\tilde{\mathcal{F}}_\mu$ are very related, as described in the following proposition.

**Proposition 1.** *Let $\mathcal{F} : \mathcal{P}_2(\mathbb{R}^d) \to \mathbb{R} \cup \{+\infty\}$ be a Wasserstein differentiable functional on $D(\mathcal{F})$. Let $\mu \in \mathcal{P}_2(\mathbb{R}^d)$ and $\tilde{\mathcal{F}}_\mu(\mathrm{T}) = \mathcal{F}(\mathrm{T}_{\#}\mu)$ for all $\mathrm{T} \in D(\tilde{\mathcal{F}}_\mu)$. Then, $\tilde{\mathcal{F}}_\mu$ is Fréchet differentiable, and for all $\mathrm{S} \in D(\tilde{\mathcal{F}}_\mu)$, $\nabla\tilde{\mathcal{F}}_\mu(\mathrm{S}) = \nabla_{\mathrm{W}_2}\mathcal{F}(\mathrm{S}_{\#}\mu) \circ \mathrm{S}$.*

The Wasserstein differentiable functionals include $c$-Wasserstein costs on $\mathcal{P}_{2,\mathrm{ac}}(\mathbb{R}^d)$ [74, Proposition 2.10 and 2.11], potential energies $\mathcal{V}(\mu) = \int V \mathrm{d}\mu$ or interaction energies $\mathcal{W}(\mu) = \frac{1}{2}\iint W(x-y)\,\mathrm{d}\mu(x)\mathrm{d}\mu(y)$ for $V : \mathbb{R}^d \to \mathbb{R}$ and $W : \mathbb{R}^d \to \mathbb{R}$ differentiable and with bounded Hessian [74, Section 2.4]. In particular, their Wasserstein gradients read as $\nabla_{\mathrm{W}_2}\mathcal{V}(\mu) = \nabla V$ and $\nabla_{\mathrm{W}_2}\mathcal{W}(\mu) = \nabla W \star \mu$. However, entropy functionals, *e.g.* the negative entropy defined as $\mathcal{H}(\mu) = \int \log\big(\rho(x)\big)\mathrm{d}\mu(x)$ for distributions $\mu$ admitting a density $\rho$ *w.r.t.* the Lebesgue measure, are not Wasserstein differentiable. In this case, we can consider subgradients $\nabla_{\mathrm{W}_2}\mathcal{F}(\mu)$ at $\mu$ for which (2) becomes an inequality. To guarantee that the Wasserstein subgradient is not empty, we need $\rho$ to satisfy some Sobolev regularity, see *e.g.* [5, Theorem 10.4.13] or [108]. Then, if $\nabla \log \rho \in L^2(\mu)$, the only subgradient of $\mathcal{H}$ in the tangent space is $\nabla_{\mathrm{W}_2}\mathcal{H}(\mu) = \nabla \log \rho$, see [5, Theorem 10.4.17] and [47, Proposition 4.3]. Then, free energies are functionals that write as sums of potential, interaction and entropy terms [110, Chapter 7]. It is notably the case for the KL to a fixed target distribution, that is the sum of a potential and entropy term [129], or the MMD as a sum of a potential and interaction term [7].

**Examples of functionals.** The definitions of Bregman divergences on $L^2(\mu)$ and of Wasserstein differentiability enable us to consider alternative Bregman potentials than the $L^2(\mu)$-norm mentioned above. For instance, for $V$ convex, differentiable and $L$-smooth, we can use potential energies $\phi_\mu^V(\mathrm{T}) := \mathcal{V}(\mathrm{T}_{\#}\mu)$, for which $\mathrm{d}_{\phi_\mu^V}(\mathrm{T},\mathrm{S}) = \int \mathrm{d}_V\big(\mathrm{T}(x),\mathrm{S}(x)\big)\mathrm{d}\mu(x)$ where $\mathrm{d}_V$ is the Bregman divergence of $V$ on $\mathbb{R}^d$. Notice that $\phi_\mu(\mathrm{T}) = \frac{1}{2}\|\mathrm{T}\|^2_{L^2(\mu)}$ is a specific example of a potential energy where $V = \frac{1}{2}\|\cdot\|^2$. Moreover, we will consider interaction energies $\phi_\mu^W(\mathrm{T}) := \mathcal{W}(\mathrm{T}_{\#}\mu)$ with $W$ convex, differentiable, $L$-smooth, and satisfying $W(-x) = W(x)$; for which $\mathrm{d}_{\phi_\mu^W}(\mathrm{T},\mathrm{S}) = \frac{1}{2}\iint \mathrm{d}_W\big(\mathrm{T}(x) - \mathrm{T}(x'), \mathrm{S}(x) - \mathrm{S}(x')\big)\mathrm{d}\mu(x)\mathrm{d}\mu(x')$ (see Appendix I.3). We will also use $\phi_\mu^{\mathcal{H}}(\mathrm{T}) = \mathcal{H}(\mathrm{T}_{\#}\mu)$ with $\mathcal{H}$ the negative entropy. Note that Bregman divergences on the Wasserstein space using these functionals were proposed by Li [80], but only for $\mathrm{S} = \mathrm{Id}$ and optimal transport maps $\mathrm{T}$.

**Convexity and smoothness in $(\mathcal{P}_2(\mathbb{R}^d), \mathrm{W}_2)$.** In order to study the convergence of gradient flows and their discrete-time counterparts, it is important to have suitable notions of convexity and smoothness. On $(\mathcal{P}_2(\mathbb{R}^d), \mathrm{W}_2)$, different such notions have been proposed based on specific choices of curves. The most popular one is to require the functional $\mathcal{F}$ to be $\alpha$-convex along geodesics (see Definition 10), which are of the form $\mu_t = \big((1-t)\mathrm{Id} + t\mathrm{T}_{\mu_0}^{\mu_1}\big)_{\#}\mu_0$ if $\mu_0 \in \mathcal{P}_{2,\mathrm{ac}}(\mathbb{R}^d)$ and $\mu_1 \in \mathcal{P}_2(\mathbb{R}^d)$, with $\mathrm{T}_{\mu_0}^{\mu_1}$ the OT map between them. In that setting,

$$\frac{\alpha}{2}\mathrm{W}_2^2(\mu_0,\mu_1) = \frac{\alpha}{2}\|\mathrm{T}_{\mu_0}^{\mu_1} - \mathrm{Id}\|^2_{L^2(\mu_0)} \leq \mathcal{F}(\mu_1) - \mathcal{F}(\mu_0) - \langle\nabla_{\mathrm{W}_2}\mathcal{F}(\mu_0), \mathrm{T}_{\mu_0}^{\mu_1} - \mathrm{Id}\rangle_{L^2(\mu_0)}. \quad (3)$$

For instance, free energies such as potential or interaction energies with convex $V$ or $W$, or the negative entropy, are convex along geodesics [110, Section 7.3]. However, some popular functionals, such as the Wasserstein-2 distance $\mu \mapsto \frac{1}{2}\mathrm{W}_2^2(\mu,\eta)$ itself, for a given $\eta \in \mathcal{P}_2(\mathbb{R}^d)$, are not convex along geodesics. Instead Ambrosio et al. [5, Theorem 4.0.4] showed that it was sufficient for the convergence of the gradient flow to be convex along other curves, *e.g.* along particular generalized geodesics for the Wasserstein-2 distance [5, Lemma 9.2.7], which, for $\mu,\nu \in \mathcal{P}_2(\mathbb{R}^d)$, are of the form $\mu_t = \big((1-t)\mathrm{T}_\eta^\mu + t\mathrm{T}_\eta^\nu\big)_{\#}\eta$ for $\mathrm{T}_\eta^\mu, \mathrm{T}_\eta^\nu$ OT maps from $\eta$ to $\mu$ and $\nu$. Observing that for $\phi_\mu(\mathrm{T}) = \frac{1}{2}\|\mathrm{T}\|^2_{L^2(\mu)}$, we can rewrite (3) as $\alpha\mathrm{d}_{\phi_{\mu_0}}(\mathrm{T}_{\mu_0}^{\mu_1}, \mathrm{Id}) \leq \mathrm{d}_{\tilde{\mathcal{F}}_{\mu_0}}(\mathrm{T}_{\mu_0}^{\mu_1}, \mathrm{Id})$ and see that being convex along geodesics boils down to being convex in the $L^2$ sense for $\mathrm{S} = \mathrm{Id}$ and $\mathrm{T}$ chosen as an OT map. This observation motivates us to consider a more refined notion of convexity along curves.

**Definition 3.** *Let $\mu \in \mathcal{P}_2(\mathbb{R}^d)$, $\mathrm{T},\mathrm{S} \in L^2(\mu)$ and for all $t \in [0,1]$, $\mu_t = (\mathrm{T}_t)_{\#}\mu$ with $\mathrm{T}_t = (1-t)\mathrm{S} + t\mathrm{T}$. We say that $\mathcal{F} : \mathcal{P}_2(\mathbb{R}^d) \to \mathbb{R}$ is $\alpha$-convex (resp. $\beta$-smooth) relative to $\mathcal{G} : \mathcal{P}_2(\mathbb{R}^d) \to \mathbb{R}$ along $t \mapsto \mu_t$ if for all $s,t \in [0,1]$, $\mathrm{d}_{\tilde{\mathcal{F}}_\mu}(\mathrm{T}_s,\mathrm{T}_t) \geq \alpha\mathrm{d}_{\tilde{\mathcal{G}}_\mu}(\mathrm{T}_s,\mathrm{T}_t)$ (resp. $\mathrm{d}_{\tilde{\mathcal{F}}_\mu}(\mathrm{T}_s,\mathrm{T}_t) \leq \beta\mathrm{d}_{\tilde{\mathcal{G}}_\mu}(\mathrm{T}_s,\mathrm{T}_t)$).*

Notice that in contrast with Definition 2, Definition 3 is stated for a fixed distribution $\mu$ and directions $(\mathrm{S}, \mathrm{T})$, and involves comparisons between Bregman divergences depending on $\mu$ and curves $(\mathrm{T}_s)_{s \in [0,1]}$ depending on $\mathrm{S}, \mathrm{T}$. The larger family of $\mathrm{S}$ and $\mathrm{T}$ for which Definition 3 holds, the more restricted is the notion of convexity of $\mathcal{F} - \alpha\mathcal{G}$ (resp. of $\beta\mathcal{G} - \mathcal{F}$) on $\mathcal{P}_2(\mathbb{R}^d)$. For instance, Wasserstein-2 generalized geodesics with anchor $\eta \in \mathcal{P}_2(\mathbb{R}^d)$ correspond to considering $\mathrm{S}, \mathrm{T}$ as all the OT maps originating from $\eta$, among which geodesics are particular cases when taking $\eta = \mu$ (hence $\mathrm{S} = \mathrm{Id}$). If we furthermore ask for $\alpha$-convexity to hold for all $\mu \in \mathcal{P}_2(\mathbb{R}^d)$ and $\mathrm{T}, \mathrm{S} \in L^2(\mu)$ (*i.e.*, not only OT maps), then we recover the convexity along acceleration free-curves as introduced in [28, 98, 117]. Our motivation behind introducing Definition 3 is that the convergence proofs of MD and preconditioned GD require relative smoothness and convexity properties to hold only along specific curves.

**Mirror descent and preconditioned gradient descent on $\mathbb{R}^d$.** These schemes read respectively as $\nabla\phi(x_{k+1}) - \nabla\phi(x_k) = -\tau\nabla f(x_k)$ [12] and $y_{k+1} - y_k = -\tau\nabla h^*\big(\nabla g(y_k)\big)$ [89], where the objectives $f, g$ and the regularizers $h, \phi$ are convex $C^1$ functions from $\mathbb{R}^d$ to $\mathbb{R}$. The algorithms are closely related since, using the Fenchel transform and setting $g = \phi^*$ and $h^* = f$, we see that, for $y = \nabla\phi(x)$, the two schemes are equivalent when permuting the roles of the objective and of the regularizer. For MD, convergence of $f$ is ensured if $f$ is both $1/\tau$-smooth and $\alpha$-convex relative to $\phi$ [88, Theorem 3.1]. Concerning preconditioned GD, assuming that $h, g$ are Legendre, $\big(g(y_k)\big)_k$ converges to the minimum of $g$ if $h^*$ is both $1/\tau$-smooth and $\alpha$-convex relative to $g^*$ with $\alpha > 0$ [89, Theorem 3.9].

# 3  Mirror descent

For every $\mu \in \mathcal{P}_2(\mathbb{R}^d)$, let $\phi_\mu : L^2(\mu) \to \mathbb{R}$ be strictly convex, proper and differentiable and assume that the (sub)gradient $\nabla_{\mathrm{W}_2}\mathcal{F}(\mu) \in L^2(\mu)$ exists. In this section, we are interested in analyzing the scheme (1) where the cost $\mathrm{d}$ is chosen as a Bregman divergence, *i.e.* $\mathrm{d}_{\phi_\mu}$ as defined in Definition 1. This corresponds to a mirror descent scheme in $\mathcal{P}_2(\mathbb{R}^d)$. For $\tau > 0$ and $k \geq 0$, it writes:

$$\mathrm{T}_{k+1} = \underset{\mathrm{T} \in L^2(\mu_k)}{\mathrm{argmin}} \ \mathrm{d}_{\phi_{\mu_k}}(\mathrm{T}, \mathrm{Id}) + \tau\langle\nabla_{\mathrm{W}_2}\mathcal{F}(\mu_k), \mathrm{T} - \mathrm{Id}\rangle_{L^2(\mu_k)}, \quad \mu_{k+1} = (\mathrm{T}_{k+1})_\#\mu_k. \quad (4)$$

**Iterates of mirror descent.** In all that follows, we assume that the iterates (4) exist, which is true *e.g.* for a superlinear $\phi_{\mu_k}$, since the objective is a sum of linear functions and of the continuous $\phi_{\mu_k}$. In the previous section, we have seen that the second term in the proximal scheme (4) can be interpreted as a linearization of the functional $\mathcal{F}$ at $\mu_k$ for Wasserstein (sub)differentiable functionals. Now define for all $\mathrm{T} \in L^2(\mu_k)$, $\mathrm{J}(\mathrm{T}) = \mathrm{d}_{\phi_{\mu_k}}(\mathrm{T}, \mathrm{Id}) + \tau\langle\nabla_{\mathrm{W}_2}\mathcal{F}(\mu_k), \mathrm{T} - \mathrm{Id}\rangle_{L^2(\mu_k)}$. Then, deriving the first order conditions of (4) as $\nabla\mathrm{J}(\mathrm{T}_{k+1}) = 0$, we obtain $\mu_k$-a.e.,

$$\nabla\phi_{\mu_k}(\mathrm{T}_{k+1}) = \nabla\phi_{\mu_k}(\mathrm{Id}) - \tau\nabla_{\mathrm{W}_2}\mathcal{F}(\mu_k) \iff \mathrm{T}_{k+1} = \nabla\phi_{\mu_k}^*\big(\nabla\phi_{\mu_k}(\mathrm{Id}) - \tau\nabla_{\mathrm{W}_2}\mathcal{F}(\mu_k)\big). \quad (5)$$

Note that for $\phi_\mu(\mathrm{T}) = \frac{1}{2}\|\mathrm{T}\|_{L^2(\mu)}^2$, the update (5) translates as $\mathrm{T}_{k+1} = \mathrm{Id} - \tau\nabla_{\mathrm{W}_2}\mathcal{F}(\mu_k)$, and our scheme recovers Wasserstein gradient descent [35, 91]. This is analogous to mirror descent recovering GD when the Bregman potential is chosen as the Euclidean squared norm in $\mathbb{R}^d$ [12]. We discuss in Appendix D.2 the continuous formulation of (4), showing it coincides with the gradient flow of the mirror Langevin [3, 130], the limit of the JKO scheme with Bregman groundcosts [104], Information Newton's flows [126], or Sinkhorn's flow [41] for specific choices of $\phi$ and $\mathcal{F}$.

Our proof of convergence of the mirror descent algorithm will require the Bregman divergence to satisfy the following property, which is reminiscent of conditions of optimality for couplings in OT.

**Assumption 1.** *For $\mu, \rho \in \mathcal{P}_{2,\mathrm{ac}}(\mathbb{R}^d)$ and $\nu \in \mathcal{P}_2(\mathbb{R}^d)$, setting $\mathrm{T}_{\phi_\mu}^{\mu,\nu} = \mathrm{argmin}_{\mathrm{T}_\#\mu=\nu} \ \mathrm{d}_{\phi_\mu}(\mathrm{T}, \mathrm{Id})$, $\mathrm{U}_{\phi_\rho}^{\rho,\nu} = \mathrm{argmin}_{\mathrm{U}_\#\rho=\nu} \ \mathrm{d}_{\phi_\rho}(\mathrm{U}, \mathrm{Id})$, the functional $\phi_\mu$ is such that, for any $\mathrm{S} \in L^2(\mu)$ satisfying $\mathrm{S}_\#\mu = \rho$, we have $\mathrm{d}_{\phi_\mu}(\mathrm{T}_{\phi_\mu}^{\mu,\nu}, \mathrm{S}) \geq \mathrm{d}_{\phi_\rho}(\mathrm{U}_{\phi_\rho}^{\rho,\nu}, \mathrm{Id})$.*

The inequality in Assumption 1 can be interpreted as follows: the "distance" between $\rho$ and $\nu$ is greater when observed from an anchor $\mu$ that differs from $\rho$ and $\nu$. We demonstrate that Bregman divergences satisfy this assumption under the following conditions on the Bregman potential $\phi$.

**Proposition 2.** *Let $\mu, \rho \in \mathcal{P}_{2,\mathrm{ac}}(\mathbb{R}^d)$ and $\nu \in \mathcal{P}_2(\mathbb{R}^d)$. Let $\phi_\mu$ be a pushforward compatible functional, i.e. there exists $\phi : \mathcal{P}_2(\mathbb{R}^d) \to \mathbb{R}$ such that for all $\mathrm{T} \in L^2(\mu)$, $\phi_\mu(\mathrm{T}) = \phi(\mathrm{T}_\#\mu)$. Assume furthermore $\nabla_{\mathrm{W}_2}\phi(\mu)$ and $\nabla_{\mathrm{W}_2}\phi(\rho)$ invertible (on $\mathbb{R}^d$). Then, $\phi_\mu$ satisfies Assumption 1.*

All the maps $\phi_\mu^V$, $\phi_\mu^W$ and $\phi_\mu^{\mathcal{H}}$ defined in Section 2 satisfy the assumptions of Proposition 2 under mild requirements, see Appendix D.1. The proof of Proposition 2 is given in Appendix H.2. It relies on the definition of an appropriate optimal transport problem

$$W_\phi(\nu, \mu) = \inf_{\gamma \in \Pi(\nu, \mu)} \phi(\nu) - \phi(\mu) - \int \langle \nabla_{W_2} \phi(\mu)(y), x - y \rangle \, \mathrm{d}\gamma(x, y), \tag{6}$$

and on the proof of existence of OT maps for absolutely continuous measures (see Proposition 15), which implies $W_\phi(\nu, \mu) = \mathrm{d}_{\phi_\mu}(\mathrm{T}_{\phi_\mu}^{\mu,\nu}, \mathrm{Id})$ with $\mathrm{T}_{\phi_\mu}^{\mu,\nu}$ defined as in Assumption 1. From there, we can conclude that $\phi_\mu$ satisfies Assumption 1. We notice that the corresponding transport problem recovers previously considered objects such as OT problems with Bregman divergence costs [25, 103], but is strictly more general (as our results pertain to the existence of OT maps), as detailed in Appendix D.1.

We now analyze the convergence of the MD scheme. Under a relative smoothness condition along curves generated by $\mathrm{S} = \mathrm{Id}$ and $\mathrm{T} = \mathrm{T}_{k+1}$ solutions of (4) for all $k \geq 0$, we derive the following descent lemma, which ensures that $\big(\mathcal{F}(\mu_k)\big)_k$ is non-increasing. Its proof can be found in Appendix H.3 and relies on the three-point inequality [29], which we extended to $L^2(\mu)$ in Lemma 29.

**Proposition 3.** *Let $\beta > 0$, $\tau \leq \frac{1}{\beta}$. Assume for all $k \geq 0$, $\mathcal{F}$ is $\beta$-smooth relative to $\phi$ along $t \mapsto \big((1-t)\mathrm{Id} + t\mathrm{T}_{k+1}\big)_{\#}\mu_k$, which implies $\beta\mathrm{d}_{\phi_{\mu_k}}(\mathrm{T}_{k+1}, \mathrm{Id}) \geq \mathrm{d}_{\tilde{\mathcal{F}}_{\mu_k}}(\mathrm{T}_{k+1}, \mathrm{Id})$. Then, for all $k \geq 0$,*

$$\mathcal{F}(\mu_{k+1}) \leq \mathcal{F}(\mu_k) - \frac{1}{\tau}\mathrm{d}_{\phi_{\mu_k}}(\mathrm{Id}, \mathrm{T}_{k+1}). \tag{7}$$

Assuming additionally the convexity of $\mathcal{F}$ along the curves $\mu_t = \big((1-t)\mathrm{Id} + t\mathrm{T}_{\phi_\mu}^{\mu,\nu}\big)_{\#}\mu$, $t \in [0, 1]$ and that $\phi$ satisfies Assumption 1, we can obtain global convergence.

**Proposition 4.** *Let $\nu \in \mathcal{P}_2(\mathbb{R}^d)$, $\alpha \geq 0$. Suppose Assumption 1 and the conditions of Proposition 3 hold, and that $\mathcal{F}$ is $\alpha$-convex relative to $\phi$ along the curves $t \mapsto \big((1-t)\mathrm{Id} + t\mathrm{T}_{\phi_{\mu_k}}^{\mu_k,\nu}\big)_{\#}\mu_k$. Then, for all $k \geq 1$,*

$$\mathcal{F}(\mu_k) - \mathcal{F}(\nu) \leq \frac{\alpha}{(1 - \tau\alpha)^{-k} - 1}W_\phi(\nu, \mu_0) \leq \frac{1 - \alpha\tau}{k\tau}W_\phi(\nu, \mu_0). \tag{8}$$

*Moreover, if $\alpha > 0$, taking $\nu = \mu^*$ the minimizer of $\mathcal{F}$, we obtain a linear rate: for all $k \geq 0$, $W_\phi(\mu^*, \mu_k) \leq (1 - \tau\alpha)^k W_\phi(\mu^*, \mu_0)$.*

The proof of Proposition 4 can be found in Appendix H.4, and requires Assumption 1 to hold so that consecutive distances between iterates and the global minimizer telescope. This is not as direct as in the proofs of [88] over $\mathbb{R}^d$, because the minimization problem of each iteration (4) happens in a different space $L^2(\mu_k)$. We discuss in Section 5 how to verify the relative smoothness and convexity on some examples. In particular, when both $\mathcal{F}$ and $\phi$ are potential energies, it is inherited from the relative smoothness and convexity on $\mathbb{R}^d$, and the conditions are similar with those for MD on $\mathbb{R}^d$. We also note that relative smoothness assumptions *along descent directions* as stated in Proposition 3 and relative strong convexity *along optimal curves between the iterates and a minimizer* as stated in Proposition 4 have been used already in the literature of optimization over measures in very specific cases, *e.g.* for descent results for the KL along SVGD [71] or for Sinkhorn convergence in [9]. We further analyze in Appendix F the convergence of Bregman proximal gradient scheme [11, 123] for objectives of the form $\mathcal{F}(\mu) = \mathcal{G}(\mu) + \mathcal{H}(\mu)$ with $\mathcal{H}$ non smooth; which includes the KL divergence decomposed as a potential energy plus the negative entropy.

**Implementation.** We now discuss the practical implementation of MD on $(\mathcal{P}_2(\mathbb{R}^d), W_2)$ as written in (5). If $\phi_\mu$ is pushforward compatible, we have $\nabla\phi_{\mu_k}(\mathrm{T}_{k+1}) = \nabla_{W_2}\phi\big((\mathrm{T}_{k+1})_{\#}\mu_k\big) \circ \mathrm{T}_{k+1}$; but if $\nabla\phi_{\mu_{k+1}}^*$ is unknown, the scheme is implicit in $\mathrm{T}_{k+1}$. A possible solution is to rely on a root finding algorithm such as Newton's method to find the zero of $\nabla\mathrm{J}$ at each step, which we use in Section 5 for $\phi_\mu^W$ as Bregman potential. However, this procedure may be computationally costly and scale badly *w.r.t.* the dimension and the number of samples, see Appendix G.1. Nonetheless, in the special case $\phi_\mu^V(\mathrm{T}) = \int V \circ \mathrm{T} \, \mathrm{d}\mu$ with $V$ differentiable, strongly convex and $L$-smooth, since $\nabla_{W_2}\mathcal{V}(\mu) = \nabla V$ and $(\nabla V)^{-1} = \nabla V^*$, the scheme reads as

$$\forall k \geq 0, \ \mathrm{T}_{k+1} = \nabla V^* \circ \big(\nabla V - \tau\nabla_{W_2}\mathcal{F}(\mu_k)\big), \tag{9}$$

and can be implemented on particles, *i.e.* for $\hat{\mu}_k = \frac{1}{n}\sum_{i=1}^n \delta_{x_i^k}$, $x_i^{k+1} = \nabla V^* \big(\nabla V(x_i^k) - \tau \nabla_{W_2}\mathcal{F}(\hat{\mu}_k)(x_i^k)\big)$ for all $k \geq 0$, $i \in \{1,\dots,n\}$. This scheme is analogous to MD in $\mathbb{R}^d$ [12] and has been introduced as the mirror Wasserstein gradient descent [113]. Moreover, for $V = \frac{1}{2}\|\cdot\|_2^2$, as observed earlier, we recover the usual Wasserstein gradient descent, *i.e.* $T_{k+1} = \mathrm{Id} - \tau\nabla_{W_2}\mathcal{F}(\mu_k)$. The scheme can also be implemented for Bregman potentials that are not pushforward compatible. For specific $\phi$, it recovers notably SVGD and its variants [83, 84, 114, 131] or the Kalman-Wasserstein gradient descent [56]. We refer to Appendix D.4 for more details.

## 4 Preconditioned gradient descent

As seen in Section 2, preconditioned gradient descent on $\mathbb{R}^d$ has dual convergence conditions compared to mirror descent. Our goal is to extend these to (1) and $\mathcal{P}_2(\mathbb{R}^d)$. Let $\tau > 0$, $\mu \in \mathcal{P}_2(\mathbb{R}^d)$ and $h : \mathbb{R}^d \to \mathbb{R}$ proper and strictly convex on $\mathbb{R}^d$. We consider in this section $\phi_\mu^h(T) = \int h \circ T \, \mathrm{d}\mu$ and $\mathrm{d}(T, \mathrm{Id}) = \phi_{\mu_k}^h\big((\mathrm{Id} - T)/\tau\big)\tau = \int h\big((x - T(x))/\tau\big)\tau \, \mathrm{d}\mu_k(x)$. This type of discrepancy is analogous to OT costs with translation-invariant ground cost $c(x,y) = h(x-y)$, which have been popular as they induce an OT map [110, Box 1.12]. Such costs have been introduced *e.g.* in [39, 70] to promote sparse transport maps. More generally, for $\phi_\mu$ strictly convex, proper, differentiable and superlinear, we have $(\nabla\phi_\mu)^{-1} = \nabla\phi_\mu^*$ and the following theory is still valid. For simplicity, we leave studying more general $\phi$ for future works. Here, the scheme (1) results in:

$$T_{k+1} = \operatorname*{argmin}_{T\in L^2(\mu_k)} \int h\left(\frac{x - T(x)}{\tau}\right)\tau \, \mathrm{d}\mu_k(x) + \langle\nabla_{W_2}\mathcal{F}(\mu_k), T - \mathrm{Id}\rangle_{L^2(\mu_k)}, \quad \mu_{k+1} = (T_{k+1})_\#\mu_k. \tag{10}$$

Deriving the first order conditions similarly to (5) in Section 3, we obtain the following update:

$$\forall k \geq 0, \ T_{k+1} = \mathrm{Id} - \tau(\nabla\phi_{\mu_k}^h)^{-1}\big(\nabla_{W_2}\mathcal{F}(\mu_k)\big) = \mathrm{Id} - \tau\nabla h^* \circ \nabla_{W_2}\mathcal{F}(\mu_k). \tag{11}$$

Notice that for $h = \frac{1}{2}\|\cdot\|_2^2$ the squared Euclidean norm, $\phi_\mu^h$ and $\phi_\mu^{h^*}$ recover the squared $L^2(\mu)$ norm, and schemes (4) and (10) coincide. The scheme (10) is analogous to preconditioned gradient descent [68, 75, 76, 89, 119], which provides a dual alternative to mirror descent. For the latter, the goal is to find a suitable preconditioner $h^*$ allowing to have convergence guarantees, or to speed-up the convergence for ill-conditioned problems. It was recently considered on the Wasserstein space by Cheng et al. [31] and Dong et al. [44] with a focus on the KL divergence as objective $\mathcal{F}$ and for $h = \|\cdot\|_p^p$ with $p > 1$ [31] or $h$ quadratic [44]. Moreover, their theoretical analysis [1] was mostly done using the continuous formulation. Instead we focus on deriving conditions for the convergence of the discrete-time scheme (11) for more general functionals objectives.

**Convergence guarantees.** Inspired by [89], we now provide a descent lemma on $\big(\phi_{\mu_k}^{h^*}(\nabla_{W_2}\mathcal{F}(\mu_k))\big)_k$ under a technical inequality between the Bregman divergences of $\phi_{\mu_k}^{h^*}$ and $\tilde{\mathcal{F}}_{\mu_k}$ for all $k \geq 0$. Additionally, we also suppose that $\mathcal{F}$ is convex along the curves generated by $S = T_{k+1}$ and $T = \mathrm{Id}$. This last hypothesis ensures that $\mathrm{d}_{\tilde{\mathcal{F}}_{\mu_k}}(T_{k+1}, \mathrm{Id}) \geq 0$, and thus that $\big(\phi_{\mu_k}^{h^*}(\nabla_{W_2}\mathcal{F}(\mu_k))\big)_k$ is non-increasing. Analogously to the Euclidean case, $\phi_\mu^{h^*}$ quantifies the magnitude of the gradient, and provides a second quantifier of convergence leading to possibly different efficient methods compared to mirror descent [68]. The proof relies mainly on the three-point identity (see *e.g.* [54, Appendix B.7] or Lemma 28) and algebra with the definition of Bregman divergences.

**Proposition 5.** *Let $\beta > 0$. Assume $\tau \leq \frac{1}{\beta}$, and for all $k \geq 0$, $\mathcal{F}$ convex along $t \mapsto \big((1-t)T_{k+1} + t\mathrm{Id}\big)_\#\mu_k$ and $\mathrm{d}_{\phi_{\mu_k}^{h^*}}\big(\nabla_{W_2}\mathcal{F}(\mu_{k+1}) \circ T_{k+1}, \nabla_{W_2}\mathcal{F}(\mu_k)\big) \leq \beta\mathrm{d}_{\tilde{\mathcal{F}}_{\mu_k}}(\mathrm{Id}, T_{k+1})$. Then, for all $k \geq 0$,*

$$\phi_{\mu_{k+1}}^{h^*}\big(\nabla_{W_2}\mathcal{F}(\mu_{k+1})\big) \leq \phi_{\mu_k}^{h^*}\big(\nabla_{W_2}\mathcal{F}(\mu_k)\big) - \frac{1}{\tau}\mathrm{d}_{\tilde{\mathcal{F}}_{\mu_k}}(T_{k+1}, \mathrm{Id}). \tag{12}$$

Under an additional assumption of a reverse inequality between the Bregman divergences of $\phi_{\mu_k}^{h^*}$ and $\tilde{\mathcal{F}}_{\mu_k}$, and assuming that $\phi_\mu^{h^*}$ attains its minimum in $0$, we can show the convergence of the gradient quantified by $\phi_\mu^{h^*}$ (see Lemma 21), and the convergence of $\big(\mathcal{F}(\mu_k)\big)_k$ towards the minimum of $\mathcal{F}$.

**Proposition 6.** *Let $\alpha \geq 0$ and $\mu^* \in \mathcal{P}_2(\mathbb{R}^d)$ be the minimizer of $\mathcal{F}$. Assume the conditions of Proposition 5 hold, and that for $\bar{T} = \operatorname{argmin}_{T, T_\#\mu_k = \mu^*} \mathrm{d}_{\tilde{\mathcal{F}}_{\mu_k}}(\mathrm{Id}, T), \quad \alpha\mathrm{d}_{\tilde{\mathcal{F}}_{\mu_k}}(\mathrm{Id}, \bar{T}) \leq$*

$\mathrm{d}_{\phi_{\mu_k}^{h^*}}\big(\nabla_{\mathrm{W}_2}\mathcal{F}(\bar{\mathrm{T}}_{\#}\mu_k)\circ\bar{\mathrm{T}},\nabla_{\mathrm{W}_2}\mathcal{F}(\mu_k)\big)$. *Then, for all $k \geq 1$, since $\nabla_{\mathrm{W}_2}\mathcal{F}(\mu^*) = 0$ and $\phi_{\mu_k}^{h^*}(0) = h^*(0)$,*

$$\phi_{\mu_k}^{h^*}\big(\nabla_{\mathrm{W}_2}\mathcal{F}(\mu_k)\big) - h^*(0) \leq \frac{\alpha}{(1-\tau\alpha)^{-k}-1}\big(\mathcal{F}(\mu_0)-\mathcal{F}(\mu^*)\big) \leq \frac{1-\tau\alpha}{\tau k}\big(\mathcal{F}(\mu_0)-\mathcal{F}(\mu^*)\big).$$
(13)

*Moreover, assuming that $h^*$ attains its minimum at $0$ and $\alpha > 0$, $\mathcal{F}$ converges towards its minimum at a linear rate, i.e. for all $k \geq 0$, $\mathcal{F}(\mu_k) - \mathcal{F}(\mu^*) \leq (1-\tau\alpha)^k\big(\mathcal{F}(\mu_0)-\mathcal{F}(\mu^*)\big)$.*

The proofs of Proposition 5 and Proposition 6 can be found respectively in Appendix H.5 and Appendix H.6.

We now discuss sufficient conditions to obtain the inequalities between the Bregman divergences required in Proposition 5 and Proposition 6. Maddison et al. [89] showed on $\mathbb{R}^d$ for a cost $h$ and an objective function $g$, that these conditions were equivalent to $\beta$-smoothness and $\alpha$-convexity of the preconditioner $h^*$ (analogous to $\phi_\mu^*$) relative to the convex conjugate of the objective $g^*$ (analogous to $\tilde{\mathcal{F}}_\mu^*$). To write the inequalities we assumed as a relative smoothness/convexity property of $\phi_{\mu_k}^{h^*}$ w.r.t. $\tilde{\mathcal{F}}_{\mu_k}^*$, we would need at least to ensure that $\tilde{\mathcal{F}}_{\mu_k}^*$ is differentiable, in order to define its Bregman divergence according to Definition 1. This can be done *e.g.* by assuming $\tilde{\mathcal{F}}_{\mu_k}$ strictly convex and superlinear (see Lemma 18). The latter is true for several examples of functionals $\mathcal{F}$ we already mentioned, such as potential or interaction energies with strongly convex potentials.

Under this assumption, we can show that the inequalities in Proposition 5 and Proposition 6 are implied by relative smoothness and convexity along suitable curves.

**Proposition 7.** *Let $\mu \in \mathcal{P}_2(\mathbb{R}^d)$ and $\mathrm{T} \in L^2(\mu)$. Assume $\tilde{\mathcal{F}}_\mu^*$ is Gâteaux differentiable and define $\mathcal{F}_\mu^*$ on $t \mapsto \mu_t = (\mathrm{T}_t)_{\#}\mu$ as $\mathcal{F}_\mu^*(\mu_t) = \tilde{\mathcal{F}}_\mu^*(\mathrm{T}_t)$ for $\mathrm{T}_t = (1-t)\mathrm{U} + t\mathrm{S}$ for all $t \in [0,1]$, $\mathrm{S}, \mathrm{U} \in L^2(\mu)$.*

*If $\phi^{h^*}$ is $\beta$-smooth relative to $\mathcal{F}_\mu^*$ along $t \mapsto \big((1-t)\nabla_{\mathrm{W}_2}\mathcal{F}(\mu) + t\nabla_{\mathrm{W}_2}\mathcal{F}(\mathrm{T}_{\#}\mu)\circ\mathrm{T}\big)_{\#}\mu$. Then,*

$$\mathrm{d}_{\phi_\mu^{h^*}}\big(\nabla_{\mathrm{W}_2}\mathcal{F}(\mathrm{T}_{\#}\mu)\circ\mathrm{T},\nabla_{\mathrm{W}_2}\mathcal{F}(\mu)\big) \leq \beta\mathrm{d}_{\tilde{\mathcal{F}}_\mu}(\mathrm{Id},\mathrm{T}).$$
(14)

*Likewise, if $\phi^{h^*}$ is $\alpha$-convex relative to $\mathcal{F}_\mu^*$ along $t \mapsto \big((1-t)\nabla_{\mathrm{W}_2}\mathcal{F}(\mu) + t\nabla_{\mathrm{W}_2}\mathcal{F}(\mathrm{T}_{\#}\mu)\circ\mathrm{T}\big)_{\#}\mu$, then*

$$\alpha\mathrm{d}_{\tilde{\mathcal{F}}_\mu}(\mathrm{Id},\mathrm{T}) \leq \mathrm{d}_{\phi_\mu^{h^*}}\big(\nabla_{\mathrm{W}_2}\mathcal{F}(\mathrm{T}_{\#}\mu)\circ\mathrm{T},\nabla_{\mathrm{W}_2}\mathcal{F}(\mu)\big).$$
(15)

In particular, for $\mathcal{F}$ a potential energy, the conditions coincide with those of [89] in $\mathbb{R}^d$. We refer to Appendix E.1 for more details.

## 5 Applications and Experiments

In this section, we first discuss how to verify the relative convexity and smoothness between functionals in practice. Then, we provide some examples of mirror descent and preconditioned gradient descent on different objectives. We refer to Appendix G for more details on the experiments[1].

**Relative convexity of functionals.** To assess relative convexity or smoothness as stated in Definition 3, we need to compare the Bregman divergences along the right curves. When both functionals $\phi$ and $\mathcal{F}$ are of the same type, for example potential (respectively interaction) energies, this property is lifted from the convexity and smoothness on $\mathbb{R}^d$ of the underlying potential functions (respectively interaction kernels) to $\mathcal{P}_2(\mathbb{R}^d)$, see Appendix E.2 for more details. When both $\phi$ and $\mathcal{F}$ are potential energies, the schemes (4) and (10) are equivalent to parallel MD and preconditioned GD since there are no interactions between the particles. The conditions of convergence then coincide with the ones obtained for MD and preconditioned GD on $\mathbb{R}^d$ [88, 89]. In other cases, (4) and (10) provide schemes that are novel to the best of our knowledge.

For functionals which are not of the same type, it is less straightforward. Using equivalent notions of convexity (see Proposition 13), we may instead compare their Hessians along the right curves,

---

[1]The code is available at `https://github.com/clbonet/Mirror_and_Preconditioned_Gradient_Descent_in_Wasserstein_Space`.

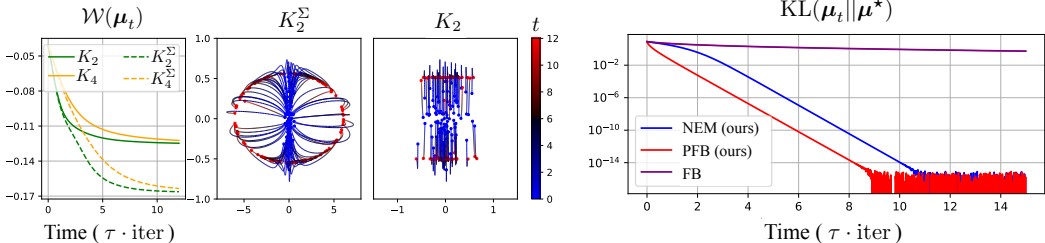

Figure 1: **(Left)** Value of $\mathcal{W}$ along the flow for two difference interaction Bregman potentials, **(Middle and Right)** Trajectories of particles to minimize $\mathcal{W}$.

Figure 2: Convergence towards Gaussians $\mathcal{N}(0, UDU^T)$ averaged over 20 covariances, with $U \sim \mathrm{Unif}(O_{10}(\mathbb{R}))$ and $D$ fixed.

see Appendix E.2 for an example between an interaction and a potential energy. We note also that for the particular case of a functional obtained as a sum $\mathcal{F} = \mathcal{G} + \mathcal{H}$ with $\tilde{\mathcal{G}}_\mu$ and $\tilde{\mathcal{H}}_\mu$ convex, since $\mathrm{d}_{\tilde{\mathcal{F}}_\mu} = \mathrm{d}_{\tilde{\mathcal{G}}_\mu} + \mathrm{d}_{\tilde{\mathcal{H}}_\mu}$, $\mathrm{d}_{\tilde{\mathcal{F}}_\mu} \geq \max\{\mathrm{d}_{\tilde{\mathcal{G}}_\mu}, \mathrm{d}_{\tilde{\mathcal{H}}_\mu}\}$, and thus $\mathcal{F}$ is 1-convex relative to $\mathcal{G}$ and $\mathcal{H}$. This includes *e.g.* the KL divergence which is convex relative to the potential and the negative entropy.

**MD on interaction energies.** We first focus on minimizing interaction energies $\mathcal{W}(\mu) = \frac{1}{2} \iint W(x - y) \, \mathrm{d}\mu(x) \mathrm{d}\mu(y)$ with kernel $W(z) = \frac{1}{4}\|z\|_{\Sigma^{-1}}^4 - \frac{1}{2}\|z\|_{\Sigma^{-1}}^2$, $\Sigma \in S_d^{++}(\mathbb{R})$, whose minimizer is an ellipsoid [27]. Since the Hessian norm of $W$ can be bounded by a polynomial of degree 2, following [88, Section 2], $W$ is $\beta$-smooth relative to $K_4(z) = \frac{1}{4}\|z\|_2^4 + \frac{1}{2}\|z\|_2^2$ with $\beta = 4$, and $\mathcal{W}$ is $\beta$-smooth relative to $\phi_\mu(\mathrm{T}) = \frac{1}{2}\iint K_4\big(\mathrm{T}(x) - \mathrm{T}(y)\big) \, \mathrm{d}\mu(x)\mathrm{d}\mu(y)$. Supposing additionally that the distributions are compactly supported, we can show that $\mathcal{W}$ is smooth relative to the interaction energy with $K_2(z) = \frac{1}{2}\|z\|_2^2$. For ill-conditioned $\Sigma$, *i.e.* for which the ratio between the largest and smallest eigenvalues is large, the convergence can be slow. Thus, we also propose to use $K_2^\Sigma(z) = \frac{1}{2}\|z\|_{\Sigma^{-1}}^2$ and $K_4^\Sigma(z) = \frac{1}{4}\|z\|_{\Sigma^{-1}}^4 + \frac{1}{2}\|z\|_{\Sigma^{-1}}^2$. We illustrate these mirror descent schemes on Figure 1 and observe the convergence we expect for the ones taking into account $\Sigma$. In practice, since $\nabla\phi_\mu(\mathrm{T}) = (\nabla K \star \mathrm{T}_{\#}\mu) \circ \mathrm{T}$, the scheme (5) needs to be approximated using Newton's algorithm which can be computationally heavy. Using $\phi_\mu^V(\mathrm{T}) = \int V \circ \mathrm{T} \, \mathrm{d}\mu$ with $V = K_2^\Sigma$, we obtain a more computationally friendly scheme with the same convergence, see Appendix G.2, but for which the smoothness is trickier to show.

**MD on KL.** We now focus on minimizing $\mathcal{F}(\mu) = \int V \mathrm{d}\mu + \mathcal{H}(\mu)$ for $V(x) = \frac{1}{2}x^T\Sigma^{-1}x$ with $\Sigma$ possibly ill-conditioned, whose minimizer is the Gaussian $\nu = \mathcal{N}(0, \Sigma)$, and for which Wasserstein gradient descent is slow to converge. We study the MD scheme in (4) with negative entropy $\mathcal{H}$ as the Bregman potential (NEM), and compare it on Figure 2 with the Forward-Backward (FB) scheme studied in [43] and the ideally preconditioned Forward-Backward scheme (PFB) with Bregman potential $\phi_\mu^V$ (see (116) in Appendix F). For computational purpose, we restrain the minimization in (4) over affine maps, which can be seen as taking the gradient over the submanifold of Gaussians [43, 73]. Starting from $\mathcal{N}(0, \Sigma_0)$, the distributions stay Gaussian over the flow, and their closed-form is reported in (62) (Appendix D.3). We note that this might not be the case for the scheme (4), and thus that this scheme does not enter into the framework developed in the previous sections. Nonetheless, it demonstrates the benefits of using different Bregman potentials. We generate 20 Gaussian targets $\nu$ on $\mathbb{R}^{10}$ with $\Sigma = UDU^T$, $D$ diagonal and scaled in log space between 1 and 100, and $U$ a uniformly sampled orthogonal matrix, and we report the averaged KL over time. Surprisingly, NEM, which does not require an ideal (and not available in general) preconditioner, is almost as fast to converge as the ideal PFB, and much faster than the FB scheme.

**Preconditioned GD for single-cells.** Predicting the response of cells to a perturbation is a central question in biology. In this context, as the measuring process is destructive, feature descriptions of control and treated cells must be dealt with as (unpaired) source $\mu$ and target distributions $\nu$. Following [112], OT theory to recover a mapping $\mathrm{T}$ between these two populations has been used in [21, 22, 23, 39, 48, 69, 122]. Inspired by the recent success of iterative refinement in generative modeling, through diffusion [61, 115] or flow-based models [81, 85], our scheme (1) follows the idea of transporting $\mu$ to $\nu$ via successive and dynamic displacements instead of, directly, with a static map $\bar{\mathrm{T}}$. We model the transition from unperturbed to perturbed states through the (preconditioned)

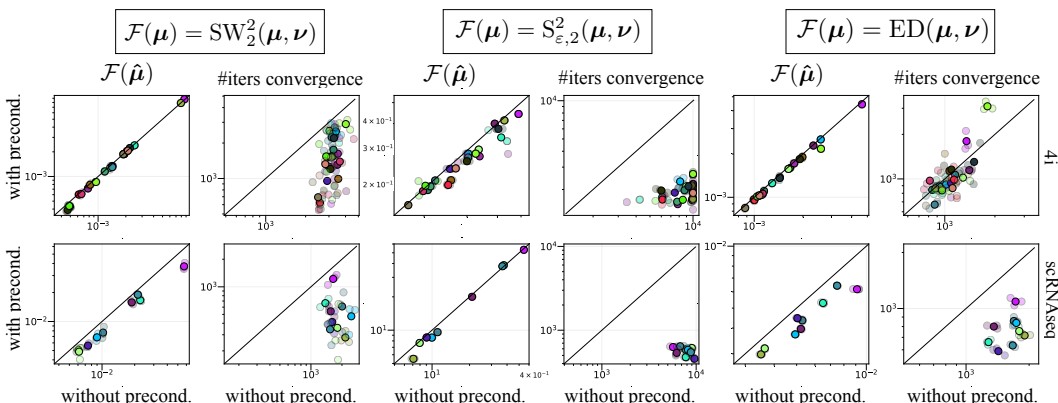

Figure 3: Preconditioned GD vs. (vanilla) GD to predict the responses of cell populations to cancer treatment on 4i (**Upper row**) and scRNAseq (**Lower row**) datasets. For each treatment, starting from the untreated cells $\mu_i$, we minimize $\mathcal{F}(\mu) = D(\mu, \nu_i)$ with $\nu_i$ the treated cells. The plot is organized as pairs of columns, each corresponding to optimizing a specific metric, with two scatter plots displaying points $z_i = (x_i, y_i)$ where (**First column**) $y_i$ is the attained minima $\mathcal{F}(\hat{\mu}) = D(\hat{\mu}, \nu_i)$ with preconditioning and $x_i$ that without preconditioning, and (**Second column**) $y_i$ is the number of iterations to reach convergence with preconditioning and $x_i$ that without preconditioning. A point below the diagonal $y = x$ then refers to an experiment in which preconditioning provides (**First column**) a better minima or (**Second column**) faster convergence. We assign a color to each treatment and plot three runs, obtained with three different initializations, along with their mean (brighter point).

gradient flow of a functional $\mathcal{F}(\mu) = D(\mu, \nu)$ initialized at $\mu_0 = \mu$, where $D$ is a distributional metric, and predict the perturbed population via $\hat{\mu} = \min_\mu \mathcal{F}(\mu)$. We focus on the datasets used in [21], consisting of cell lines analyzed using (i) 4i [58], and (ii) scRNA sequencing [118]. For each profiling technology, the response to respectively (i) 34 and (ii) 9 treatments are provided. As in [21], training is performed in data space for the 4i data and in a latent space learned by the scGen autoencoder [87] for the scRNA data. We use three metrics: the Sliced-Wasserstein distance $\mathrm{SW}_2^2$ [16], the Sinkhorn divergence $\mathrm{S}_{\varepsilon,2}^2$ [50] and the energy distance ED [59, 60, 105], and we compare the performances when minimizing this functional via preconditioned GD vs. (vanilla) GD. We measure the convergence speed when using a fixed relative tolerance tol $= 10^{-3}$, as well as the attained optimal value $\mathcal{F}(\hat{\mu})$. Note that we follow [21] and additionally consider 40% of unseen (test) target cells for evaluation, *i.e.*, for computing $\mathcal{F}(\hat{\mu}) = D(\hat{\mu}, \nu)$. As preconditioner, we use the one induced by $h^*(x) = (\|x\|_2^a + 1)^{1/a} - 1$ with $a > 0$, which is well suited to minimize functionals which grow in $\|x - x^*\|^{a/(a-1)}$ near their minimum [119]. We set the step size $\tau = 1$ for all the experiments. Then, we tune the parameter $a$ very simply: for a given metric $D$ and a profiling technology, we pick a random treatment and select $a \in \{1.25, 1.5, 1.75\}$ by grid search, and we generalize the selected $a$ for *all the other treatments*. Results are described in Figure 3: Preconditioned GD significantly outperforms GD over the 43 datasets, in terms of convergence speed and optimal value $\mathcal{F}(\hat{\mu})$. For instance, for $D = \mathrm{S}_{2,\varepsilon}^2$, we converge in 10 times less iterations while providing, on average, a better estimate of the treated population. We also compare our iterative (non parametric) approach with the use of a static (non parametric) map in Appendix G.4.

## 6   Conclusion

In this work, we extended two non-Euclidean optimization methods on $\mathbb{R}^d$ to the Wasserstein space, generalizing $\mathrm{W}_2$-gradient descent to alternative geometries. We investigated the practical benefits of these schemes, and provided rates of convergences for pairs of objectives and Bregman potentials satisfying assumptions of relative smoothness and convexity along specific curves. While these assumptions can be easily checked is some cases (*e.g.* potential or interaction energies) by comparing the Bregman divergences or Hessian operators in the Wasserstein geometry, they may be hard to verify in general. Different objectives such as the Sliced-Wasserstein distance or the Sinkhorn divergence, or alternative geometries to the Wasserstein-2 as studied in this work, require to derive specific computations on a case-by-case basis. We leave this investigation for future work.

## Acknowledgments and Disclosure of Funding

Clément Bonet acknowledges the support of the center Hi! PARIS and of ANR PEPR PDE-AI. Adam David gratefully acknowledges funding by the BMBF 01|S20053B project SALE. Pierre-Cyril Aubin-Frankowski was funded by the FWF project P 36344-N. Anna Korba acknowledges the support of ANR-22-CE23-0030.

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

# Appendix

The appendix is organized as follows. In Appendix A, we discuss related works. In Appendix B and Appendix C, we provide mathematical background respectively on $L^2$ and on the Wasserstein space. In Appendix D, we provide complementary results for the mirror descent scheme on the Wasserstein space. In Appendix E, we discuss the relative convexity and smoothness between functionals. In Appendix F, we study the Bregman proximal gradient scheme, deriving a convergence result and closed-form updates for the Gaussian case. In Appendix G, we provide more details on the experiments. In Appendix H, we report the proofs of our results. Finally, auxiliary results are given in Appendix I.

## Contents

## A  Related works

**Mirror descent on $\mathbb{R}^d$.** Mirror descent has been introduced by Nemirovskij and Yudin [92] to solve convex optimization problems. Its convergence has been first studied under the assumption that the objective has a Lipschitz gradient, see *e.g.* [12]. More recently, Bauschke et al. [11], Lu et al. [88] provided convergence guarantees by assuming relative smoothness and convexity.

**Preconditioned gradient descent on $\mathbb{R}^d$.** The preconditioned gradient descent has first been studied by Maddison et al. [89], providing convergence guarantees under assumptions on the smoothness and convexity of the preconditioner relative to the Legendre transform of the objective. Kim et al. [68] underlined connections with the mirror descent, and introduced an accelerated version of the preconditioned gradient descent. Laude and Patrinos [75], Laude et al. [76] studied a generalized version of this algorithm by minimizing an anisotropic upper bound and supposing anisotropic smoothness of the objective. In particular, their analysis for the descent lemma is also valid for a non-convex smooth objective. Tarmoun et al. [119] also studied preconditioned gradient descent for non-Lipschitz smooth non-convex problems.

**Wasserstein Gradient flows with respect to non-Euclidean geometries.** Several existing schemes are based on time-discretizations of gradient flows with respect to optimal transport metrics, but different than the Wasserstein-2 distance.

To simplify the computation of the backward scheme, Peyré [100] added an entropic regularization into the JKO scheme while Bonet et al. [14] considered using the Sliced-Wasserstein distance instead. More recently, Rankin and Wong [104] suggested using Bregman divergences *e.g.* when geodesic distances are not known in closed-forms.

The most popular objective in Wasserstein gradient flows is the KL. However, this can be intricate to compute as it requires the evaluation of the density at each step, which is not known for particles, and thus requires approximations using kernel density estimators [127] or density ratio estimators [6, 49, 128]. Restricting the velocity field to a reproducing kernel Hilbert space (RKHS), an update in closed-form can be obtained, which is given by the SVGD algorithm [83, 84]. This algorithm

can also be seen as using an alternative Wasserstein metric [46]. However, the restriction to RKHS can hinder the flexibility of the method. This motivated the introduction of new schemes based on using the Wasserstein distance with a convex translation invariant cost [31, 44]. Particle systems preconditioned by they empirical covariance matrix have also been recently considered, and can be seen as discretization of the Kalman-Wasserstein or Covariance Modulated gradient flow [24, 56].

**Mirror descent with flat geometry.** The space of probability distributions can be endowed with different metrics. When endowed with the Fisher-Rao metric instead of the Wasserstein distance, the geometry becomes very different. Notably, the shortest path between the two distributions is now a mixture between them. In this situation, the gradient is the first variation. Aubin-Frankowski et al. [9] studied the mirror descent in this space and notably showed connections with the Sinkhorn algorithm when the mirror map and the optimized functionals are KL divergences. Karimi et al. [67] extended the mirror descent algorithm for more general time steps, and notably recovered the "Wasserstein Mirror Flow" proposed by Deb et al. [41] as a special case.

**Bregman divergence on $\mathcal{P}_2(\mathbb{R}^d)$.** Several works introduced Bregman divergences on $\mathcal{P}_2(\mathbb{R}^d)$. Carlier and Jimenez [25] first studied the existence of Monge maps for the OT problem with Bregman costs $c(x, y) = \mathrm{d}_V(x, y)$ and symmetrized Bregman costs $c(x, y) = \mathrm{d}_V(x, y) + \mathrm{d}_V(y, x)$. For Bregman costs, the resulting OT problem was named the Bregman-Wasserstein divergence and its properties were studied in [37, 57, 103]. The Bregman-Wasserstein divergence has also been used by Ahn and Chewi [3] to show the convergence of the Mirror Langevin algorithm while Rankin and Wong [104] studied its JKO scheme with KL objective. Li [80] introduced the notion of Bregman divergence on Wasserstein space for a geodesically strictly convex $\mathcal{F} : \mathcal{P}_2(\mathbb{R}^d) \to \mathbb{R}$ as

$$\forall \mu, \nu \in \mathcal{P}_2(\mathbb{R}^d), \ \mathrm{d}_{\mathcal{F}}(\mu, \nu) = \mathcal{F}(\mu) - \mathcal{F}(\nu) - \langle \nabla_{\mathrm{W}_2} \mathcal{F}(\nu), \mathrm{T}_\nu^\mu - \mathrm{Id} \rangle_{L^2(\nu)}, \tag{16}$$

where $\mathrm{T}_\nu^\mu$ is the OT map between $\nu$ and $\mu$ *w.r.t* $\mathrm{W}_2$. The Bregman divergence used in our work and as defined in Definition 1 is more general as it allows using more general maps and contains as special case (16). Li [80] studied properties of this Bregman divergence for different functionals $\mathcal{F}$ and provided closed-forms for one-dimensional distributions or Gaussian, but did not use it to define a mirror scheme.

**Mirror descent on $\mathcal{P}_2(\mathbb{R}^d)$.** Deb et al. [41] defined a mirror flow by using the continuous formulation. They focused on KL objectives with Bregman potential $\phi(\mu) = \frac{1}{2} \mathrm{W}_2^2(\mu, \nu)$ with some reference measure $\nu \in \mathcal{P}_2(\mathbb{R}^d)$, and defined the flow as the solution of

$$\begin{cases} \varphi(\mu_t) = \nabla_{\mathrm{W}_2} \phi(\mu_t) \\ \frac{\mathrm{d}}{\mathrm{d}t} \varphi(\mu_t) = -\nabla_{\mathrm{W}_2} \mathcal{F}(\mu_t). \end{cases} \tag{17}$$

We note that $\phi$ is pushforward compatible and hence enters our framework. Also related to our work, Wang and Li [126] studied a Wasserstein Newton's flow, which, analogously to the relation between Newton's method and mirror descent [32], is another discretization of our scheme for $\phi = \mathcal{F}$. We clarify the link with the Mirror Descent algorithm we define in this work with the previous continuous formulation above in Appendix D.2.

# B Background on $L^2(\mu)$

## B.1 Differential calculus on $L^2(\mu)$

We recall some differentiability definitions on the Hilbert space $L^2(\mu)$ for $\mu \in \mathcal{P}_2(\mathbb{R}^d)$. Let $\phi : L^2(\mu) \to \mathbb{R}$. We start by recalling the notions of Gâteaux and Fréchet derivatives.

**Definition 4.** *A function $\phi : L^2(\mu) \to \mathbb{R}$ is said to be Gâteaux differentiable at $\mathrm{T}$ if there exists an operator $\phi'(\mathrm{T}) : L^2(\mu) \to \mathbb{R}$ such that for any direction $h \in L^2(\mu)$,*

$$\phi'(\mathrm{T})(h) = \lim_{t \to 0} \frac{\phi(\mathrm{T} + th) - \phi(\mathrm{T})}{t}, \tag{18}$$

*and $\phi'(\mathrm{T})$ is a linear function. The operator $\phi'(\mathrm{T})$ is called the Gâteaux derivative of $\phi$ at $\mathrm{T}$ and if it exists, it is unique.*

**Definition 5.** *The Fréchet derivative of $\phi$ at $\mathrm{T} \in L^2(\mu)$ in the direction $h \in L^2(\mu)$, denoted $\delta\phi(\mathrm{T}, h)$, is defined implicitly by*

$$\phi(\mathrm{T} + th) = \phi(\mathrm{T}) + t\delta\phi(\mathrm{T}, h) + to(\|h\|). \tag{19}$$

If $\phi$ is Fréchet differentiable, then it is also Gâteaux differentiable, and both derivatives agree, *i.e.* for all $\mathrm{T}, h \in L^2(\mu)$, $\delta\phi(\mathrm{T}, h) = \phi'(\mathrm{T})(h)$ [99, Proposition 1.26].

Moreover, since $L^2(\mu)$ is a Hilbert space, and $\delta\phi(\mathrm{T}, \cdot)$ and $\phi'(\mathrm{T})$ are linear and continuous, if $\phi$ is Fréchet (resp. Gâteaux) differentiable, by the Riesz representation theorem, there exists $\nabla\phi \in L^2(\mu)$ such that for all $h \in L^2(\mu)$, $\delta\phi(\mathrm{T}, h) = \langle \nabla\phi(\mathrm{T}), h \rangle_{L^2(\mu)}$ (resp. $\phi'(\mathrm{T})(h) = \langle \nabla\phi(\mathrm{T}), h \rangle_{L^2(\mu)}$).

As a brief comment on these notions in the context of convexity, if the subdifferential of a convex $f$ at $x$ contains a single element then it is the Gâteaux derivative and we have an inequality $f(y) \geq f(x) + \langle \nabla f(x), y - x \rangle$. Instead Fréchet différentiability gives an equality (19) corresponding to a series expansion.

## B.2 Convexity on $L^2(\mu)$

Let $\phi : L^2(\mu) \to \mathbb{R}$ be Gâteaux differentiable. We recall that $\phi$ is convex if for all $t \in [0, 1]$, $\mathrm{T}, \mathrm{S} \in L^2(\mu)$,

$$\phi\big((1 - t)\mathrm{T} + t\mathrm{S}\big) \leq (1 - t)\phi(\mathrm{T}) + t\phi(\mathrm{S}), \tag{20}$$

which is equivalent by [99, Proposition 3.10] with

$$\forall \mathrm{T}, \mathrm{S} \in L^2(\mu), \ \phi(\mathrm{T}) \geq \phi(\mathrm{S}) + \langle \nabla\phi(\mathrm{S}), \mathrm{T} - \mathrm{S} \rangle_{L^2(\mu)} \iff \mathrm{d}_\phi(\mathrm{T}, \mathrm{S}) \geq 0. \tag{21}$$

We now present equivalent definitions of the relative smoothness and relative convexity, which is the equivalent of [88, Proposition 1.1].

**Proposition 8.** *Let $\psi, \phi : L^2(\mu) \to \mathbb{R}$ be convex and Gâteaux differentiable functions. The following conditions are equivalent:*

- *(a1)* $\psi$ *is $\beta$-smooth relative to $\phi$*

- *(a2)* $\beta\phi - \psi$ *is a convex function on $L^2(\mu)$*

- *(a3)* *If twice Gâteaux differentiable, $\langle \nabla^2\psi(\mathrm{T})\mathrm{S}, \mathrm{S} \rangle_{L^2(\mu)} \leq \beta \langle \nabla^2\phi(\mathrm{T})\mathrm{S}, \mathrm{S} \rangle_{L^2(\mu)}$ for all $\mathrm{T}, \mathrm{S} \in L^2(\mu)$*

- *(a4)* $\langle \nabla\psi(\mathrm{T}) - \nabla\psi(\mathrm{S}), \mathrm{T} - \mathrm{S} \rangle_{L^2(\mu)} \leq \beta \langle \nabla\phi(\mathrm{T}) - \nabla\phi(\mathrm{S}), \mathrm{T} - \mathrm{S} \rangle_{L^2(\mu)}$ *for all $\mathrm{T}, \mathrm{S} \in L^2(\mu)$.*

*The following conditions are equivalent:*

- *(b1)* $\psi$ *is $\alpha$-convex relative to $\phi$*

- *(b2)* $\psi - \alpha\phi$ *is a convex function on $L^2(\mu)$*

- *(b3)* *If twice differentiable, $\langle \nabla^2\psi(\mathrm{T})\mathrm{S}, \mathrm{S} \rangle_{L^2(\mu)} \geq \alpha \langle \nabla^2\phi(\mathrm{T})\mathrm{S}, \mathrm{S} \rangle_{L^2(\mu)}$ for all $\mathrm{T}, \mathrm{S} \in L^2(\mu)$*

- *(b4)* $\langle \nabla\psi(\mathrm{T}) - \nabla\psi(\mathrm{S}), \mathrm{T} - \mathrm{S} \rangle_{L^2(\mu)} \geq \alpha \langle \nabla\phi(\mathrm{T}) - \nabla\phi(\mathrm{S}), \mathrm{T} - \mathrm{S} \rangle_{L^2(\mu)}$ *for all $\mathrm{T}, \mathrm{S} \in L^2(\mu)$.*

*Proof.* We do it only for the smoothness. It holds likewise for the convexity.

**(a1) $\iff$ (a2):**

$$\forall \mathrm{T}, \mathrm{S} \in L^2(\mu), \ \mathrm{d}_\psi(\mathrm{T}, \mathrm{S}) \leq \beta \mathrm{d}_\phi(\mathrm{T}, \mathrm{S})$$
$$\iff \forall \mathrm{T}, \mathrm{S} \in L^2(\mu), \ \psi(\mathrm{T}) - \psi(\mathrm{S}) - \langle \nabla\psi(\mathrm{S}), \mathrm{T} - \mathrm{S} \rangle_{L^2(\mu)}$$
$$\leq \beta\big(\phi(\mathrm{T}) - \phi(\mathrm{S}) - \langle \nabla\phi(\mathrm{S}), \mathrm{T} - \mathrm{S} \rangle_{L^2(\mu)}\big) \tag{22}$$
$$\iff \forall \mathrm{T}, \mathrm{S} \in L^2(\mu), \ (\beta\phi - \psi)(\mathrm{S}) - \langle \nabla(\beta\phi - \psi)(\mathrm{S}), \mathrm{T} - \mathrm{S} \rangle_{L^2(\mu)} \leq (\beta\phi - \psi)(\mathrm{T}).$$

For the rest of the equivalences, we apply [99, Proposition 3.10]. Indeed, $\beta\phi - \psi$ convex is equivalent to

$$\forall \mathrm{T}, \mathrm{S} \in L^2(\mu), \ \langle \nabla(\beta\phi - \psi)(\mathrm{T}) - \nabla(\beta\phi - \psi)(\mathrm{S}), \mathrm{T} - \mathrm{S} \rangle_{L^2(\mu)} \geq 0$$
$$\iff \forall \mathrm{T}, \mathrm{S} \in L^2(\mu), \ \beta \langle \nabla\phi(\mathrm{T}) - \nabla\phi(\mathrm{S}), \mathrm{T} - \mathrm{S} \rangle_{L^2(\mu)} \geq \langle \nabla\psi(\mathrm{T}) - \nabla\psi(\mathrm{S}), \mathrm{T} - \mathrm{S} \rangle_{L^2(\mu)}, \tag{23}$$

which gives the equivalence between (a2) and (a4). And if $\psi$ and $\phi$ are twice differentiables, it is also equivalent to

$$\forall T, S \in L^2(\mu), \ \langle \nabla^2(\beta\phi - \psi)(T)S, S\rangle_{L^2(\mu)} \geq 0$$
$$\Longleftrightarrow \ \forall T, S \in L^2(\mu), \ \beta\langle\nabla^2\phi(T)S, S\rangle_{L^2(\mu)} \geq \langle\nabla^2\psi(T)S, S\rangle_{L^2(\mu)}, \quad (24)$$

which gives the equivalence between (a2) and (a3). $\square$

## C   Background on Wasserstein space

### C.1   Wasserstein differentials

We recall the notion of Wasserstein differentiability introduced in [5, 17, 74]. First, we introduce sub- and super-differentials.

**Definition 6** (Wasserstein sub- and super-differential [17, 74]). *Let $\mathcal{F} : \mathcal{P}_2(\mathbb{R}^d) \to (-\infty, +\infty]$ lower semicontinuous and denote $D(\mathcal{F}) = \{\mu \in \mathcal{P}_2(\mathbb{R}^d), \ \mathcal{F}(\mu) < \infty\}$. Let $\mu \in D(\mathcal{F})$. Then, a map $\xi \in L^2(\mu)$ belongs to the subdifferential $\partial^-\mathcal{F}(\mu)$ of $\mathcal{F}$ at $\mu$ if for all $\nu \in \mathcal{P}_2(\mathbb{R}^d)$,*

$$\mathcal{F}(\nu) \geq \mathcal{F}(\mu) + \sup_{\gamma \in \Pi_o(\mu,\nu)} \int \langle\xi(x), y - x\rangle \, d\gamma(x,y) + o\big(W_2(\mu,\nu)\big). \quad (25)$$

*Similarly, $\xi \in L^2(\mu)$ belongs to the superdifferential $\partial^+\mathcal{F}(\mu)$ of $\mathcal{F}$ at $\mu$ if $-\xi \in \partial^-(-\mathcal{F})(\mu)$.*

Then, we say that a functional is Wasserstein differentiable if it admits sub and super differentials which coincide.

**Definition 7** (Wasserstein differentiability, Definition 2.3 in [74]). *A functional $\mathcal{F} : \mathcal{P}_2(\mathbb{R}^d) \to \mathbb{R}$ is Wasserstein differentiable at $\mu \in \mathcal{P}_2(\mathbb{R}^d)$ if $\partial^-\mathcal{F}(\mu) \cap \partial^+\mathcal{F}(\mu) \neq \emptyset$. In this case, we say that $\nabla_{W_2}\mathcal{F}(\mu) \in \partial^-\mathcal{F}(\mu) \cap \partial^+\mathcal{F}(\mu)$ is a Wasserstein gradient of $\mathcal{F}$ at $\mu$, satisfying for any $\nu \in \mathcal{P}_2(\mathbb{R}^d)$, $\gamma \in \Pi_o(\mu,\nu)$,*

$$\mathcal{F}(\nu) = \mathcal{F}(\mu) + \int \langle\nabla_{W_2}\mathcal{F}(\mu)(x), y - x\rangle \, d\gamma(x,y) + o\big(W_2(\mu,\nu)\big). \quad (26)$$

Recall that the tangent space of $\mathcal{P}_2(\mathbb{R}^d)$ at $\mu \in \mathcal{P}_2(\mathbb{R}^d)$ is defined as

$$\mathcal{T}_\mu\mathcal{P}_2(\mathbb{R}^d) = \overline{\{\nabla\psi, \ \psi \in \mathcal{C}_c^\infty(\mathbb{R}^d)\}} \subset L^2(\mu), \quad (27)$$

where the closure is taken in $L^2(\mu)$, see Ambrosio et al. [5, Definition 8.4.1]. Lanzetti et al. [74, Proposition 2.5] showed that if $\mathcal{F}$ is Wasserstein differentiable, then there is always a unique gradient living in the tangent space, and we can restrict ourselves without loss of generality to this gradient.

Lanzetti et al. [74] further showed that Wasserstein gradients provide linear approximations even if the perturbations are not induced by OT plans, *i.e.* differentials are "strong Fréchet differentials".

**Proposition 9** (Proposition 2.6 in [74]). *Let $\mu, \nu \in \mathcal{P}_2(\mathbb{R}^d)$, $\gamma \in \Pi(\mu,\nu)$ any coupling and let $\mathcal{F} : \mathcal{P}_2(\mathbb{R}^d) \to \mathbb{R}$ be Wasserstein differentiable at $\mu$ with Wasserstein gradient $\nabla_{W_2}\mathcal{F}(\mu) \in \mathcal{T}_\mu\mathcal{P}_2(\mathbb{R}^d)$. Then,*

$$\mathcal{F}(\nu) = \mathcal{F}(\mu) + \int \langle\nabla_{W_2}\mathcal{F}(\mu)(x), y - x\rangle \, d\gamma(x,y) + o\left(\sqrt{\int \|x - y\|_2^2 \, d\gamma(x,y)}\right). \quad (28)$$

Under regularity assumptions, the Wasserstein gradient of $\mathcal{F}$ can be computed in practice using the first variation $\frac{\delta\mathcal{F}}{\delta\mu}$ [110, Definition 7.12], which is defined, if it exists, as the unique function (up to a constant) such that, for $\chi$ satisfying $\int d\chi = 0$,

$$\frac{d}{dt}\mathcal{F}(\mu + t\chi)\Big|_{t=0} = \lim_{t\to 0} \frac{\mathcal{F}(\mu + t\chi) - \mathcal{F}(\mu)}{t} = \int \frac{\delta\mathcal{F}}{\delta\mu}(\mu) \, d\chi. \quad (29)$$

Then the Wasserstein gradient can be computed as $\nabla_{W_2}\mathcal{F}(\mu) = \nabla\frac{\delta\mathcal{F}}{\delta\mu}(\mu)$, see *e.g.* [34, Proposition 5.10].

We now show that we can relate the Fréchet derivative of $\tilde{\mathcal{F}}_\mu(T) := \mathcal{F}(T_\#\mu)$ with the Wasserstein gradient of $\mathcal{F}$ belonging to the tangent space of $\mathcal{P}_2(\mathbb{R}^d)$ at $\mu$.

**Proposition 10.** *Let $\mathcal{F} : \mathcal{P}_2(\mathbb{R}^d) \to \mathbb{R} \cup \{+\infty\}$ be a Wasserstein differentiable functional on $D(\mathcal{F})$. Let $\mu \in \mathcal{P}_2(\mathbb{R}^d)$ and $\tilde{\mathcal{F}}_\mu(\mathrm{T}) = \mathcal{F}(\mathrm{T}_{\#}\mu)$ for all $\mathrm{T} \in D(\tilde{\mathcal{F}}_\mu)$. Then, $\tilde{\mathcal{F}}_\mu$ is Fréchet differentiable, and for all $\mathrm{S} \in D(\tilde{\mathcal{F}}_\mu)$, $\nabla \tilde{\mathcal{F}}_\mu(\mathrm{S}) = \nabla_{\mathrm{W}_2} \mathcal{F}(\mathrm{S}_{\#}\mu) \circ \mathrm{S}$.*

*Proof.* See Appendix H.1. □

A similar formula can be found in Gangbo and Tudorascu [55, Corollary 3.22], however the space $H$ used there is not $L^2(\mu)$ but a lifting $L^2(\Omega; \mathbb{R}^d)$ of measures on random variables. They should not be confused.

## C.2 Wasserstein Hessians

A natural object of interest in the context of optimization over the Wasserstein space is the Hessian of the objective $\mathcal{F}$, which we define below, according to the original definitions of [97, Section 3] and [124, Chapter 8]. This notion is usually defined along Wasserstein geodesics.

**Definition 8** (Chapter 8 in [124]). *The Wasserstein Hessian of $\mathcal{F}$, denoted $\mathrm{H}\mathcal{F}_\mu$ is an operator over $\mathcal{T}_\mu \mathcal{P}_2(\mathbb{R}^d)$ verifying $\langle \mathrm{H}\mathcal{F}_\mu v_0, v_0 \rangle_{L^2(\mu)} = \frac{\mathrm{d}^2}{\mathrm{d}t^2} \mathcal{F}(\rho_t)\big|_{t=0}$ if $(\rho_t, v_t)_{t \in [0,1]}$ is a Wasserstein geodesic starting at $\mu$.*

If $\mu$ is absolutely continuous, Wassertein geodesics starting from $\mu$ are curves of the form $\rho_t = (\mathrm{Id} + t\nabla\psi)_{\#}\mu$ for $\psi \in \mathcal{C}_c^\infty(\mathbb{R}^d)$. Using this fact, one can compute the Wasserstein Hessians of Kullback–Leibler divergence [97], Maximum Mean Discrepancy [7] or Kernel Stein Discrepancy [72] and many other functionals.

However in this work, we are interested in more general curves, which we call acceleration free, *i.e.* $\mu_t = (\mathrm{S} + tv)_{\#}\mu$ with $\mathrm{S}, v \in L^2(\mu)$. Thus, we define analogously the Hessian along such curves.

**Definition 9.** *We define the Hessian operator $\mathrm{H}\mathcal{F}_{\mu,t} : L^2(\mu) \to L^2(\mu)$ as the operator satisfying for all $t \in [0,1]$,*

$$\frac{\mathrm{d}^2}{\mathrm{d}t^2} \mathcal{F}(\mu_t) = \langle \mathrm{H}\mathcal{F}_{\mu,t} v, v \rangle_{L^2(\mu)}, \tag{30}$$

*where $t \mapsto \mu_t = (\mathrm{S} + tv)_{\#}\mu$ for $\mathrm{S}, v \in L^2(\mu)$.*

Note that the Hessian $\mathrm{H}\mathcal{F}_{\mu,t}$ is taken at time $t$ but that the vector field $v \in L^2(\mu)$ is in the tangent space at $t = 0$, hence the discrepancy with Definition 8 besides the fact that we can have $\mathrm{S} \neq \mathrm{Id}$.

Wang and Li [126] derived a general closed form of the Wasserstein Hessian on tangent spaces through the first variation of $\mathcal{F}$. Here, we extend their formula along any curve $\mu_t = (\mathrm{S} + tv)_{\#}\mu$ with $\mathrm{S}, v \in L^2(\mu)$. We first provide a lemma computing the derivative of the Wasserstein gradient.

**Lemma 11.** *Let $\mathcal{F} : \mathcal{P}_2(\mathbb{R}^d) \to \mathbb{R}$ be twice continuously differentiable and assume that $\frac{\delta}{\delta\mu} \nabla \frac{\delta\mathcal{F}}{\delta\mu}(\mu) = \nabla \frac{\delta^2\mathcal{F}}{\delta\mu^2}(\mu)$ for all $\mu \in \mathcal{P}_2(\mathbb{R}^d)$. Let $\mu \in \mathcal{P}_2(\mathbb{R}^d)$ and for all $t \in [0,1]$, $\mu_t = (\mathrm{T}_t)_{\#}\mu$ where $\mathrm{T}_t$ is differentiable w.r.t. $t$ with $\frac{\mathrm{d}\mathrm{T}_t}{\mathrm{d}t} \in L^2(\mu)$. Then,*

$$\frac{\mathrm{d}}{\mathrm{d}t}(\nabla_{\mathrm{W}_2}\mathcal{F}(\mu_t) \circ \mathrm{T}_t)(x) = \int \left[ \nabla_y \nabla_x \frac{\delta^2\mathcal{F}}{\delta\mu^2}\big((\mathrm{T}_t)_{\#}\mu\big)\big(\mathrm{T}_t(x), \mathrm{T}_t(y)\big) \frac{\mathrm{d}\mathrm{T}_t}{\mathrm{d}t}(y) \right] \mathrm{d}\mu(y)$$
$$+ \nabla^2 \frac{\delta\mathcal{F}}{\delta\mu}\big((\mathrm{T}_t)_{\#}\mu\big)\big(\mathrm{T}_t(x)\big) \frac{\mathrm{d}\mathrm{T}_t}{\mathrm{d}t}(x). \tag{31}$$

*Proof.* See Appendix H.8. □

This allows us to define a closed-form for $\mathrm{H}\mathcal{F}_{\mu,t}$.

**Proposition 12.** *Under the same assumptions as in Lemma 11, let $\mu_t = (\mathrm{T}_t)_{\#}\mu$ with $\mathrm{T}_t = \mathrm{S} + tv$, $\mathrm{S}, v \in L^2(\mu)$, then $\mathrm{H}\mathcal{F}_{\mu,t} : L^2(\mu) \to L^2(\mu)$ (as defined in Definition 9) is defined for all $v \in L^2(\mu)$, $x \in \mathbb{R}^d$ as:*

$$\mathrm{H}\mathcal{F}_{\mu,t}[v](x) = \int \nabla_y \nabla_x \frac{\delta^2\mathcal{F}}{\delta\mu^2}\big((\mathrm{T}_t)_{\#}\mu\big)\big(\mathrm{T}_t(x), \mathrm{T}_t(y)\big) v(y) \, \mathrm{d}\mu(y) + \nabla^2 \frac{\delta\mathcal{F}}{\delta\mu}\big((\mathrm{T}_t)_{\#}\mu\big)\big(\mathrm{T}_t(x)\big) v(x). \tag{32}$$

*Proof.* See Appendix H.9. □

While $\mathrm{H}\mathcal{F}_{\mu,t}$ and $\mathrm{H}\mathcal{F}_{\mu_t}$ differ in general, in some simple cases their relation boils down to composition with a pushforward. For instance, if $\mathrm{S}$ is invertible, we can write $\mu_t$ as a curve starting from $\mathrm{S}_{\#}\mu$ with a velocity field $v \circ \mathrm{S}^{-1}$, *i.e.* $\mu_t = (\mathrm{Id} + tv \circ \mathrm{S}^{-1})_{\#}\mathrm{S}_{\#}\mu$. Thus, we recover the original definition of the Wasserstein Hessian at $t = 0$ as $\mathrm{H}\mathcal{F}_{\mu,0} = \mathrm{H}\mathcal{F}_{\mathrm{S}_{\#}\mu}$. However, in general, this does not need to be the case.

Similarly, if $\mathrm{T}_t$ is invertible for all $t$, setting $v_t = v \circ \mathrm{T}_t^{-1}$, we can write

$$
\begin{aligned}
\frac{\mathrm{d}^2}{\mathrm{d}t^2}\mathcal{F}(\mu_t) &= \langle \mathrm{H}\mathcal{F}_{\mu,t}v, v\rangle_{L^2(\mu)} \\
&= \int \langle \mathrm{H}\mathcal{F}_{\mu,t}[v](x), v(x)\rangle \, \mathrm{d}\mu(x) \\
&= \int \langle \mathrm{H}\mathcal{F}_{\mu,t}[v]\big(\mathrm{T}_t^{-1}(x_t)\big), v_t(x_t)\rangle \, \mathrm{d}\mu_t(x_t) \\
&= \langle \mathrm{H}\mathcal{F}_{\mu_t}v_t, v_t\rangle_{L^2(\mu_t)}.
\end{aligned}
\tag{33}
$$

The last line is obtained by leveraging the closed form in Proposition 12 and that $\mu_t = (\mathrm{T}_t)_{\#}\mu$, as for all $x \in \mathrm{supp}(\mu)$,

$$
\begin{aligned}
\mathrm{H}\mathcal{F}_{\mu,t}[v](x) &= \int \nabla_y \nabla_x \frac{\delta^2 \mathcal{F}}{\delta\mu^2}(\mu_t)\big(\mathrm{T}_t(x), \mathrm{T}_t(y)\big)v(y) \, \mathrm{d}\mu(y) + \nabla^2 \frac{\delta\mathcal{F}}{\delta\mu}(\mu_t)\big(\mathrm{T}_t(x)\big)v(x) \\
&= \int \nabla_y \nabla_x \frac{\delta^2 \mathcal{F}}{\delta\mu^2}(\mu_t)(\mathrm{T}_t(x), y_t)v_t(y_t) \, \mathrm{d}\mu_t(y) + \nabla^2 \frac{\delta\mathcal{F}}{\delta\mu}(\mu_t)\big(\mathrm{T}_t(x)\big)v(x) \\
&= \mathrm{H}\mathcal{F}_{\mu_t}[v_t]\big(\mathrm{T}_t(x)\big),
\end{aligned}
\tag{34}
$$

and thus $\mathrm{H}\mathcal{F}_{\mu,t}[v]\big(\mathrm{T}_t^{-1}(x)\big) = \mathrm{H}\mathcal{F}_{\mu_t}[v_t](x)$.

Here are two examples of $\mathcal{F}$ satisfying $\frac{\delta}{\delta\mu}\nabla\frac{\delta\mathcal{F}}{\delta\mu} = \nabla\frac{\delta^2\mathcal{F}}{\delta\mu^2}$ for which Proposition 12 provides an expression of the Wasserstein Hessian.

**Example 1** (Potential energy). *Let $\mathcal{V}(\mu) = \int V \, \mathrm{d}\mu$ with $V$ twice differentiable with bounded Hessian. Then, we have $\frac{\delta\mathcal{V}}{\delta\mu}(\mu) = V$ and $\frac{\delta^2\mathcal{V}}{\delta\mu^2}(\mu) = 0$ (using (29)). Thus, applying Proposition 12, we recover for $\mu_t = (\mathrm{T}_t)_{\#}\mu$,*

$$
\frac{\mathrm{d}^2}{\mathrm{d}t^2}\mathcal{V}(\mu_t) = \int \big\langle \nabla^2 V\big(\mathrm{T}_t(x)\big)v(x), v(x)\big\rangle \, \mathrm{d}\mu(x).
\tag{35}
$$

**Example 2** (Interaction energy). *Let $\mathcal{W}(\mu) = \frac{1}{2}\iint W(x-y) \, \mathrm{d}\mu(x)\mathrm{d}\mu(y)$ with $W$ symmetric, twice differentiable and with bounded Hessian. Then, we have for all $x, y \in \mathbb{R}^d$, $\frac{\delta\mathrm{W}}{\delta\mu}(\mu)(x) = (W \star \mu)(x)$ and $\frac{\delta^2\mathrm{W}}{\delta\mu^2}(\mu)(x,y) = W(x-y)$ (see e.g. [126, Example 7]), and thus applying Proposition 12, for $\mu_t = (\mathrm{T}_t)_{\#}\mu$, the operator is*

$$
\mathrm{H}\mathcal{W}_{\mu,t}[v](x) = -\int \nabla^2 W\big(\mathrm{T}_t(x) - \mathrm{T}_t(y)\big)v(y) \, \mathrm{d}\mu(y) + (\nabla^2 W \star (\mathrm{T}_t)_{\#}\mu)(\mathrm{T}_t(x))v(x),
\tag{36}
$$

*and*

$$
\frac{\mathrm{d}^2}{\mathrm{d}t^2}\mathcal{W}(\mu_t) = \iint \big\langle \nabla^2 W\big(\mathrm{T}_t(x) - \mathrm{T}_t(y)\big)\big(v(x) - v(y)\big), v(x)\big\rangle \, \mathrm{d}\mu(y)\mathrm{d}\mu(x).
\tag{37}
$$

### C.3 Convexity in Wasserstein space

We first recall the definition of $\alpha$-convex functionals [5, Definition 9.1.1].

**Definition 10.** *$\mathcal{F}$ is $\alpha$-convex along geodesics if for all $\mu_0, \mu_1 \in \mathcal{P}_2(\mathbb{R}^d)$,*

$$
\forall t \in [0,1], \ \mathcal{F}(\mu_t) \le (1-t)\mathcal{F}(\mu_0) + t\mathcal{F}(\mu_1) - \alpha\frac{t(1-t)}{2}\mathrm{W}_2^2(\mu_0, \mu_1),
\tag{38}
$$

*where $(\mu_t)_{t \in [0,1]}$ is a Wasserstein geodesic between $\mu_0$ and $\mu_1$.*

If we want to derive the minimal set of assumptions for the convergence of the gradient descent algorithms on Wasserstein space, we can actually restrict the smoothness and convexity to specific curves. In the next proposition, we characterize the convexity along one curve. The relative smoothness or convexity follows by considering the convexity of respectively $\beta\mathcal{G} - \mathcal{F}$ or $\mathcal{F} - \alpha\mathcal{G}$.

**Proposition 13.** *Let* $\mathcal{F} : \mathcal{P}_2(\mathbb{R}^d) \to \mathbb{R}$ *be twice continuously differentiable. Let* $\mu \in \mathcal{P}_2(\mathbb{R}^d)$, $\mathrm{T}, \mathrm{S} \in L^2(\mu)$, $\mu_t = (\mathrm{T}_t)_{\#}\mu$ *for all* $t \in [0, 1]$ *where* $\mathrm{T}_t = (1 - t)\mathrm{S} + t\mathrm{T}$. *Furthermore, denote for* $t_1, t_2 \in [0, 1]$, $\tilde{\mu}_t^{t_1 \to t_2} = ((1 - t)\mathrm{T}_{t_1} + t\mathrm{T}_{t_2})_{\#}\mu$. *Then, the following statement are equivalent:*

**(c1)** *For all* $t_1, t_2, t \in [0, 1]$, $\mathcal{F}(\tilde{\mu}_t^{t_1 \to t_2}) \leq (1 - t)\mathcal{F}((\mathrm{T}_{t_1})_{\#}\mu) + t\mathcal{F}((\mathrm{T}_{t_2})_{\#}\mu)$, *i.e.* $\mathcal{F}$ *is convex along* $t \mapsto \mu_t$.

**(c2)** *For all* $t_1, t_2 \in [0, 1]$, *we have* $\mathrm{d}_{\tilde{\mathcal{F}}_\mu}(\mathrm{T}_{t_2}, \mathrm{T}_{t_1}) \geq 0$, *i.e.*

$$\mathcal{F}((\mathrm{T}_{t_2})_{\#}\mu) - \mathcal{F}((\mathrm{T}_{t_1})_{\#}\mu) - \langle\nabla_{\mathrm{W}_2}\mathcal{F}((\mathrm{T}_{t_1})_{\#}\mu) \circ \mathrm{T}_{t_1}, \mathrm{T}_{t_2} - \mathrm{T}_{t_1}\rangle_{L^2(\mu)} \geq 0.$$

**(c3)** *For all* $t_1, t_2 \in [0, 1]$,

$$\langle\nabla_{\mathrm{W}_2}\mathcal{F}((\mathrm{T}_{t_2})_{\#}\mu) \circ \mathrm{T}_{t_2} - \nabla_{\mathrm{W}_2}\mathcal{F}((\mathrm{T}_{t_1})_{\#}\mu) \circ \mathrm{T}_{t_1}, \mathrm{T}_{t_2} - \mathrm{T}_{t_1}\rangle_{L^2(\mu)} \geq 0.$$

**(c4)** *For all* $s \in [0, 1]$, $\frac{\mathrm{d}^2}{\mathrm{d}t^2}\mathcal{F}(\mu_t)\big|_{t=s} \geq 0$.

*Proof.* See Appendix H.10. □

As stated in Section 2, if we require the convexity to hold along all curves with $\mathrm{S} = \mathrm{Id}$ and $\mathrm{T}$ the gradient of some convex function, *i.e.* an OT map, then $\mathcal{F}$ is convex along geodesics. Likewise, if the convexity holds for all $\mathrm{S}, \mathrm{T}$ that are gradients of convex functions, then we obtain the convexity along generalized geodesics.

If we require the convexity and the smoothness to hold along any curve of the form $\mu_t = ((1 - t)\mathrm{S} + t\mathrm{T})_{\#}\mu$ for $\mu \in \mathcal{P}_2(\mathbb{R}^d)$ and $\mathrm{T}, \mathrm{S} \in L^2(\mu)$, then it coincides with the transport convexity and smoothness recently introduced by Tanaka [117, Definitions 4.1 and 4.5]. As by Proposition 1, $\delta\tilde{\mathcal{F}}_\mu(\mathrm{S}, \mathrm{T} - \mathrm{S}) = \langle\nabla_{\mathrm{W}_2}\mathcal{F}(\mathrm{S}_{\#}\mu) \circ \mathrm{S}, \mathrm{T} - \mathrm{S}\rangle_{L^2(\mu)}$, and thus the convexity of $\mathcal{F}$ on such a curve reads as follows

$$\mathrm{d}_{\tilde{\mathcal{F}}_\mu}(\mathrm{T}, \mathrm{S}) = \mathcal{F}(\mathrm{T}_{\#}\mu) - \mathcal{F}(\mathrm{S}_{\#}\mu) - \langle\nabla_{\mathrm{W}_2}\mathcal{F}(\mathrm{S}_{\#}\mu) \circ \mathrm{S}, \mathrm{T} - \mathrm{S}\rangle_{L^2(\mu)} \geq 0. \tag{39}$$

And for $\tilde{\mathcal{G}}_\mu(\mathrm{T}) = \frac{1}{2}\|\mathrm{T}\|_{L^2(\mu)}$, the $\beta$-smoothness of $\mathcal{F}$ relative to $\mathcal{G}$ expresses as

$$\mathrm{d}_{\tilde{\mathcal{F}}_\mu}(\mathrm{T}, \mathrm{S}) = \mathcal{F}(\mathrm{T}_{\#}\mu) - \mathcal{F}(\mathrm{S}_{\#}\mu) - \langle\nabla_{\mathrm{W}_2}\mathcal{F}(\mathrm{S}_{\#}\mu) \circ \mathrm{S}, \mathrm{T} - \mathrm{S}\rangle_{L^2(\mu)} \leq \frac{\beta}{2}\|\mathrm{T} - \mathrm{S}\|_{L^2(\mu)} = \beta\mathrm{d}_{\tilde{\mathcal{G}}_\mu}(\mathrm{T}, \mathrm{S}). \tag{40}$$

This type of convexity is actually a particular case of the notion of convexity along acceleration-free curves introduced by Parker [98] (also introduced by Cavagnari et al. [28] under the name of total convexity). The latter requires convexity to hold along any curve of the form $\mu_t = ((1 - t)\pi^1 + t\pi^2)_{\#}\gamma$ with $\gamma \in \Pi(\mu, \nu)$, $\mu, \nu \in \mathcal{P}_2(\mathbb{R}^d)$ and $\pi^1(x, y) = x$, $\pi^2(x, y) = y$. The transport convexity of Tanaka [117] is thus a particular case for couplings obtained through maps, *i.e.* $\gamma = (\mathrm{T}, \mathrm{S})_{\#}\mu$. Parker [98] notably showed that this notion of convexity is equivalent to the geodesic convexity for Wasserstein differentiable functionals.

We can also define the strict convexity using strict inequalities in Proposition 13-**(c1)**-**(c2)**-**(c3)**, but not in **(c4)** as there are counter examples already for real functions. For instance, $f : \mathbb{R} \to \mathbb{R}$, defined as $f(x) = x^4$ for all $x \in \mathbb{R}$, is strictly convex but $f''(0) = 0$. Thus, for $\mathcal{F}(\mu) = \int f \, \mathrm{d}\mu$, choosing $\mu = \delta_0$ and $\mathrm{T}_0 = \mathrm{Id}$, by Example 1, we have that $\frac{\mathrm{d}^2}{\mathrm{d}t^2}\mathcal{F}(\mu_t)\big|_{t=0} = f''(0)v(0)^2 = 0$. But $\mathcal{F}(\mu_t) = f(t\mathrm{T}_1(0)) < tf(\mathrm{T}_1(0))$ since $f$ is strictly convex and thus $\mathcal{F}$ is well strictly convex.

Finally, as we defined the relative $\alpha$-convexity and $\beta$-smoothness of $\mathcal{F}$ relative to $\mathcal{G}$ using Bregman divergences in Definition 3, we can show that it is equivalent to $\mathcal{F} - \alpha\mathcal{G}$ and $\beta\mathcal{G} - \mathcal{F}$ being convex.

**Proposition 14.** *Let $\mathcal{F}, \mathcal{G} : \mathcal{P}_2(\mathbb{R}^d) \to \mathbb{R}$ be two differentiable functionals. Let $\mu \in \mathcal{P}_2(\mathbb{R}^d)$, $\mathrm{T}, \mathrm{S} \in L^2(\mu)$ and for all $t \in [0,1]$, $\mu_t = (\mathrm{T}_t)_{\#}\mu$ with $\mathrm{T}_t = (1-t)\mathrm{S} + t\mathrm{T}$. Then, $\mathcal{F}$ is $\alpha$-convex (resp. $\beta$-smooth) relative to $\mathcal{G}$ along $t \mapsto \mu_t$ if and only if $\mathcal{F} - \alpha\mathcal{G}$ (resp. $\beta\mathcal{G} - \mathcal{F}$) is convex along $t \mapsto \mu_t$.*

*Proof.* By Definition 3, $\mathcal{F}$ is $\alpha$-convex relative to $\mathcal{G}$ along $t \mapsto \mu_t$ if for all $s, t \in [0,1]$, $\mathrm{d}_{\tilde{\mathcal{F}}_\mu}(\mathrm{T}_s, \mathrm{T}_t) \geq \alpha\mathrm{d}_{\tilde{\mathcal{G}}_\mu}(\mathrm{T}_s, \mathrm{T}_t)$. This is equivalent to $\mathrm{d}_{\tilde{\mathcal{F}}_\mu - \alpha\tilde{\mathcal{G}}_\mu}(\mathrm{T}_s, \mathrm{T}_t) \geq 0$, which is equivalent by Proposition 13 **(c2)** (and using Proposition 1) to $\mathcal{F} - \alpha\mathcal{G}$ convex along $t \mapsto \mu_t$. The result for the $\beta$-smoothness follows likewise. $\square$

# D    Additional results on mirror descent

## D.1    Optimal transport maps for mirror descent

Let $\phi : \mathcal{P}_2(\mathbb{R}^d) \to \mathbb{R}$ be a strictly convex functional along all acceleration-free curves $\mu_t = (\mathrm{T}_t)_{\#}\mu$, $t \in [0,1]$ with $\mathrm{T}_t = (1-t)\mathrm{S} + t\mathrm{T}$ for $\mathrm{T}, \mathrm{S} \in L^2(\mu)$. Denote for $\mu \in L^2(\mu)$, $\phi_\mu(\mathrm{T}) = \phi(\mathrm{T}_{\#}\mu)$. Since $\phi$ is strictly convex along all acceleration-free curves, by Proposition 13, for all $\mathrm{T} \neq \mathrm{S} \in L^2(\mu)$, $\mathrm{d}_{\phi_\mu}(\mathrm{T}, \mathrm{S}) > 0$ and thus $\phi_\mu$ is strictly convex. Indeed, recall that

$$\forall \mathrm{T}, \mathrm{S} \in L^2(\mu), \; \mathrm{d}_{\phi_\mu}(\mathrm{T}, \mathrm{S}) = \phi_\mu(\mathrm{T}) - \phi_\mu(\mathrm{S}) - \langle \nabla\phi_\mu(\mathrm{S}), \mathrm{T} - \mathrm{S} \rangle_{L^2(\mu)}$$
$$= \phi(\mathrm{T}_{\#}\mu) - \phi(\mathrm{S}_{\#}\mu) - \langle \nabla_{\mathrm{W}_2}\phi(\mathrm{S}_{\#}\mu) \circ \mathrm{S}, \mathrm{T} - \mathrm{S} \rangle_{L^2(\mu)}, \tag{41}$$

where we used Proposition 1 for the computation of the gradient.

Let us now define for all $\mu, \nu \in \mathcal{P}_2(\mathbb{R}^d)$,

$$\mathrm{W}_\phi(\nu, \mu) = \inf_{\gamma \in \Pi(\nu, \mu)} \phi(\nu) - \phi(\mu) - \int \langle \nabla_{\mathrm{W}_2}\phi(\mu)(y), x - y \rangle \, \mathrm{d}\gamma(x, y). \tag{42}$$

This problem encompasses several previously considered objects, as discussed in more detail in Remark 1. Our motivation for introducing Equation (42) is to prove that for $\phi_\mu$ verifying the assumptions of Proposition 2, its associated Bregman divergence $\mathrm{d}_{\phi_\mu}$ satisfies the property given in Assumption 1. First, we can observe that as $\gamma = (\mathrm{T}, \mathrm{S})_{\#}\mu \in \Pi(\mathrm{T}_{\#}\mu, \mathrm{S}_{\#}\mu)$, we have $\mathrm{d}_{\phi_\mu}(\mathrm{T}, \mathrm{S}) \geq \mathrm{W}_\phi(\mathrm{T}_{\#}\mu, \mathrm{S}_{\#}\mu)$. Then, for $\mu \in \mathcal{P}_{2,\mathrm{ac}}(\mathbb{R}^d)$, assuming that $\nabla_{\mathrm{W}_2}\phi(\mu) = \nabla\phi_\mu(\mathrm{Id})$ is invertible, we can leverage Brenier's theorem [20], and show in Proposition 15 that the optimal coupling of Equation (42) is of the form $(\mathrm{T}_{\phi_\mu}^{\mu,\nu}, \mathrm{Id})_{\#}\mu$ with $\mathrm{T}_{\phi_\mu}^{\mu,\nu} = \mathrm{argmin}_{\mathrm{T}_{\#}\mu=\nu} \; \mathrm{d}_{\phi_\mu}(\mathrm{T}, \mathrm{Id})$. Moreover, if $\nu \in \mathcal{P}_{2,\mathrm{ac}}(\mathbb{R}^d)$, we also have that $\mathrm{T}_{\phi_\mu}^{\mu,\nu}$ is invertible with inverse $\bar{\mathrm{T}}_{\phi_\nu}^{\nu,\mu} = \mathrm{argmin}_{\mathrm{T}_{\#}\nu=\mu} \; \mathrm{d}_{\phi_\nu}(\mathrm{Id}, \mathrm{T})$.

**Proposition 15.** *Let $\mu \in \mathcal{P}_{2,\mathrm{ac}}(\mathbb{R}^d)$, $\nu \in \mathcal{P}_2(\mathbb{R}^d)$ and assume $\nabla_{\mathrm{W}_2}\phi(\mu)$ invertible. Then,*

1. *There exists a unique minimizer $\gamma$ of (42). Besides, there exists a uniquely determined $\mu$-almost everywhere (a.e.) map $\mathrm{T}_{\phi_\mu}^{\mu,\nu} : \mathbb{R}^d \to \mathbb{R}^d$ such that $\gamma = (\mathrm{T}_{\phi_\mu}^{\mu,\nu}, \mathrm{Id})_{\#}\mu$. Finally, there exists a convex function $u : \mathbb{R}^d \to \mathbb{R}$ such that $\mathrm{T}_{\phi_\mu}^{\mu,\nu} = \nabla u \circ \nabla_{\mathrm{W}_2}\phi(\mu)$ $\mu$-a.e.*

2. *Assume further that $\nu \in \mathcal{P}_{2,\mathrm{ac}}(\mathbb{R}^d)$. Then there exists a uniquely determined $\nu$-a.e. map $\bar{\mathrm{T}}_{\phi_\nu}^{\nu,\mu} : \mathbb{R}^d \to \mathbb{R}^d$ such that $\gamma = (\mathrm{Id}, \bar{\mathrm{T}}_{\phi_\nu}^{\nu,\mu})_{\#}\nu$. Moreover, there exists a convex function $v : \mathbb{R}^d \to \mathbb{R}$ such that $\bar{\mathrm{T}}_{\phi_\nu}^{\nu,\mu} = \nabla_{\mathrm{W}_2}\phi(\mu)^{-1} \circ \nabla v$ $\nu$-a.e., and $\mathrm{T}_{\phi_\mu}^{\mu,\nu} \circ \bar{\mathrm{T}}_{\phi_\nu}^{\nu,\mu} = \mathrm{Id}$ $\nu$-a.e. and $\bar{\mathrm{T}}_{\phi_\nu}^{\nu,\mu} \circ \mathrm{T}_{\phi_\mu}^{\mu,\nu} = \mathrm{Id}$ $\mu$-a.e.*

3. *As a corollary, $\mathrm{W}_\phi(\nu, \mu) = \min_{\mathrm{T}_{\#}\mu=\nu} \; \mathrm{d}_{\phi_\mu}(\mathrm{T}, \mathrm{Id}) = \min_{\mathrm{T}_{\#}\nu=\mu} \; \mathrm{d}_{\phi_\nu}(\mathrm{Id}, \mathrm{T})$.*

*Proof.* 1. Observe that problem (42) is equivalent to

$$\inf_{\gamma \in \Pi(\nu, \mu)} \int \|x - \nabla_{\mathrm{W}_2}\phi(\mu)(y)\|_2^2 \, \mathrm{d}\gamma(x, y). \tag{43}$$

Then, since for any $\gamma \in \Pi(\nu, \mu)$, $\big(\mathrm{Id}, \nabla_{\mathrm{W}_2}\phi(\mu)\big)_{\#}\gamma \in \Pi\big(\nu, \nabla_{\mathrm{W}_2}\phi(\mu)_{\#}\mu\big)$, we have

$$\inf_{\gamma \in \Pi(\nu, \mu)} \int \|x - \nabla_{\mathrm{W}_2}\phi(\mu)(y)\|_2^2 \, \mathrm{d}\gamma(x, y) \geq \inf_{\tilde{\gamma} \in \Pi\big(\nu, \nabla_{\mathrm{W}_2}\phi(\mu)_{\#}\mu\big)} \int \|x - z\|_2^2 \, \mathrm{d}\tilde{\gamma}(x, z). \tag{44}$$

Let $\mu \in \mathcal{P}_{2,\mathrm{ac}}(\mathbb{R}^d)$. Since $\nabla_{\mathrm{W}_2}\phi(\mu)$ is invertible, $\nabla_{\mathrm{W}_2}\phi(\mu)_{\#}\mu \in \mathcal{P}_{2,\mathrm{ac}}(\mathbb{R}^d)$. By Brenier's theorem, there exists a convex function $u$ such that $(\nabla u)_{\#}(\nabla_{\mathrm{W}_2}\phi(\mu))_{\#}\mu = \nu$ and the optimal coupling is of the form $\tilde{\gamma}^* = (\nabla u, \mathrm{Id})_{\#}\nabla_{\mathrm{W}_2}\phi(\mu)_{\#}\mu$. Let $\gamma = (\nabla u \circ \nabla_{\mathrm{W}_2}\phi(\mu), \mathrm{Id})_{\#}\mu \in \Pi(\nu, \mu)$, then

$$
\begin{aligned}
\int \|z - \tilde{y}\|_2^2 \,\mathrm{d}\tilde{\gamma}^*(z, \tilde{y}) &= \int \|\nabla u\big(\nabla_{\mathrm{W}_2}\phi(\mu)(y)\big) - \nabla_{\mathrm{W}_2}\phi(\mu)(y)\|_2^2 \,\mathrm{d}\mu(y) \\
&= \int \|x - \nabla_{\mathrm{W}_2}\phi(\mu)(y)\|_2^2 \,\mathrm{d}\gamma(x, y).
\end{aligned}
\tag{45}
$$

Thus, since $\gamma \in \Pi(\nu, \mu)$, $\gamma$ is an optimal coupling for (42).

2. We symmetrize the arguments. Assuming $\nu \in \mathcal{P}_{2,\mathrm{ac}}(\mathbb{R}^d)$ and $\nabla\phi_\mu(\mathrm{Id}) = \nabla_{\mathrm{W}_2}\phi(\mu)$ invertible, by Brenier's theorem, there exists a convex function $v$ such that $(\nabla v)_{\#}\nu = \nabla_{\mathrm{W}_2}\phi(\mu)_{\#}\mu$ (and such that $\nabla u \circ \nabla v = \mathrm{Id}$ $\nu$-a.e. and $\nabla v \circ \nabla u = \mathrm{Id}$ $\nabla_{\mathrm{W}_2}\phi(\mu)_{\#}\mu$-a.e.) and the optimal coupling is of the form $\tilde{\gamma}^* = (\mathrm{Id}, \nabla v)_{\#}\nu$. Let $\gamma = (\mathrm{Id}, \nabla_{\mathrm{W}_2}\phi(\mu)^{-1} \circ \nabla v)_{\#}\nu \in \Pi(\nu, \mu)$, then

$$
\begin{aligned}
\int \|x - z\|_2^2 \,\mathrm{d}\tilde{\gamma}^*(x, z) &= \int \|x - \nabla v(x)\|_2^2 \,\mathrm{d}\nu(x) \\
&= \int \|x - \nabla_{\mathrm{W}_2}\phi(\mu)\big((\nabla_{\mathrm{W}_2}\phi(\mu))^{-1}(\nabla v(x))\big)\|_2^2 \,\mathrm{d}\nu(x) \\
&= \int \|x - \nabla_{\mathrm{W}_2}\phi(\mu)(y)\|_2^2 \,\mathrm{d}\gamma(x, y).
\end{aligned}
\tag{46}
$$

Thus, since $\gamma \in \Pi(\nu, \mu)$, $\gamma$ is an optimal coupling for (42). Moreover, noting $\mathrm{T}_{\phi_\mu}^{\mu,\nu} = \nabla u \circ \nabla_{\mathrm{W}_2}\phi(\mu)$ and $\bar{\mathrm{T}}_{\phi_\nu}^{\nu,\mu} = \nabla_{\mathrm{W}_2}\phi(\mu)^{-1} \circ \nabla v$, we have $\mu$-a.e., $\bar{\mathrm{T}}_{\phi_\nu}^{\nu,\mu} \circ \mathrm{T}_{\phi_\mu}^{\mu,\nu} = \nabla_{\mathrm{W}_2}\phi(\mu)^{-1} \circ \nabla v \circ \nabla u \circ \nabla_{\mathrm{W}_2}\phi(\mu) = \mathrm{Id}$ and $\nu$-a.e., $\mathrm{T}_{\phi_\mu}^{\mu,\nu} \circ \bar{\mathrm{T}}_{\phi_\nu}^{\nu,\mu} = \nabla u \circ \nabla_{\mathrm{W}_2}\phi(\mu) \circ \nabla_{\mathrm{W}_2}\phi(\mu)^{-1} \circ \nabla v = \mathrm{Id}$ from the aforementioned consequences of Brenier's theorem. $\qquad\square$

We continue this section with additional results relative to the invertibility of mirror maps, which are required in Proposition 2. For a potential energy $\mathcal{V}(\mu) = \int V \,\mathrm{d}\mu$, since $\nabla_{\mathrm{W}_2}\mathcal{V} = \nabla V$, then $\nabla_{\mathrm{W}_2}\mathcal{V}$ is invertible provided $\nabla V$ is. This is the case *e.g.* for $V$ strictly convex. We now state in the two next lemmas conditions for an interaction energy and for the negative entropy to satisfy the invertibility requirements.

**Lemma 16.** *Let $\mu \in \mathcal{P}_2(\mathbb{R}^d)$ and let $W : \mathbb{R}^d \to \mathbb{R}$ be even, $\epsilon$-strongly convex for $\epsilon > 0$ and differentiable. Then, for $\mathcal{W}(\mu) = \iint W(x - y) \,\mathrm{d}\mu(x)\mathrm{d}\mu(y)$, $\nabla_{\mathrm{W}_2}\mathcal{W}(\mu)$ is invertible.*

*Proof.* On one hand, $\nabla_{\mathrm{W}_2}\mathcal{W}(\mu) = \nabla W \star \mu$. Moreover, $W$ $\epsilon$-strongly convex is equivalent to

$$
\forall x, y \in \mathbb{R}^d,\ x \neq y,\ \langle \nabla W(x) - \nabla W(y), x - y\rangle \geq \epsilon\|x - y\|_2^2,
\tag{47}
$$

which implies for all $x, y, z \in \mathbb{R}^d$, $\langle \nabla W(x - z) - \nabla W(y - z), x - y\rangle \geq \epsilon\|x - y\|_2^2$. By integrating with respect to $\mu$, it implies

$$
\langle (\nabla W \star \mu)(x) - (\nabla W \star \mu)(y), x - y\rangle = \int \langle \nabla W(x - z) - \nabla W(y - z), x - y\rangle \,\mathrm{d}\mu(z) \geq \epsilon\|x - y\|_2^2.
\tag{48}
$$

Thus, $\nabla W \star \mu$ is $\epsilon$-strongly monotone, and in particular invertible [2, Theorem 1]. $\qquad\square$

**Lemma 17.** *Let $\mu \in \mathcal{P}_{2,\mathrm{ac}}(\mathbb{R}^d)$ such that its density is of the form $\rho \propto e^{-V}$ with $V : \mathbb{R}^d \to \mathbb{R}$ $\epsilon$-strongly convex for $\epsilon > 0$. Then, for $\mathcal{H}(\mu) = \int \log\big(\rho(x)\big) \,\mathrm{d}\mu(x)$ with $\rho$ the density of $\mu$ w.r.t the Lebesgue measure, $\nabla_{\mathrm{W}_2}\mathcal{H}(\mu)$ is invertible.*

*Proof.* Let $\mu$ such distribution. Then, $\nabla_{\mathrm{W}_2}\mathcal{H}(\mu) = \nabla \log \rho = -\nabla V$. Since $V$ is $\epsilon$-strongly convex, then $\nabla V$ is $\epsilon$-strongly monotone and in particular invertible [2, Theorem 1]. $\qquad\square$

We conclude this section with a discussion of (42) with respect to related work.

**Remark 1.** *The OT problem* (42) *recovers other OT costs for specific choices of $\phi$. For instance, for $\phi_\mu(\mathrm{T}) = \frac{1}{2}\|\mathrm{T}\|^2_{L^2(\mu)}$, it coincides with the squared Wasserstein-2 distance. And more generally, for $\phi^V_\mu(\mathrm{T}) = \int V \circ \mathrm{T}\,\mathrm{d}\mu$, since by Lemma 31, for all $\mathrm{T}, \mathrm{S} \in L^2(\mu)$,*

$$d_{\phi^V_\mu}(\mathrm{T}, \mathrm{S}) = \int d_V\big(\mathrm{T}(x), \mathrm{S}(x)\big)\,\mathrm{d}\mu(x), \tag{49}$$

*where $d_V$ is the Euclidean Bregman divergence,* i.e. *for all $x, y \in \mathbb{R}^d$, $d_V(x, y) = V(x) - V(y) - \langle \nabla V(y), x - y \rangle$, $\mathrm{W}_\phi$ coincides with the Bregman-Wasserstein divergence [103]*

$$\mathcal{B}_V(\mu, \nu) = \inf_{\gamma \in \Pi(\mu, \nu)} \int d_V(x, y)\,\mathrm{d}\gamma(x, y). \tag{50}$$

## D.2 Continuous formulation

Let $\phi : L^2(\mu) \to \mathbb{R}$ be pushforward compatible and superlinear. Introducing the (mirror) map $\varphi(\mu) = \nabla_{\mathrm{W}_2}\phi(\mu)$, we can write informally the mirror descent scheme (4) and its continuous-time counterpart when $\tau \to 0$ as

$$\begin{cases} \varphi(\mu_k) = \nabla_{\mathrm{W}_2}\phi(\mu_k) \\ \varphi(\mu_{k+1}) \circ \mathrm{T}_{k+1} = \varphi(\mu_k) - \tau\nabla_{\mathrm{W}_2}\mathcal{F}(\mu_k) \end{cases} \xrightarrow{\tau \to 0} \begin{cases} \varphi(\mu_t) = \nabla_{\mathrm{W}_2}\phi(\mu_t) \\ \frac{\mathrm{d}}{\mathrm{d}t}\varphi(\mu_t) = -\nabla_{\mathrm{W}_2}\mathcal{F}(\mu_t). \end{cases} \tag{51}$$

However, $\frac{\mathrm{d}}{\mathrm{d}t}\varphi(\mu_t) = \frac{\mathrm{d}}{\mathrm{d}t}\nabla_{\mathrm{W}_2}\phi(\mu_t) = \mathrm{H}\phi_{\mu_t}(v_t)$ where $\mathrm{H}\phi_{\mu_t} : L^2(\mu_t) \to L^2(\mu_t)$ is the Hessian operator (defined in Appendix C.2) such that $\frac{\mathrm{d}^2}{\mathrm{d}t^2}\phi(\mu_t) = \langle \mathrm{H}\phi_{\mu_t}(v_t), v_t\rangle_{L^2(\mu_t)}$ and $v_t \in L^2(\mu_t)$ is a velocity field satisfying $\partial_t\mu_t + \mathrm{div}(\mu_t v_t) = 0$. Thus, the continuity equation corresponding to the Mirror Flow is given by

$$\partial_t\mu_t - \mathrm{div}\big(\mu_t(\mathrm{H}\phi_{\mu_t})^{-1}\nabla_{\mathrm{W}_2}\mathcal{F}(\mu_t)\big) = 0. \tag{52}$$

For $\phi^V_\mu$ as Bregman potential, since $\mathrm{H}\phi^V_\mu(v) = (\nabla^2 V)v$ (see Appendix C.2), the flow is a solution of $\partial_t\mu_t - \mathrm{div}\big(\mu_t(\nabla^2 V)^{-1}\nabla_{\mathrm{W}_2}\mathcal{F}(\mu_t)\big) = 0$. For $\mathcal{F}(\mu) = \mathrm{KL}(\mu\|\nu)$ with $\nu \propto e^{-U}$, this coincides with the gradient flow of the mirror Langevin [3, 130] and with the continuity equation obtained in [104] as the limit of the JKO scheme with Bregman groundcosts. For $\phi = \mathcal{F}$, this coincides with Information Newton's flows [126]. Note also that Deb et al. [41] defined mirror flows through the scheme $\tau \to 0$ of (51), but focused on $\mathcal{F}(\mu) = \mathrm{KL}(\mu\|\nu)$ and $\phi(\mu) = \frac{1}{2}\mathrm{W}^2_2(\mu, \eta)$.

## D.3 Derivation in specific settings

In this section, we analyze several novel mirror schemes obtained through the use of different Bregman potential maps in (4), and used in various applications in Section 5. We start by discussing the scheme with an interaction energy as Bregman potential. Next, we study mirror descent with negative entropy or KL divergence as Bregman potential. For the last two, we derive closed-forms for the case where every distribution is Gaussian, which is equivalent to working on the Bures-Wasserstein space, and to use the gradient on the Bures-Wasserstein space [43]. In particular, this space is a submanifold of $\mathcal{P}_{2,\mathrm{ac}}(\mathbb{R}^d)$ and the tangent space is the space of affine maps with symmetric linear term, *i.e.* of the form $T(x) = b + S(x - m)$ with $S \in S_d(\mathbb{R})$.

**Interaction mirror scheme.** Consider as Bregman potential an interaction energy $\phi_\mu(\mathrm{T}) = \frac{1}{2}\iint W\big((T(x) - T(x'))\,\mathrm{d}\mu(x)\mathrm{d}\mu(x')$. The mirror descent scheme (4) is given by

$$\forall k \geq 0,\ (\nabla W \star \mu_{k+1}) \circ \mathrm{T}_{k+1} = \nabla W \star \mu_k - \tau\nabla_{\mathrm{W}_2}\mathcal{F}(\mu_k). \tag{53}$$

For the particular case $W(x) = \frac{1}{2}\|x\|^2_2$, the scheme can be made more explicit as $\nabla W \star \mu(x) = \int \nabla W(x - y)\,\mathrm{d}\mu(y) = \int (x - y)\,\mathrm{d}\mu(y) = x - m(\mu)$ with $m(\mu) = \int y\,\mathrm{d}\mu(y)$ the expectation, and thus (53) translates as

$$\forall k \geq 0,\ x_{k+1} - m(\mu_{k+1}) = x_k - m(\mu_k) - \tau\nabla_{\mathrm{W}_2}\mathcal{F}(\mu_k),\ x_k \sim \mu_k. \tag{54}$$

On one hand, recall from Example 2 that the Hessian of $\phi$ is given, for $\mu \in \mathcal{P}_2(\mathbb{R}^d)$, $v \in L^2(\mu)$, by

$$\forall x \in \mathbb{R}^d,\ \mathrm{H}\phi_\mu[v](x) = -\int v(y)\,\mathrm{d}\mu(y) + v(x), \tag{55}$$

since $\nabla^2 W = I_d$. On the other hand, the mirror descent scheme (54) can be written as, for all $k \geq 0$,

$$y_{k+1} = y_k - \tau \nabla_{W_2} \mathcal{F}(\mu_k)(x_k), \ y_k = x_k - m(\mu_k), \ x_k \sim \mu_k. \tag{56}$$

Passing to the limit $\tau \to 0$, we get

$$\frac{\mathrm{d}y_t}{\mathrm{d}t} = -\nabla_{W_2} \mathcal{F}(\mu_t)(x_t), \ y_t = x_t - m(\mu_t), \ x_t \sim \mu_t, \tag{57}$$

where $\frac{\mathrm{d}y_t}{\mathrm{d}t} = \frac{\mathrm{d}x_t}{\mathrm{d}t} - \frac{\mathrm{d}m(\mu_t)}{\mathrm{d}t}$. Now, by setting $v_t(x) = \frac{\mathrm{d}x_t}{\mathrm{d}t}$, by integration by part, we have

$$\frac{\mathrm{d}}{\mathrm{d}t} m(\mu_t) = \int x \, \partial_t \mu_t = -\int x \cdot \mathrm{div}(\mu_t v_t) = \int v_t(y) \, \mathrm{d}\mu_t(y). \tag{58}$$

Combining the latter equation with (55), we obtain as expected that $\frac{\mathrm{d}y_t}{\mathrm{d}t} = \mathrm{H}\phi_{\mu_t}[v_t](x)$.

**Negative entropy mirror scheme.** Consider the negative entropy $\phi(\mu) = \int \log(\rho(x)) \, \mathrm{d}\mu(x)$ where $\mathrm{d}\mu(x) = \rho(x)\mathrm{d}x$ and $\phi_\mu(\mathrm{T}) = \phi(\mathrm{T}_{\#}\mu)$. For such Bregman potential, the mirror scheme (4) can be written for all $k \geq 0$ as

$$\nabla \log \rho_{k+1} \circ \mathrm{T}_{k+1} = \nabla \log \rho_k - \tau \nabla_{W_2} \mathcal{F}(\mu_k). \tag{59}$$

In general, this scheme is not tractable. Nonetheless, supposing that $\mu_k = \mathcal{N}(m_k, \Sigma_k)$ for all $k \geq 0$, the scheme translates as

$$-\Sigma_{k+1}^{-1}(\mathrm{T}_{k+1}(x_k) - m_{k+1}) = -\Sigma_k^{-1}(x_k - m_k) - \tau \nabla_{W_2} \mathcal{F}(\mu_k), \ x_k \sim \mu_k. \tag{60}$$

- For an objective functional $\mathcal{F}(\mu) = \mathcal{H}(\mu) + \mathcal{V}(\mu)$ with $V(x) = \frac{1}{2}x^T \Sigma^{-1} x$, the scheme is

$$\begin{aligned} -\Sigma_{k+1}^{-1}(x_{k+1} - m_{k+1}) &= -\Sigma_k^{-1}(x_k - m_k) - \tau\left(-\Sigma_k^{-1}(x_k - m_k) + \Sigma^{-1}x_k\right) \\ &= -(1-\tau)\Sigma_k^{-1}(x_k - m_k) - \tau \Sigma^{-1}x_k \\ &= -\left((1-\tau)\Sigma_k^{-1} + \tau\Sigma^{-1}\right)x_k + (1-\tau)\Sigma_k^{-1}m_k. \end{aligned} \tag{61}$$

Assuming $m_k = 0$ for all $k$, we obtain the following update rule for the covariance matrices:

$$\Sigma_{k+1}^{-1} = \left((1-\tau)\Sigma_k^{-1} + \tau\Sigma^{-1}\right)^T \Sigma_k \left((1-\tau)\Sigma_k^{-1} + \tau\Sigma^{-1}\right). \tag{62}$$

We illustrate this scheme in Figure 2.

- For $\mathcal{F}(\mu) = \mathcal{H}(\mu)$, we obtain

$$-\Sigma_{k+1}^{-1}(\mathrm{T}_{k+1}(x_k) - m_{k+1}) = -(1-\tau)\Sigma_k^{-1}(x_k - m_k), \ x_k \sim \mu_k. \tag{63}$$

Assuming $m_k = 0$ for all $k$, for $\tau < 1$, we obtain the following update rule for the covariance matrices:

$$\Sigma_{k+1}^{-1} = (1-\tau)^2 \Sigma_k^{-1}, \ \text{i.e.,} \tag{64}$$

$$\Sigma_{k+1} = \frac{1}{(1-\tau)^2}\Sigma_k = \frac{1}{(1-\tau)^{2k}}\Sigma_0 \underset{\tau \to 0}{\sim} e^{2\tau k}\Sigma_0. \tag{65}$$

The continuous time analog of this scheme is thus $\mu_t : t \mapsto \mathcal{N}(0, \Sigma_t)$ with $\Sigma_t = e^{2t}\Sigma_0$ and the negative entropy decreases along this curve as

$$\begin{aligned} \mathcal{H}(\mu_t) &= \int \log\left(\rho_t(x)\right) \mathrm{d}\mu_t(x) \\ &= \int \log\left(\frac{1}{(2\pi)^{\frac{d}{2}}\sqrt{\det \Sigma_t}} e^{-\frac{1}{2}x^T \Sigma_t^{-1} x}\right) \mathrm{d}\mu_t(x) \\ &= -\frac{d}{2}\log(2\pi) - \frac{1}{2}\log\det\left(e^{2t}\Sigma_0\right) - \frac{1}{2}\mathrm{Tr}\left(\Sigma_t^{-1}\int xx^T \mathrm{d}\mu_t(x)\right) \\ &= -\frac{d}{2}\log(2\pi e) - dt - \frac{1}{2}\sum_{i=1}^{d}\log(\lambda_i), \end{aligned} \tag{66}$$

where $(\lambda_i)_i$ denote the eigenvalues of $\Sigma_0$. This is much faster than the heat flow for which the negative entropy decreases as [129, Appendix E.2]

$$\mathcal{H}(\rho_t) = -\frac{d}{2}\log(2\pi e) - \frac{1}{2}\sum_{i=1}^{d}\log(\lambda_i + 2t), \tag{67}$$

with the scheme given by [129, Example 6]

$$\forall k \geq 0, \quad \begin{cases} m_{k+1} = m_0 \\ \Sigma_{k+1} = \Sigma_k(I_d + \tau\Sigma_k^{-1})^2. \end{cases} \tag{68}$$

With our notations, the heat flow is the continuous time limit of the scheme (4) for the same objective $\mathcal{F}$ but for a quadratic Bregman potential $\phi_\mu(\mathrm{T}) = \frac{1}{2}\|\mathrm{T}\|_{L^2(\mu)}^2$ (which recovers the Wasserstein-2 geometry, hence Wasserstein-2 gradient flows).

**KL mirror scheme.** Suppose we want to optimize the KL divergence, *i.e.* a functional of the form $\mathcal{F}(\mu) = \mathcal{G}(\mu) + \mathcal{H}(\mu)$ where $\mathcal{G}(\mu) = \int U \mathrm{d}\mu$. Then, a natural choice of Bregman potential is also a functional of the form $\phi(\mu) = \Psi(\mu) + \mathcal{H}(\mu)$ with $\Psi(\mu) = \int V \mathrm{d}\mu$, with $U$ $\alpha$-convex and $\beta$-smooth relative to $V$.

In that case, we obtain the smoothness of $\mathcal{F}$ relative to $\phi$. Recall we denote $\tilde{\mathcal{F}}_\mu(\mathrm{T}) = \mathcal{F}(\mathrm{T}_{\#}\mu)$ for $\mathrm{T} \in L^2(\mu)$. Then for all $\mathrm{T}, \mathrm{S} \in L^2(\mu)$, we have $\alpha \mathrm{d}_{\tilde{\Psi}_\mu}(\mathrm{T}, \mathrm{S}) \leq \mathrm{d}_{\tilde{\mathcal{G}}_\mu}(\mathrm{T}, \mathrm{S}) \leq \beta \mathrm{d}_{\tilde{\Psi}_\mu}(\mathrm{T}, \mathrm{S})$, hence

$$\mathrm{d}_{\tilde{\mathcal{F}}_\mu}(\mathrm{T}, \mathrm{S}) = \mathrm{d}_{\tilde{\mathcal{H}}_\mu}(\mathrm{T}, \mathrm{S}) + \mathrm{d}_{\tilde{\mathcal{G}}_\mu}(\mathrm{T}, \mathrm{S}) \leq \mathrm{d}_{\tilde{\mathcal{H}}_\mu}(\mathrm{T}, \mathrm{S}) + \beta \mathrm{d}_{\tilde{\Psi}_\mu}(\mathrm{T}, \mathrm{S}) \leq \max(1, \beta)\mathrm{d}_{\phi_\mu}(\mathrm{T}, \mathrm{S}). \tag{69}$$

Similarly, $\mathrm{d}_{\tilde{\mathcal{F}}_\mu}(\mathrm{T}, \mathrm{S}) \geq \min(1, \alpha)\mathrm{d}_{\phi_\mu}(\mathrm{T}, \mathrm{S})$.

We now focus on the case where all measures are Gaussian in order to be able to compute a closed-form, *i.e.* $U(x) = \frac{1}{2}(x-m)^T\Sigma^{-1}(x-m)$, $V(x) = \frac{1}{2}x^T\Lambda^{-1}x$ and for all $k \geq 0$, $\mu_k = \mathcal{N}(m_k, \Sigma_k)$. In this case, recall that $\nabla\log\mu_k(x) = -\Sigma_k^{-1}(x-m_k)$. Then, at each step, the mirror descent scheme (4) writes for $x_k \sim \mu_k$, $k \geq 0$ as

$$\nabla V(x_{k+1}) + \nabla\log\big(\mu_{k+1}(x_{k+1})\big) = \nabla V(x_k) + \nabla\log\big(\mu_k(x_k)\big) - \tau\big(\nabla U(x_k) + \nabla\log\big(\mu_k(x_k)\big)\big)$$

$$\iff \Lambda^{-1}x_{k+1} - \Sigma_{k+1}^{-1}(x_{k+1} - m_{k+1})$$

$$= \Lambda^{-1}x_k - \Sigma_k^{-1}(x_k - m_k) - \tau\big(\Sigma^{-1}(x_k - m) - \Sigma_k^{-1}(x_k - m_k)\big)$$

$$\iff (\Lambda^{-1} - \Sigma_{k+1}^{-1})x_{k+1} + \Sigma_{k+1}^{-1}m_{k+1}$$

$$= \big(\Lambda^{-1} - (1-\tau)\Sigma_k^{-1} - \tau\Sigma^{-1}\big)x_k + (1-\tau)\Sigma_k^{-1}m_k + \tau\Sigma^{-1}m. \tag{70}$$

Thus, we get for the expectation that

$$(\Lambda^{-1} - \Sigma_{k+1}^{-1})m_{k+1} + \Sigma_{k+1}^{-1}m_{k+1} = \big(\Lambda^{-1} - (1-\tau)\Sigma_k^{-1} - \tau\Sigma^{-1}\big)m_k(1-\tau)\Sigma_k^{-1}m_k + \tau\Sigma^{-1}m$$

$$\iff \Lambda^{-1}m_{k+1} = (\Lambda^{-1} - \tau\Sigma^{-1})m_k + \tau\Sigma^{-1}m$$

$$\iff m_{k+1} = (I_d - \tau\Lambda\Sigma^{-1})m_k + \tau\Lambda\Sigma^{-1}m. \tag{71}$$

We note that the latter update on the means coincides with the forward Euler method in the forward-backward scheme, see (116) in Appendix F, which uses as Bregman potential $\phi = \Psi$. Thus, the entropy does not affect the convergence towards the mean, which can be done simply by (preconditioned) gradient descent.

For the covariance part, we get

$$(\Lambda^{-1} - \Sigma_{k+1}^{-1})^T\Sigma_{k+1}(\Lambda^{-1} - \Sigma_{k+1}^{-1})$$

$$= \big(\Lambda^{-1} - \tau\Sigma^{-1} - (1-\tau)\Sigma_k^{-1}\big)^T\Sigma_k\big(\Lambda^{-1} - \tau\Sigma^{-1} - (1-\tau)\Sigma_k^{-1}\big). \tag{72}$$

Now, supposing that all matrices commute, we get

$$\Lambda^{-2}\Sigma_{k+1} - 2\Lambda^{-1} + \Sigma_{k+1}^{-1} = (\Lambda^{-1} - \tau\Sigma^{-1})^2\Sigma_k - 2(1-\tau)\Lambda^{-1} + 2\tau(1-\tau)\Sigma^{-1}$$

$$+ (1-\tau)^2\Sigma_k^{-1} \tag{73}$$

$$\iff \Lambda^{-2}\Sigma_{k+1} + \Sigma_{k+1}^{-1} = (\Lambda^{-1} - \tau\Sigma^{-1})^2\Sigma_k + 2\tau\Lambda^{-1} + 2\tau(1-\tau)\Sigma^{-1} + (1-\tau)^2\Sigma_k^{-1}$$

$$\iff \Sigma_{k+1} + \Lambda^2\Sigma_{k+1}^{-1} = (I_d - \tau\Lambda\Sigma^{-1})^2\Sigma_k + 2\tau\Lambda + 2\tau(1-\tau)\Lambda^2\Sigma^{-1} + (1-\tau)^2\Lambda^2\Sigma_k^{-1}.$$

Denoting

$$C = (I_d - \tau\Lambda\Sigma^{-1})^2\Sigma_k + 2\tau\Lambda + 2\tau(1-\tau)\Lambda^2\Sigma^{-1} + (1-\tau)^2\Lambda^2\Sigma_k^{-1}, \tag{74}$$

the update on covariances is equivalent to

$$\Sigma_{k+1}^2 - C\Sigma_{k+1} + \Lambda^2 = 0. \tag{75}$$

Thus, $\Sigma_{k+1} = \frac{1}{2}\left(C \pm (C^2 - 4\Lambda^2)^{\frac{1}{2}}\right)$.

### D.4  Mirror scheme with non-pushforward compatible Bregman potentials

We study in this Section schemes for which the Bregman potential $\phi_\mu$ is not pushforward compatible, and thus for which we cannot apply Proposition 2 and thus Assumption 1 may not hold a priori. An example of such potential is $\phi_\mu(\mathrm{T}) = \langle \mathrm{T}, P_\mu\mathrm{T}\rangle_{L^2(\mu)}$ where $P_\mu : L^2(\mu) \to L^2(\mu)$ is a linear autoadjoint and invertible operator. Since $\nabla\phi_\mu(\mathrm{T}) = P_\mu\mathrm{T}$, taking the first order conditions, we obtain the following scheme:

$$\forall k \geq 0, \ \mathrm{T}_{k+1} = \mathrm{Id} - P_{\mu_k}^{-1}\nabla_{\mathrm{W}_2}\mathcal{F}(\mu_k). \tag{76}$$

In particular, this includes SVGD [71, 83, 84] if we pose $P_\mu^{-1}\mathrm{T} = \iota S_\mu\mathrm{T}$ with $S_\mu : L^2(\mu) \to \mathcal{H}$ defined as $S_\mu\mathrm{T} = \int k(x,\cdot)\mathrm{T}(x)\mathrm{d}\mu(x)$ which maps functions from $L^2(\mu)$ to the reproducing kernel Hilbert space $\mathcal{H}$ with kernel $k$, and with $\iota : \mathcal{H} \to L^2(\mu)$ the inclusion operator that is the adjoint of $S_\mu$ [71]. It also includes the Kalman-Wasserstein gradient descent [56] for which $P_\mu^{-1} = \int \left(x - m(\mu)\right) \otimes \left(x - m(\mu)\right)\mathrm{d}\mu(x)$ is the covariance matrix, where $m(\mu) = \int x \, \mathrm{d}\mu(x)$.

More generally, for $\phi_\mu(\mathrm{T}) = \int P_\mu(V \circ \mathrm{T})\mathrm{d}\mu$, we can recover their mirrored version, including mirrored SVGD [113, 114], *i.e.* $\mathrm{T}_{k+1} = \nabla V^* \circ \left(\nabla V - \tau P_{\mu_k}^{-1}\nabla_{\mathrm{W}_2}\mathcal{F}(\mu_k)\right)$.

**Kalman-Wasserstein.**  We focus now on a particular choice of linear operator $P_\mu$. Namely, we take $P_\mu\mathrm{T} = C(\mu)\mathrm{T}$ with $C(\mu) = \left(\int \left(x - m(\mu)\right) \otimes \left(x - m(\mu)\right)\mathrm{d}\mu(x)\right)^{-1}$ the inverse of the covariance matrix. In this case, (76) corresponds to the discretization of the Kalman-Wasserstein gradient flow [56]. We now show that it satisfies Assumption 1. First, let us compute the Bregman divergence associated to $\phi$:

$$\begin{aligned}\forall \mathrm{T}, \mathrm{S} \in L^2(\mu), \ \mathrm{d}_{\phi_\mu}(\mathrm{T}, \mathrm{S}) &= \frac{1}{2}\langle \mathrm{T}, C(\mu)\mathrm{T}\rangle_{L^2(\mu)} + \frac{1}{2}\langle \mathrm{S}, C(\mu)\mathrm{S}\rangle_{L^2(\mu)} - \langle C(\mu)\mathrm{S}, \mathrm{T}\rangle_{L^2(\mu)} \\ &= \frac{1}{2}\left(\langle \mathrm{T}, C(\mu)(\mathrm{T} - \mathrm{S})\rangle_{L^2(\mu)} + \langle \mathrm{S} - \mathrm{T}, C(\mu)\mathrm{S}\rangle_{L^2(\mu)}\right) \\ &= \frac{1}{2}\|C(\mu)^{\frac{1}{2}}(\mathrm{T} - \mathrm{S})\|_{L^2(\mu)}^2.\end{aligned} \tag{77}$$

For $\gamma = (\mathrm{T}, \mathrm{S})_{\#}\mu$, we can write

$$\mathrm{d}_{\phi_\mu}(\mathrm{T}, \mathrm{S}) = \frac{1}{2}\int \|C(\mu)^{\frac{1}{2}}(x - y)\|_2^2 \, \mathrm{d}\gamma(x, y). \tag{78}$$

Moreover, the problem $\inf_{\gamma\in\Pi(\alpha,\beta)} \int \|C(\mu)^{\frac{1}{2}}(x - y)\|_2^2 \, \mathrm{d}\gamma(x, y)$ is equivalent to

$$\inf_{\gamma\in\Pi(\alpha,\beta)} - \int x^T C(\mu)y \, \mathrm{d}\gamma(x, y), \tag{79}$$

which is a squared OT problem. Thus, it admits an OT map if $C(\mu)$ is invertible and $\mu$ or $\nu$ is absolutely continuous.

**Second point of view.**  Another point of view would be to use the linearization with the gradient corresponding to the associated generalized Wasserstein distance, which is of the form $\nabla_{\mathrm{W}}\mathcal{F}(\mu) = P_\mu^{-1}\nabla_{\mathrm{W}_2}\mathcal{F}(\mu)$ [46, 56], *i.e.* considering

$$\mathrm{T}_{k+1} = \underset{\mathrm{T}\in L^2(\mu)}{\operatorname{argmin}} \ \mathrm{d}_{\phi_\mu}(\mathrm{T}, \mathrm{Id}) + \langle\nabla_{\mathrm{W}}\mathcal{F}(\mu), \mathrm{T} - \mathrm{Id}\rangle_{L^2(\mu)}, \tag{80}$$

where we assume that $\nabla_{\mathrm{W}}\mathcal{F}(\mu) \in L^2(\mu)$. In that case, using the first order conditions,

$$\nabla\mathrm{J}(\mathrm{T}_{k+1}) = 0 \iff \nabla_{\mathrm{W}_2}\phi\left((\mathrm{T}_{k+1})_{\#}\mu_k\right) \circ \mathrm{T}_{k+1} = \nabla_{\mathrm{W}_2}\phi(\mu_k) - \tau P_{\mu_k}^{-1}\nabla_{\mathrm{W}_2}\mathcal{F}(\mu_k). \tag{81}$$

Then, for $\phi_\mu$ satisfying Assumption 1, the convergence will hold under relative smoothness and convexity assumptions similarly as for the analysis derived in Section 3.

# E   Relative convexity and smoothness

## E.1   Relative convexity and smoothness between Fenchel transforms

In this Section, we show sufficient conditions to satisfy the inequalities assumed in Proposition 5 and Proposition 6 under the additional assumption that, for all $k \geq 0$, $\tilde{\mathcal{F}}_{\mu_k}$ is superlinear, lower semicontinuous and strictly convex. In this case, we can show that $\tilde{\mathcal{F}}_{\mu_k}^*$ is Gâteaux differentiable, and thus we can use the Bregman divergence of $\tilde{\mathcal{F}}_{\mu_k}^*$.

**Lemma 18.** *Let $\phi : L^2(\mu) \to \mathbb{R}$ be a superlinear, lower semicontinuous and strictly convex function. Then, $\phi^*$ is Gâteaux differentiable.*

*Proof.* Fix $g \in L^2(\mu)$. Notice that

$$\bar{f} \in \partial \phi^*(g) \iff \phi^*(g) = \langle \bar{f}, g \rangle - \phi(\bar{f}) = \sup_{f \in L^2(\mu)} \langle f, g \rangle - \phi(f). \tag{82}$$

So to prove there is a unique element in $\partial \phi^*(g)$, we just need to show that, setting $\phi_g(f) := -\langle f, g \rangle + \phi(f)$, the problem $\inf_{f \in L^2(\mu)} \phi_g(f)$ has a unique solution. Under our assumptions, $\phi_g$ is lower semicontinuous and strictly convex. Since $\phi$ is superlinear, $\phi_g$ is coercive, *i.e.* $\lim_{\|f\| \to \infty} \phi_g(f) = +\infty$. There thus exists a solution [8, Theorem 3.3.4], which is unique by strict convexity. Hence $\partial \phi^*(g)$ is reduced to a point, which is necessarily the Gâteaux derivative of $\phi^*$ at $g$.  □

This allows us to relate the Bregman divergence of $\phi^*$ to the Bregman divergence of $\phi$.

**Lemma 19.** *Let $\phi : L^2(\mu) \to \mathbb{R}$ be a proper superlinear and strictly convex differentiable function, then for all $T, S \in L^2(\mu)$, $d_{\phi^*}\big(\nabla \phi(T), \nabla \phi(S)\big) = d_\phi(S, T)$.*

*Proof.* By [99, Corollary 3.44], we have $\phi^*\big(\nabla \phi(T)\big) = \langle T, \nabla \phi(T) \rangle_{L^2(\mu)} - \phi(T)$ for all $T \in L^2(\mu)$ since $\phi$ is convex and differentiable. By Lemma 18, $\phi^*$ is invertible and by [10, Corollary 16.24], since $\phi$ is proper, lower semicontinuous and convex, then $(\nabla \phi)^{-1} = \nabla \phi^*$.

Thus, for all $T, S \in L^2(\mu)$,

$$\begin{aligned}
d_{\phi^*}\big(\nabla \phi(T), \nabla \phi(S)\big) &= \phi^*\big(\nabla \phi(T)\big) - \phi^*\big(\nabla \phi(S)\big) - \langle \nabla \phi^*\big(\nabla \phi(S)\big), \nabla \phi(T) - \nabla \phi(S) \rangle_{L^2(\mu)} \\
&= \phi^*\big(\nabla \phi(T)\big) - \phi^*\big(\nabla \phi(S)\big) - \langle S, \nabla \phi(T) - \nabla \phi(S) \rangle_{L^2(\mu)} \\
&= \langle \nabla \phi(T), T \rangle_{L^2(\mu)} - \phi(T) - \langle \nabla \phi(S), S \rangle_{L^2(\mu)} + \phi(S) \\
&\quad - \langle S, \nabla \phi(T) - \nabla \phi(S) \rangle_{L^2(\mu)} \\
&= \phi(S) - \phi(T) - \langle \nabla \phi(T), S - T \rangle_{L^2(\mu)} \\
&= d_\phi(S, T). \hspace{4cm} \square
\end{aligned} \tag{83}$$

Finally, we can relate the relative convexity of $\phi$ relative to $\psi^*$ by using an inequality between the Bregman divergences of $\phi$ and $\psi$. In particular, we recover the assumptions of Propositions 5 and 6 for $\phi_{\mu_k}^{h^*}$ that is $\beta$-smooth and $\alpha$-convex relative to $\tilde{\mathcal{F}}_{\mu_k}^*$.

**Proposition 20.** *Let $\phi, \psi : L^2(\mu) \to \mathbb{R}$ proper, superlinear, strictly convex and differentiable. $\phi$ is $\beta$-smooth (resp. $\alpha$-convex) relative to $\psi^*$ if and only if $\forall T, S \in L^2(\mu)$, $d_\phi\big(\nabla \psi(T), \nabla \psi(S)\big) \leq \beta d_\psi(S, T)$ (resp. $d_\phi\big(\nabla \psi(T), \nabla \psi(S)\big) \geq \alpha d_\psi(S, T)$).*

*Proof of Proposition 20.* First, suppose that $\phi$ is $\beta$-smooth relative to $\psi^*$. Then, by definition,

$$\forall T, S \in L^2(\mu), \ d_\phi(T, S) \leq \beta d_{\psi^*}(T, S). \tag{84}$$

In particular,

$$d_\phi\big(\nabla \psi(T), \nabla \psi(S)\big) \leq \beta d_{\psi^*}\big(\nabla \psi(T), \nabla \psi(S)\big) = \beta d_\psi(S, T), \tag{85}$$

using Lemma 19.

On the other hand, suppose for all $T, S \in L^2(\mu)$, $d_\phi(\nabla\psi(T), \nabla\psi(S)) \le \beta d_\psi(S, T)$. Then, by first using Lemma 19 and then the supposed inequality, we have for all $T, S \in L^2(\mu)$,

$$\beta d_{\psi^*}(\nabla\psi(T), \nabla\psi(S)) = \beta d_\psi(S, T) \ge d_\phi(\nabla\psi(T), \nabla\psi(S)). \tag{86}$$

Likewise, we can show that $\phi$ is $\alpha$-convex relative to $\psi$ if and only if $d_\phi(\nabla\psi(T), \nabla\psi(S)) \ge \alpha d_\psi(S, T)$ for all $T, S \in L^2(\mu)$. $\qquad\square$

**Links with the conditions of Proposition 5 and Proposition 6.** Proposition 20 allows to translate the inequality hypothesis of Proposition 5 and Proposition 6. Assume that for all $k$, $\tilde{\mathcal{F}}_{\mu_k}$ is strictly convex, differentiable and superlinear. We first note that it implies that $\mathcal{F}_{\mu_k}$ is convex along $t \mapsto \big((1-t)T_{k+1} + t\mathrm{Id}\big)_{\#}\mu_k$. Moreover, by Lemma 18, $\nabla\tilde{\mathcal{F}}^*_{\mu_k}$ is differentiable.

Note that this assumption is satisfied, *e.g.* by $\phi_\mu(T) = \int V \circ T \, d\mu$ for $V$ $\eta$-strongly convex and differentiable. Indeed, in this case, $\phi_\mu$ is also $\eta$-strongly convex, and satisfies for all $T, S \in L^2(\mu)$,

$$d_{\phi_\mu}(T, S) = \phi_\mu(T) - \phi_\mu(S) - \langle\nabla\phi_\mu(S), T - S\rangle_{L^2(\mu)} \ge \frac{\eta}{2}\|T - S\|^2_{L^2(\mu)}$$
$$\iff \phi_\mu(T) \ge \phi_\mu(S) + \langle\nabla\phi_\mu(S), T - S\rangle_{L^2(\mu)} + \frac{\eta}{2}\|T - S\|^2_{L^2(\mu)}. \tag{87}$$

For $S = 0$, and dividing by $\|T\|_{L^2(\mu)}$ the right term diverges to $+\infty$ when $\|T\|_{L^2(\mu)} \to +\infty$, and thus $\lim_{\|T\|_{L^2(\mu)} \to \infty} \phi_\mu(T)/\|T\|_{L^2(\mu)} = +\infty$, and $\phi_\mu$ is superlinear.

This assumption is also satisfied for interaction energies $\phi^W_\mu(T) = \iint W\big(T(x) - T(y)\big) \, d\mu(x)d\mu(y)$ with $W$ $\eta$-strongly convex, even and differentiable. Indeed, by strong convexity of $W$ in 0, we have for all $x, y \in \mathbb{R}^d$,

$$W\big(T(x) - T(y)\big) - W(0) - \langle\nabla W(0), T(x) - T(y)\rangle \ge \frac{\eta}{2}\|T(x) - T(y)\|^2_2$$
$$\ge \frac{\eta}{2}\inf_{z\in\mathbb{R}^d}\|T(x) - z\|^2_2. \tag{88}$$

Integrating *w.r.t.* $\mu \otimes \mu$, we get

$$\phi^W_\mu(T) - W(0) \ge \frac{\eta}{2}\inf_{z\in\mathbb{R}^d}\int\|T(x) - z\|^2_2 \, d\mu(x), \tag{89}$$

and dividing by $\|T\|_{L^2(\mu)}$, we get that $\phi^W_\mu$ is superlinear.

For a curve $t \mapsto \mu_t$, we define $\mathcal{F}^*_\mu$ on $\mu_t$ as $\mathcal{F}^*_\mu(\mu_t) := \tilde{\mathcal{F}}^*_\mu(T_t)$ with $\tilde{\mathcal{F}}^*_\mu$ the convex conjugate of $\tilde{\mathcal{F}}_\mu$ in the $L^2(\mu)$ sense. Then, we can apply Proposition 20, and we obtain that the inequality hypothesis of Proposition 5 is implied by the $\beta$-smoothness of $\phi^{h^*}$ relative to $\mathcal{F}^*_{\mu_k}$ along $t \mapsto \big((1-t)\nabla_{W_2}\mathcal{F}(\mu_k) + t\nabla_{W_2}\mathcal{F}(\mu_{k+1}) \circ T_{k+1}\big)_{\#}\mu_k$ since

$$d_{\phi^{h^*}_{\mu_k}}\big(\nabla_{W_2}\mathcal{F}(\mu_{k+1}) \circ T_{k+1}, \nabla_{W_2}\mathcal{F}(\mu_k)\big) \le \beta d_{\tilde{\mathcal{F}}_{\mu_k}}(\mathrm{Id}, T_{k+1})$$
$$= \beta d_{\tilde{\mathcal{F}}^*_{\mu_k}}\big(\nabla_{W_2}\mathcal{F}(\mu_{k+1}) \circ T_{k+1}, \nabla_{W_2}\mathcal{F}(\mu_k)\big), \tag{90}$$

where we used Proposition 1 to compute the gradient $\nabla\tilde{\mathcal{F}}_{\mu_k}(T_{k+1}) = \nabla_{W_2}\mathcal{F}(\mu_{k+1}) \circ T_{k+1}$.

Similarly, the condition of Proposition 6

$$d_{\phi^{h^*}_{\mu_k}}\big(\nabla_{W_2}\mathcal{F}(T_{\#}\mu_k) \circ T, \nabla_{W_2}\mathcal{F}(\mu_k)\big) \ge \alpha d_{\tilde{\mathcal{F}}_{\mu_k}}(\mathrm{Id}, T)$$
$$= \alpha d_{\tilde{\mathcal{F}}^*_{\mu_k}}\big(\nabla_{W_2}\mathcal{F}(T_{\#}\mu_k) \circ T, \nabla_{W_2}\mathcal{F}(\mu_k)\big) \tag{91}$$

is implied by the $\alpha$-convexity of $\phi^{h^*}$ relative to $\mathcal{F}^*_{\mu_k}$ along $t \mapsto \big((1-t)\nabla_{W_2}\mathcal{F}(\mu_k) + t\nabla_{W_2}\mathcal{F}(T_{\#}\mu_k) \circ T\big)_{\#}\mu_k$.

These results are summarized in Proposition 7 and shown formally in Appendix H.7.

**Convergence towards the minimizer in Proposition 6.** We add an additional result justifying the convergence towards the minimizer in Proposition 6.

**Lemma 21.** *Let $(X, \tau)$ be a metrizable topological space, and $f : X \to \mathbb{R} \cup \{+\infty\}$ be strictly convex, $\tau$-lower semicontinuous and with one $\tau$-compact sublevel set. Let $x_0 \in X$ be the minimizer of $f$ and take a sequence $(x_n)_{n \in \mathbb{N}}$ such that $f(x_n) \to f(x_0)$. Then, $(x_n)_{n \in \mathbb{N}}$ $\tau$-converges to $x_0$.*

*Proof.* The existence of the minimum is given by [8, Theorem 3.2.2]. For $N$ large enough, $(x_n)_{n \geq N}$ lives in the $\tau$-compact sublevel set of $f$, since $x_0$ belongs to it and $f(x_0)$ is minimal. We can then consider a subsequence $\tau$-converging to some $x^*$. By $\tau$-lower semicontinuity, we have $f(x_0) \leq f(x^*) \leq \liminf f(x_{\sigma(n)}) = f(x_0)$, so $f(x_0) = f(x^*)$ and by strict convexity $x_0 = x^*$. Since all subsequences of $(x_n)_{n \geq N}$ converge to $x^*$ and the space is metrizable, $(x_n)_{n \in \mathbb{N}}$ $\tau$-converges to $x_0$. □

The typical case is when $X$ is a Hilbert space and $\tau$ is the weak topology. One could wish to have strong convergence under a coercivity assumption, however "In infinite dimensional spaces, the topologies which are directly related to coercivity are the weak topologies" [8, p86]. Nevertheless Gâteaux differentiability implies continuity, which paired with convexity gives weak lower semicontinuity [8, Theorem 3.3.3]. We cannot hope for convergence of the norm of $x_n$ to come for free, as the weak convergence would then imply the strong convergence.

### E.2 Relative convexity and smoothness between functionals

Let $U, V : \mathbb{R}^d \to \mathbb{R}$ be differentiable and convex functions. We recall that $V$ is $\alpha$-convex relative to $U$ if [88]

$$\forall x, y \in \mathbb{R}^d, \ \mathrm{d}_V(x, y) \geq \alpha \mathrm{d}_U(x, y). \tag{92}$$

Likewise, $V$ is $\beta$-smooth relative to $U$ if

$$\mathrm{d}_V(x, y) \leq \beta \mathrm{d}_U(x, y). \tag{93}$$

**Relative convexity and smoothness between potential energies.** By Lemma 31, for Bregman potentials of the form $\phi_\mu(\mathrm{T}) = \int V \circ \mathrm{T} \, \mathrm{d}\mu$, the Bregman divergence can be written as

$$\forall \mathrm{T}, \mathrm{S} \in L^2(\mu), \ \mathrm{d}_{\phi_\mu}(\mathrm{T}, \mathrm{S}) = \int \mathrm{d}_V\big(\mathrm{T}(x), \mathrm{S}(x)\big) \, \mathrm{d}\mu(x). \tag{94}$$

Thus, leveraging this result, we can show that relative convexity and smoothness of $\phi_\mu^V$ relative to $\phi_\mu^U$ is inherited by the relative convexity and smoothness of $V$ relative to $U$.

**Proposition 22.** *Let $\mu \in \mathcal{P}_2(\mathbb{R}^d)$, $\phi_\mu(\mathrm{T}) = \int V \circ \mathrm{T} \, \mathrm{d}\mu$ and $\psi_\mu(\mathrm{T}) = \int U \circ \mathrm{T} \, \mathrm{d}\mu$ where $V, U : \mathbb{R}^d \to \mathbb{R}$ are $C^1$. If $V$ is $\alpha$-convex (resp. $\beta$-smooth) relative to $U : \mathbb{R}^d \to \mathbb{R}$, then $\phi_\mu$ is $\alpha$-convex (resp $\beta$-smooth) relative to $\psi_\mu$.*

*Proof.* First, observe (Lemma 31) that

$$\forall \mu \in \mathcal{P}_2(\mathbb{R}^d), \mathrm{T}, \mathrm{S} \in L^2(\mu), \ \mathrm{d}_{\phi_\mu}(\mathrm{T}, \mathrm{S}) = \int \mathrm{d}_V\big(\mathrm{T}(x), \mathrm{S}(x)\big) \, \mathrm{d}\mu(x). \tag{95}$$

Let $\mu \in \mathcal{P}_2(\mathbb{R}^d)$, $\mathrm{T}, \mathrm{S} \in L^2(\mu)$. If $V$ is $\alpha$-convex relatively to $U$, we have for all $x, y \in \mathbb{R}^d$,

$$\mathrm{d}_V\big(\mathrm{T}(x), \mathrm{S}(y)\big) \geq \alpha \mathrm{d}_U\big(\mathrm{T}(x), \mathrm{S}(y)\big), \tag{96}$$

and hence by integrating on both sides with respect to $\mu$,

$$\mathrm{d}_{\phi_\mu}(\mathrm{T}, \mathrm{S}) \geq \alpha \mathrm{d}_{\psi_\mu}(\mathrm{T}, \mathrm{S}). \tag{97}$$

Likewise, we have the result for the $\beta$-smoothness. □

**Relative convexity and smoothness between interaction energies.** Similarly, by Lemma 32, for Bregman potentials obtained through interaction energies, *i.e.* $\phi_\mu(\mathrm{T}) = \frac{1}{2} \iint W(\mathrm{T}(x) - \mathrm{T}(x')) \, \mathrm{d}\mu(x)\mathrm{d}\mu(x')$, then

$$\forall \mathrm{T}, \mathrm{S} \in L^2(\mu), \; \mathrm{d}_{\phi_\mu}(\mathrm{T}, \mathrm{S}) = \frac{1}{2} \iint \mathrm{d}_W(\mathrm{T}(x) - \mathrm{T}(x'), \mathrm{S}(x) - \mathrm{S}(x')) \, \mathrm{d}\mu(x)\mathrm{d}\mu(x'). \quad (98)$$

It also allows to inherit the relative convexity and smoothness results from $\mathbb{R}^d$.

**Proposition 23.** *Let $\mu \in \mathcal{P}_2(\mathbb{R}^d)$, $W, K : \mathbb{R}^d \to \mathbb{R}$ be symmetric, $C^1$ and convex. Let $\phi_\mu(\mathrm{T}) = \frac{1}{2} \iint W(\mathrm{T}(x) - \mathrm{T}(x')) \, \mathrm{d}\mu(x)\mathrm{d}\mu(x')$ and $\psi_\mu(\mathrm{T}) = \frac{1}{2} \iint K(\mathrm{T}(x) - \mathrm{T}(x')) \, \mathrm{d}\mu(x)\mathrm{d}\mu(x')$. If $W$ is $\alpha$-convex relative to $K$, then $\phi_\mu$ is $\alpha$-convex relatively to $\psi_\mu$. Likewise, if $W$ is $\beta$-smooth relatively to $K$, then $\phi_\mu$ is $\beta$-smooth relatively to $\psi_\mu$.*

*Proof.* We use first Lemma 32 and then that $W$ is $\alpha$-convex relatively to $K$:

$$\begin{aligned} \mathrm{d}_{\phi_\mu}(\mathrm{T}, \mathrm{S}) &= \frac{1}{2} \iint \mathrm{d}_W(\mathrm{T}(x) - \mathrm{T}(x'), \mathrm{S}(x) - \mathrm{S}(x')) \, \mathrm{d}\mu(x)\mathrm{d}\mu(x') \\ &\geq \frac{\alpha}{2} \iint \mathrm{d}_K(\mathrm{T}(x) - \mathrm{T}(x'), \mathrm{S}(x) - \mathrm{S}(x')) \, \mathrm{d}\mu(x)\mathrm{d}\mu(x') \\ &= \alpha \mathrm{d}_{\psi_\mu}(\mathrm{T}, \mathrm{S}). \end{aligned} \quad (99)$$

Likewise, we have the result for the $\beta$-smoothness. $\qquad\square$

Thus, in situations where the objective functional and the Bregman potential are of the same type and either potential energies or interaction energies, we only need to show the convexity and smoothness of the underlying potentials or interaction kernels. For instance, let $V : \mathbb{R}^d \to \mathbb{R}$ be a twice-differentiable convex function, such that $\|\nabla^2 V\|_{\mathrm{op}} \leq p_r(\|x\|_2)$ with $p_r$ a polynomial function of degree $r$ and $\|\cdot\|_{\mathrm{op}}$ the operator norm. Then, by [88, Proposition 2.1], $V$ is $\beta$-smooth relative to $h$ where for all $x \in \mathbb{R}^d$, $h(x) = \frac{1}{r+2}\|x\|_2^{r+2} + \frac{1}{2}\|x\|_2^2$.

**Relative convexity and smoothness between functionals of different types.** When the functionals do not belong to the same type, comparing directly the Bregman divergences is less straightforward in general. In that case, one might instead leverage the equivalence relations given by Proposition 13 and Proposition 14, and show that $\beta\mathcal{G} - \mathcal{F}$ or $\mathcal{F} - \alpha\mathcal{G}$ is convex in order to show respectively the $\beta$-smoothness and $\alpha$-convexity of $\mathcal{F}$ relative to $\mathcal{G}$. For instance, we can use the characterization through Hessians, and thus we would aim at showing

$$\frac{\mathrm{d}^2}{\mathrm{d}t^2}\mathcal{F}(\mu_t) \leq \beta\frac{\mathrm{d}^2}{\mathrm{d}t^2}\mathcal{G}(\mu_t), \quad \frac{\mathrm{d}^2}{\mathrm{d}t^2}\mathcal{F}(\mu_t) \geq \alpha\frac{\mathrm{d}^2}{\mathrm{d}t^2}\mathcal{G}(\mu_t), \quad (100)$$

along the right curve $t \mapsto \mu_t$.

For instance, consider an objective functional $\mathcal{F}(\mu) = \frac{1}{2} \iint W(x - y) \, \mathrm{d}\mu(x)\mathrm{d}\mu(x')$ and another functional $\mathcal{G}(\mu) = \int V \mathrm{d}\mu$. Then, by Example 1 and Example 2, we have, for $\mu_t = (\mathrm{T}_t)_{\#}\mu$ and $\mathrm{T}_t = \mathrm{S} + tv$,

$$\frac{\mathrm{d}^2}{\mathrm{d}t^2}\mathcal{G}(\mu_t) = \int \langle \nabla^2 V(\mathrm{T}_t(x))v(x), v(x) \rangle \, \mathrm{d}\mu(x), \quad (101)$$

and

$$\frac{\mathrm{d}^2}{\mathrm{d}t^2}\mathcal{F}(\mu_t) = \iint \langle \nabla^2 W(\mathrm{T}_t(x) - \mathrm{T}_t(y))(v(x) - v(y)), v(x) \rangle \, \mathrm{d}\mu(x)\mathrm{d}\mu(y). \quad (102)$$

To show the conditions of Proposition 3, we need to take $S = Id$ and $v = T_{k+1} - Id$, and to verify for $t = s \in [0, 1]$ the inequality, *i.e.*

$$\frac{d^2}{dt^2}\mathcal{F}(\mu_t)\Big|_{t=s} \leq \beta \frac{d^2}{dt^2}\mathcal{G}(\mu_t)\Big|_{t=s}$$

$$\Longleftrightarrow$$

$$\iint \langle \nabla^2 W(T_s(x) - T_s(y))(v(x) - v(y)), v(x) \rangle \, d\mu_k(x) d\mu_k(y)$$

$$\leq \beta \int \langle \nabla^2 V(T_s(x))v(x), v(x) \rangle \, d\mu_k(x) \qquad (103)$$

$$\Longleftrightarrow$$

$$\int \Big\langle v(x), \int \Big( (\nabla^2 W(T_s(x) - T_s(y)) - \beta \nabla^2 V(T_s(x)))v(x)$$

$$- \nabla^2 W(T_s(x) - T_s(y))v(y) \Big) d\,\mu_k(y) \Big\rangle d\mu_k(x) \leq 0.$$

For example, choosing $W(x) = \frac{1}{2}\|x\|_2^2$, then $\nabla^2 W = I_d$ and $\mathcal{F}$ is $\beta$-smooth relative to $\mathcal{G}$ as long as $\nabla^2 V \circ T_s \succeq \frac{1}{\beta}I_d$ for any $s \in [0, 1]$.

## F Bregman proximal gradient scheme

In this section, we are interested into minimizing a functional $\mathcal{F}$ of the form $\mathcal{F}(\mu) = \mathcal{G}(\mu) + \mathcal{H}(\mu)$ where $\mathcal{G}$ is smooth relative to some function $\phi$ and $\mathcal{H}$ is convex on $L^2(\mu)$. Different strategies can be used to tackle this problem. For instance, Jiang et al. [65] restrict the space to particular directions along which $\mathcal{H}$ is smooth while Diao et al. [43], Salim et al. [109] use Proximal Gradient algorithms. We focus here on the latter and generalize the Bregman Proximal Gradient algorithm [11], also known as the Forward-Backward scheme. It consists of alternating a forward step on $\mathcal{G}$ and then a backward step on $\mathcal{H}$, *i.e.* for $k \geq 0$,

$$\begin{cases} S_{k+1} = \text{argmin}_{S \in L^2(\mu_k)} \, d_{\phi_{\mu_k}}(S, Id) + \tau\langle \nabla_{W_2}\mathcal{G}(\mu_k), S - Id \rangle_{L^2(\mu_k)}, & \nu_{k+1} = (S_{k+1})_{\#}\mu_k \\ T_{k+1} = \text{argmin}_{T \in L^2(\nu_{k+1})} \, d_{\phi_{\nu_{k+1}}}(T, Id) + \tau\mathcal{H}(T_{\#}\nu_{k+1}), & \mu_{k+1} = (T_{k+1})_{\#}\nu_{k+1}. \end{cases}$$
$$(104)$$

The first step of our analysis is to show that this scheme is equivalent to

$$\begin{cases} \tilde{T}_{k+1} = \text{argmin}_{T \in L^2(\mu_k)} \, d_{\phi_{\mu_k}}(T, Id) + \tau\big(\langle \nabla_{W_2}\mathcal{G}(\mu_k), T - Id \rangle_{L^2(\mu_k)} + \mathcal{H}(T_{\#}\mu_k)\big) \\ \mu_{k+1} = (\tilde{T}_{k+1})_{\#}\mu_k. \end{cases}$$
$$(105)$$

This is true under the condition that $\mu_k \in \mathcal{P}_{2,\text{ac}}(\mathbb{R}^d)$ implies that $\nu_{k+1} \in \mathcal{P}_{2,\text{ac}}(\mathbb{R}^d)$.

**Proposition 24.** *Let $\phi_\mu$ be pushforward compatible, $\mu_0 \in \mathcal{P}_{2,\text{ac}}(\mathbb{R}^d)$ and assume that if $\mu_k \in \mathcal{P}_{2,\text{ac}}(\mathbb{R}^d)$ then $\nu_{k+1} \in \mathcal{P}_{2,\text{ac}}(\mathbb{R}^d)$. Then the schemes (104) and (105) are equivalent.*

*Proof.* See Appendix H.11. □

We are now ready to state the convergence results for the proximal gradient scheme.

**Proposition 25.** *Let $\mu_0 \in \mathcal{P}_{2,\text{ac}}(\mathbb{R}^d)$, $\tau \leq \frac{1}{\beta}$ and $\mathcal{F}(\mu) = \mathcal{G}(\mu) + \mathcal{H}(\mu)$. Consider the iterates of the Bregman proximal gradient scheme (104), equivalently (105). Let $k \geq 0$. Assume $\tilde{\mathcal{H}}_{\mu_k}$ is convex on $L^2(\mu_k)$ and $\mathcal{G}$ $\beta$-smooth relative to $\phi$ along $t \mapsto ((1-t)Id + t\tilde{T}_{k+1})_{\#}\mu_k$. Then, for all $T \in L^2(\mu_k)$,*

$$\mathcal{F}(\mu_{k+1}) \leq \mathcal{H}(T_{\#}\mu_k) + \mathcal{G}(\mu_k) + \langle \nabla_{W_2}\mathcal{G}(\mu_k), T - Id \rangle_{L^2(\mu_k)} + \frac{1}{\tau}d_{\phi_{\mu_k}}(T, Id) - \frac{1}{\tau}d_{\phi_{\mu_k}}(T, \tilde{T}_{k+1}).$$
$$(106)$$

*Moreover, for $T = Id$,*

$$\mathcal{F}(\mu_{k+1}) \leq \mathcal{F}(\mu_k) - \frac{1}{\tau}d_{\phi_{\mu_k}}(Id, \tilde{T}_{k+1}), \qquad (107)$$

*i.e., the scheme decreases the objective at each iteration. Additionally, let $\alpha \geq 0$, $\nu \in \mathcal{P}_2(\mathbb{R}^d)$ and suppose that $\phi_\mu$ satisfies Assumption 1. If $\mathcal{G}$ is $\alpha$-convex relative to $\phi$ along $t \mapsto \big((1-t)\mathrm{Id} + t\mathrm{T}_{\phi_{\mu_k}}^{\mu_k,\nu}\big)_{\#}\mu_k$, then for all $k \geq 1$,*

$$\mathcal{F}(\mu_k) - \mathcal{F}(\nu) \leq \frac{\alpha}{(1-\tau\alpha)^{-k} - 1}\mathrm{W}_\phi(\nu,\mu_0) \leq \frac{1-\alpha\tau}{k\tau}\mathrm{W}_\phi(\nu,\mu_0). \tag{108}$$

*Proof.* See Appendix H.12. $\qquad\qquad\square$

We verify now that Proposition 24 can be applied for mirror schemes of interest. Salim et al. [109, Lemma 2] showed that it holds for the Wasserstein proximal gradient when using some potential energies, more precisely with $\phi(\mu) = \int \frac{1}{2}\|\cdot\|_2^2\,\mathrm{d}\mu$ and $\mathcal{G}(\mu) = \int U\,\mathrm{d}\mu$ with $U$ (strictly) convex. We extend their result for $\mathcal{G}(\mu) = \int U\,\mathrm{d}\mu$ and $\phi(\mu) = \int V\,\mathrm{d}\mu$ for $V$ strictly convex and $U$ $\beta$-smooth relative to $V$.

**Lemma 26.** *Let $\mu \in \mathcal{P}_{2,\mathrm{ac}}(\mathbb{R}^d)$, $\mathcal{G}(\mu) = \int U\,\mathrm{d}\mu$, $\phi_\mu(\mathrm{T}) = \int V \circ \mathrm{T}\,\mathrm{d}\mu$ with $V$ strongly convex and $U$ $\beta$-smooth relative to $V$, and $\mathrm{T} = \nabla V^* \circ (\nabla V - \tau\nabla U)$. Assume $\tau < \frac{1}{\beta}$, then $\mathrm{T}_{\#}\mu \in \mathcal{P}_{2,\mathrm{ac}}(\mathbb{R}^d)$.*

*Sketch of the proof.* The proof of the lemma is inspired from [109, Lemma 2]. We apply [5, Lemma 5.5.3], which requires to show that $\mathrm{T}$ is injective almost everywhere and that $|\det \nabla\mathrm{T}| > 0$ almost everywhere. See Appendix H.13 for the full proof. $\qquad\square$

To apply Proposition 25, we need $\mathcal{H}$ to be convex along some curve. We discuss here the convexity of the negative entropy along acceleration free curves. Let $\mu \in \mathcal{P}_{2,\mathrm{ac}}(\mathbb{R}^d)$, and denote $\rho$ its density *w.r.t* the Lebesgue measure. For $\mathcal{H}(\mu) = \int f\big(\rho(x)\big)\,\mathrm{d}x$ where $f : \mathbb{R} \to \mathbb{R}$ is $C^1$ and satisfies $f(0) = 0$, $\lim_{x\to 0} xf'(x) = 0$ and $x \mapsto f(x^{-d})x^d$ is convex and non-increasing on $\mathbb{R}_+$, then by [117, Theorem 4.2], $\mathcal{H}$ is convex along curves $\mu_t = \big((1-t)\mathrm{S} + t\mathrm{T}\big)_{\#}\mu$ obtained with $\mathrm{S}$ and $\mathrm{T}$ with positive definite Jacobians. This is the case *e.g.* for $f(x) = x\log x$, for which $\mathcal{H}$ corresponds to the negative entropy.

By Remark 3, to be able to apply the three-point inequality (that is necessary to obtain the descent lemma), we actually only need $\mathcal{H}$ to be convex along $\big((1-t)\tilde{\mathrm{T}}_{k+1} + t\mathrm{Id}\big)_{\#}\mu_k$ and along $\big((1-t)\tilde{\mathrm{T}}_{k+1} + t\mathrm{T}_{\phi_{\mu_k}}^{\mu_k,\nu}\big)_{\#}\mu_k$ for the convergence.

**Gaussian target.** In what follows, we focus on $\mathcal{G}(\mu) = \int U\,\mathrm{d}\mu$ with $U(x) = \frac{1}{2}(x-m)^T\Sigma^{-1}(x-m)$ for $\Sigma \in S_d^{++}(\mathbb{R})$, $\mathcal{H}$ the negative entropy and with a Bregman potential of the form $\phi(\mu) = \int V\,\mathrm{d}\mu$ with $V(x) = \frac{1}{2}x^T\Lambda^{-1}x$. Moreover, we suppose $\mu_0 = \mathcal{N}(m_0,\Sigma_0)$. In this situation, each distribution $\mu_k$ is also Gaussian, as the forward and backward steps are affine operations.

Assuming the covariances matrices are full rank, $\tilde{\mathrm{T}}_{k+1}$ is affine and its gradient is invertible. Moreover, by Proposition 15, $\mathrm{T}_{\phi_{\mu_k}}^{\mu_k,\nu} = \nabla u \circ \nabla_{\mathrm{W}_2}\phi(\mu_k)$ for $\nabla u$ an OT map between $\nabla_{\mathrm{W}_2}\phi(\mu_k)_{\#}\mu_k$ and $\nu$. Since each distribution is Gaussian, and $\nabla_{\mathrm{W}_2}\phi(\mu_k)(x) = \Lambda^{-1}x$ is affine, it has a positive definite Jacobian. Thus, using [117, Theorem 4.2], we can conclude that we can apply Proposition 25.

**Closed-form for Gaussians.** Let $\mathcal{G}(\mu) = \int U\,\mathrm{d}\mu$ with $U(x) = \frac{1}{2}(x-m)^T\Sigma^{-1}(x-m)$, $\Sigma \in S_d^{++}(\mathbb{R})$, $m \in \mathbb{R}^d$, and $\mathcal{H}(\mu) = \int \log\big(\rho(x)\big)\,\mathrm{d}\mu(x)$ for $\mathrm{d}\mu = \rho(x)\mathrm{d}x$. For the Bregman potential, we will choose $\phi(\mu) = \int V\,\mathrm{d}\mu$ for $V(x) = \frac{1}{2}\langle x,\Lambda^{-1}x\rangle$. Recall that the forward step reads as

$$\mathrm{S}_{k+1} = \nabla V^* \circ \big(\nabla V - \tau\nabla_{\mathrm{W}_2}\mathcal{G}(\mu_k)\big), \quad \nu_{k+1} = (\mathrm{S}_{k+1})_{\#}\mu_k. \tag{109}$$

Since $\nabla V(x) = \Lambda^{-1}x$, and $\mu_k = \mathcal{N}(m_k,\Sigma_k)$, we obtain for $x_k \sim \mu_k$,

$$\mathrm{S}_{k+1}(x_k) = \Lambda\big(\Lambda^{-1}x_k - \tau\Sigma^{-1}(x_k-m)\big) = x_k - \tau\Lambda\Sigma^{-1}(x_k-m). \tag{110}$$

Thus, the output of the forward step is still a Gaussian of the form $\nu_{k+1} = \mathcal{N}(m_{k+\frac{1}{2}},\Sigma_{k+\frac{1}{2}})$ with

$$\begin{cases} m_{k+\frac{1}{2}} = (I_d - \tau\Lambda\Sigma^{-1})m_k + \tau\Lambda\Sigma^{-1}m \\ \Sigma_{k+\frac{1}{2}} = (I_d - \tau\Lambda\Sigma^{-1})^T\Sigma_k(I_d - \tau\Lambda\Sigma^{-1}). \end{cases} \tag{111}$$

Since $\nabla V$ is linear, the output of the backward step stays Gaussian. Moreover, the first order conditions give

$$\nabla V \circ \mathrm{T}_{k+1} + \tau \nabla \log(\rho_{k+1} \circ \mathrm{T}_{k+1}) = \nabla V$$
$$\Longleftrightarrow \forall x, \; \Lambda^{-1} x = \Lambda^{-1} \mathrm{T}_{k+1}(x) - \tau \Sigma_{k+1}^{-1}(\mathrm{T}_{k+1}(x) - m_{k+1})$$
$$\Longleftrightarrow \forall x, \; x = \mathrm{T}_{k+1}(x) - \tau \Lambda \Sigma_{k+1}^{-1}(\mathrm{T}_{k+1}(x) - m_{k+1}). \quad (112)$$

Thus, the output is a Gaussian $\mathcal{N}(m_{k+1}, \Sigma_{k+1})$ with $(m_{k+1}, \Sigma_{k+1})$ satisfying

$$\begin{cases} m_{k+1} = m_{k+\frac{1}{2}} \\ \Sigma_{k+\frac{1}{2}} = (I_d - \tau \Lambda \Sigma_{k+1}^{-1})^T \Sigma_{k+1} (I_d - \tau \Lambda \Sigma_{k+1}^{-1}). \end{cases} \quad (113)$$

Moreover, if $\Lambda$ and $\Sigma_{k+1}$ commute, this is equivalent to

$$\Sigma_{k+1}^2 - (2\tau \Lambda + \Sigma_{k+\frac{1}{2}})\Sigma_{k+1} + \tau^2 \Lambda^2 = 0, \quad (114)$$

which solution is given by

$$\Sigma_{k+1} = \frac{1}{2}\left(\Sigma_{k+\frac{1}{2}} + 2\tau\Lambda + (\Sigma_{k+\frac{1}{2}}(4\tau\Lambda + \Sigma_{k+\frac{1}{2}}))^{\frac{1}{2}}\right). \quad (115)$$

To sum up, the update is

$$\begin{cases} \nu_{k+1} = \mathcal{N}\left((I_d - \tau\Lambda\Sigma^{-1})m_k + \tau\Lambda\Sigma^{-1}m, (I_d - \tau\Lambda\Sigma^{-1})^T\Sigma_k(I_d - \tau\Lambda\Sigma^{-1})\right) \\ \mu_{k+1} = \mathcal{N}\left(m_{k+\frac{1}{2}}, \frac{1}{2}(\Sigma_{k+\frac{1}{2}} + 2\tau\Lambda + (\Sigma_{k+\frac{1}{2}}(4\tau\Lambda + \Sigma_{k+\frac{1}{2}}))^{\frac{1}{2}})\right). \end{cases} \quad (116)$$

For $\Lambda = \Sigma$, we call it the ideally preconditioned Forward-Backward scheme (PFB).

## G  Additional details on experiments

### G.1  Implementing the schemes

In this subsection, we sum up how to implement the different schemes in practice, given a finite number of particles. In all cases, we first sample $x_1^{(0)}, \dots, x_n^{(0)} \sim \mu_0$, then we apply the scheme to $\hat{\mu}_n^{(k)} = \frac{1}{n}\sum_{i=1}^n \delta_{x_i^{(k)}}$.

**Mirror descent.**  In general, for $\phi$ pushforward compatible, one needs to solve at each iteration $k \geq 0$,

$$\nabla_{\mathrm{W}_2}\phi(\mu_{k+1}) \circ \mathrm{T}_{k+1} = \nabla_{\mathrm{W}_2}\phi(\mu_k) - \tau \nabla_{\mathrm{W}_2}\mathcal{F}(\mu_k). \quad (117)$$

If $\phi(\mu) = \int V \, \mathrm{d}\mu$ with $\nabla V$ having an analytical inverse, the scheme can be implemented as

$$\forall k \geq 0, \; \forall i \in \{1, \dots, n\}, \; x_i^{(k+1)} = \nabla V^*\left(\nabla V(x_i^{(k)}) - \tau\nabla_{\mathrm{W}_2}\mathcal{F}(\hat{\mu}_n^{(k)})(x_i^{(k)})\right). \quad (118)$$

Except for this case, one cannot in general invert $\nabla_{\mathrm{W}_2}\phi(\mu_{k+1}) \circ \mathrm{T}_{k+1}$ directly. A practical workaround is to solve an implicit problem, see *e.g.* [133]. Here, we use the Newton-Raphson algorithm. Suppose we have $\mu_k = \frac{1}{n}\sum_{i=1}^n \delta_{x_i^{(k)}}$ and we are looking for $\mu_{k+1} = \frac{1}{n}\sum_{i=1}^n \delta_{x_i}$. Then, the scheme is equivalent to

$$\forall j \in \{1, \dots, n\}, \; G_j(x_1, \dots, x_n) = 0, \quad (119)$$

for

$$G_j(x_1, \dots, x_n) = \nabla_{\mathrm{W}_2}\phi\left(\frac{1}{n}\sum_{i=1}^n \delta_{x_i}\right)(x_j) - \nabla_{\mathrm{W}_2}\phi(\mu_k)(x_j^{(k)}) + \tau\nabla_{\mathrm{W}_2}\mathcal{F}(\mu_k)(x_j^{(k)}). \quad (120)$$

Write $\mathcal{G}(x_1, \dots, x_n) = \left(G_1(x_1, \dots, x_n), \dots, G_n(x_1, \dots, x_n)\right)$. Then, at each step $k$, we perform the following Newton iterations, starting from $(x_1^{(k)}, \dots, x_n^{(k)})$:

$$(x_1^{(k_{\ell+1})}, \dots, x_n^{(k_{\ell+1})}) = (x_1^{(k_\ell)}, \dots, x_n^{(k_\ell)}) - \gamma\left(J_{\mathcal{G}}(x_1^{(k_\ell)}, \dots, x_n^{(k_\ell)})\right)^{-1}\mathcal{G}(x_1^{(k_\ell)}, \dots, x_n^{(k_\ell)}). \quad (121)$$

The Jacobian is of size $nd \times nd$, which does not scale well with the dimension and the number of samples. We can reduce the complexity of the algorithm by relying on inverse Hessian vector products, see *e.g.* [40].

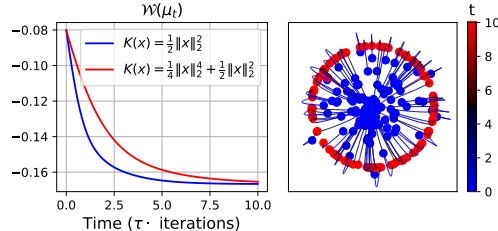
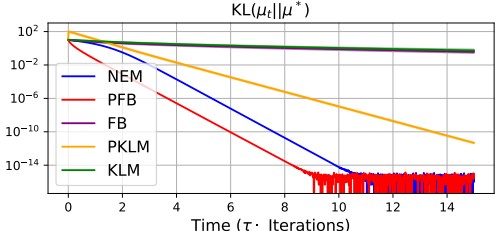

Figure 4: **(Left)** Value of $\mathcal{W}$ along the flow for two difference interaction Bregman potentials, **(Right)** Trajectories of particles to minimize $\mathcal{W}$.

Figure 5: Convergence towards Gaussian $\mathcal{N}(0, D)$ with $D$ diagonal and uniformly sampled on $[0, 50]^{10}$.

**Preconditioned gradient descent.** Plugging the empirical measure in (11), the preconditioned scheme can be implemented as

$$\forall k \geq 0, \forall i \in \{1, \ldots, n\}, \ x_i^{(k+1)} = x_i^{(k)} - \tau \nabla h^* \big( \nabla_{W_2} \mathcal{F}(\hat{\mu}_n^{(k)})(x_i^{(k)}) \big). \tag{122}$$

### G.2 Mirror descent of interaction energies

**Details of Section 5.** We detail in this Section the first experiment of Section 5. We aim at minimizing the interaction energy $\mathcal{W}(\mu) = \frac{1}{2} \iint W(x - y) \, d\mu(x) d\mu(y)$ for $W(z) = \frac{1}{4} \|z\|_2^4 - \frac{1}{2} \|z\|_2^2$. It is well-known that the stationary solution of its gradient flow is a Dirac ring [27]. Since the stationary solution is translation invariant, we project the measures to be centered.

We study here two Bregman potentials which are also interaction energies. First, observing that $\nabla^2 W(z) = 2zz^T + \big( \|z\|_2^2 - 1 \big) I_d$, we have for all $z$,

$$\|\nabla^2 W\|_{\text{op}} \leq 2\|z\|_2^2 + \|z\|_2^2 + 1 = 3\|z\|_2^2 + 1 = p_2(\|z\|_2), \tag{123}$$

with $p_2(t) = 3t^2 + 1$. Thus, by [88, Remark 2], $W$ is $\beta$-smooth relative to $K_4(z) = \frac{1}{4} \|z\|_2^4 + \frac{1}{2} \|z\|_2^2$ with $\beta = 4$. Thus, using Proposition 23, $\tilde{\mathcal{W}}_\mu$ is $\beta$-smooth relative to $\phi_\mu(T) = \frac{1}{2} \iint K \big( T(x) - T(x') \big) \, d\mu(x) d\mu(x')$ for all $\mu$, and we can apply Proposition 3.

Under the additional hypothesis that the measures are compactly supported, and thus there exists $M > 0$ such that $\|x\|_2^2 \leq M$ for $\mu$-almost every $x$, we can also show that $W$ is $\beta$-smooth relative to $K_2(z) = \frac{1}{2} \|z\|_2^2$. Indeed, on one hand, $\nabla^2 K = I_d$ and $\nabla^2 W(z) = 2zz^T + \big( \|z\|_2^2 - 1 \big) I_d$. Thus, for all $v, z \in \mathbb{R}^d$,

$$v^T \nabla^2 W(z) v = 2\langle z, v \rangle^2 + (\|z\|_2^2 - 1)\|v\|_2^2 \leq 3\|z\|_2^2 \|v\|_2^2 \leq 3M\|v\|_2^2 = 3M v^T \nabla^2 K(z) v. \tag{124}$$

In Figure 4, we plot the evolution of $\mathcal{W}$ along the flows obtained with these two Bregman potential, starting from $\mu_0 = \mathcal{N}(0, 0.25^2 I_2)$ for $n = 100$ particles, with a step size of $\tau = 0.1$ for 120 epochs.

**Ill-conditioned interaction energy.** We also study the minimization of an interaction energy with an ill-conditioned kernel $W(z) = \frac{1}{4}(z^T \Sigma^{-1} z)^2 - \frac{1}{2} z^T \Sigma^{-1} z$ where $\Sigma \in S_d^{++}(\mathbb{R})$ but is possibly badly conditioned, *i.e.* the ratio between the largest and smallest eigenvalues is large. In this case, the stationary solution becomes an ellipsoid instead of a ring. In our experiments, we take $\Sigma = \text{diag}(100, 0.1)$. For each scheme, we use $\mu_0 = \mathcal{N}(0, 0.25^2 I_2)$, $n = 100$ particles and a step size of $\tau = 0.1$.

On Figure 1, we use Bregman potentials which take into account this conditioning, namely we use $K_2^\Sigma(z) = \frac{1}{2} z^T \Sigma^{-1} z$ and $K_4^\Sigma(z) = \frac{1}{4}(z^T \Sigma^{-1} z)^2 - \frac{1}{2}(z^T \Sigma^{-1} z)$, and we observe that the convergence is much faster compared to the same kernels without preconditioning. For $K_2^\Sigma(z) = \frac{1}{2} z^T \Sigma^{-1} z$, the scheme becomes

$$(\nabla K \star \mu_{k+1}) \circ T_{k+1} = \nabla K \star \mu_k - \gamma \nabla_{W_2} \mathcal{F}(\mu_k)$$
$$\iff \Sigma^{-1} \big( T_{k+1} - m(\mu_{k+1}) \big) = \Sigma^{-1} \big( \text{Id} - m(\mu_k) \big) - \gamma \Sigma^{-1} (\text{Id}^T \Sigma^{-1} \text{Id} - 1) \text{Id} \tag{125}$$
$$\iff T_{k+1} - m(\mu_{k+1}) = \text{Id} - m(\mu_k) - \gamma (\text{Id}^T \Sigma^{-1} \text{Id} - 1) \text{Id}.$$

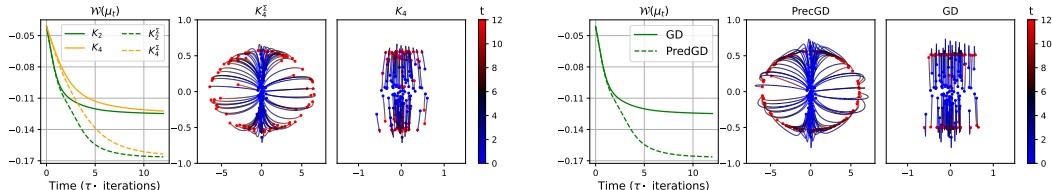

Figure 6: **(Left)** Value of $\mathcal{W}$ over time and trajectory of particles using $K_4$ and $K_4^\Sigma$ as interaction kernels. **(Right)** Value of $\mathcal{W}$ over time and trajectory of particles for the Wasserstein gradient descent and preconditioned Wasserstein gradient descent (with the ideal preconditioner $h^*(x) = \frac{1}{2}x^T\Sigma x$).

Thus, we see that $\Sigma^{-1}$ has less influence which might explain the faster convergence.

Similarly as in the case without preconditioning, using that $\nabla^2 W(z) = 2\Sigma^{-1}zz^T\Sigma^{-1} + (z^T\Sigma^{-1}z - 1)\Sigma^{-1}$, we can show that

$$v^T\nabla^2 W(z)v = 2\langle z, v\rangle_{\Sigma^{-1}}^2 + (\|z\|_{\Sigma^{-1}}^2 - 1)\|v\|_{\Sigma^{-1}}^2 \leq 3M\|v\|_{\Sigma^{-1}}^2 = 3Mv^T\nabla^2 K(z)v. \quad (126)$$

For the sake of comparison, we also report on Figure 6 the trajectories of particles for the use of $K_4$ and $K_4^\Sigma$, as well as of the usual Wasserstein gradient descent and the preconditioned Wasserstein gradient descent obtained with $h^*(x) = \frac{1}{2}x^T\Sigma x$ (which is equivalent to the Mirror Descent with $\phi_\mu^V$ as Bregman potential and $V(x) = \frac{1}{2}x^T\Sigma^{-1}x$). We observe almost the same trajectories as $K_2$, which would indicate that the target is also smooth compared to $\phi_\mu^V$.

**Runtime.** These experiments were run on a personal Laptop with a CPU Intel Core i5-9300H. For the interaction energy as Bregman potential, running the algorithm with Newton's method for $n = 100$ particles in dimension $d = 2$ for 120 epochs took about 5mn for $K_2$ and $K_2^\Sigma$, and about 1h for $K_4$ and $K_4^\Sigma$.

### G.3    Mirror descent on Gaussians

As the mirror descent scheme cannot be computed in closed-form for Bregman potentials which are not potential energies, and thus are computationally costly, we propose here to restrain ourselves to the Gaussian setting.

We choose as target distribution $\nu = \mathcal{N}(0, \Sigma)$ for $\Sigma$ a symmetric positive definite matrix in $\mathbb{R}^{10\times10}$, and the functional to be optimized is $\mathcal{F}(\mu) = \int V\,\mathrm{d}\mu + \mathcal{H}(\mu)$ with $V(x) = \frac{1}{2}x^T\Sigma^{-1}x$. The initial distribution is always chosen as $\mu_0 = \mathcal{N}(0, I_d)$. In all cases, the step size is chosen as $\tau = 0.01$, and we run the scheme for 1500 iterations. For the target distributions, we sample 20 random covariances of the form $\Sigma = UDU^T$ with $D$ evenly spaced in log scale between 1 and 100, and $U \in \mathbb{R}^{10\times10}$ chosen as a uniformly random orthogonal matrix, as in [43], and we report the averaged KL divergence over iterations in Figure 2. We add on Figure 5 the same experiments with targets of the form $\mathcal{N}(0, D)$ where $D$ is a diagonal matrix on $\mathbb{R}^{10\times10}$ sampled uniformly over $[0, 50]^{10}$. We compare here the Forward-Backward (FB) scheme of [43], the ideally preconditioned Forward-Backward scheme (PFB), which uses the closed-form (116) derived in Appendix F with $\Lambda = \Sigma$, and the Mirror Descent with negative entropy Bregman potential (NEM), whose closed-form was derived in Appendix D.3, and which we recall:

$$\forall k \geq 0, \ \Sigma_{k+1}^{-1} = \big((1-\tau)\Sigma_k^{-1} + \tau\Sigma^{-1}\big)^T\Sigma_k\big((1-\tau)\Sigma_k^{-1} + \tau\Sigma^{-1}\big). \quad (127)$$

We also experiment with the KL divergence as Bregman potential (KLM) and the ideally preconditioned KL divergence (PKLM). We observe that, even though the objective is convex relative to the Bregman potential, this scheme does not always converge. It might be due to its gradient which might not always be invertible. We leave further investigations for future works.

**Remark 2.** *We note that using as Bregman potential $\phi_\mu(\mathrm{T}) = \int \psi \circ \mathrm{T}\,\mathrm{d}\mu$ for $\psi(x) = \frac{1}{2}x^T\Lambda^{-1}x$ is equivalent to using a preconditioner with $h^*(x) = \frac{1}{2}x^T\Lambda x$.*

**Analysis of the convergence.** It is well-known that along the Wasserstein gradient flow of the KL divergence starting from a Gaussian and with a Gaussian target (Ornstein-Uhlenbeck process),

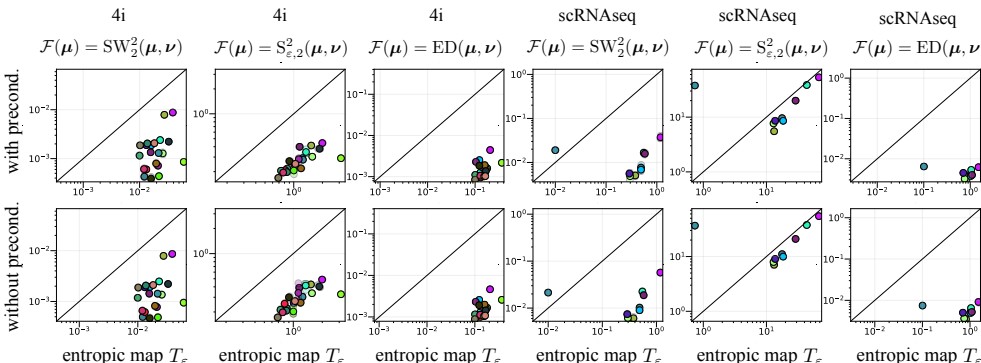

Figure 7: Preconditioned GD and (vanilla) GD vs. the entropic map $T_\varepsilon$ [101] to predict the responses of cell populations to cancer treatments on 4i and scRNAseq datasets, providing respectively 34 and 9 treatment responses. For each profiling technology and each treatment, we have a pair $(\mu_i, \nu_i)$ of source (untreated) cells and target (treated) cells. For each pair $(\mu_i, \nu_i)$, with both preconditioned GD and vanilla GD, we minimize the functional $\mathcal{F}(\mu) = D(\mu, \nu_i)$–with D a metric–to recover the effect of the perturbation. In both cases, the prediction is obtained by $\hat\mu_i = \min_\mu \mathcal{F}(\mu)$. We then fit an entropic map $T_\varepsilon$ and predict $T_\varepsilon \sharp \mu_i$. We then compare the objective function values $\mathcal{F}(\hat\mu_i)$ and $\mathcal{F}(T_\varepsilon \sharp \hat\mu_i)$. A point below the diagonal $y = x$ then refers to an experiment in which (preconditioned) WGD provides a better estimate of the perturbed population.

the measures stay Gaussian [129]. Thus, the Forward-Backward scheme has Gaussian iterates at each step [43, 109]. In this work, we also use a linearly preconditioned Forward-Backward scheme, whose closed-form is derived in (116) (Appendix F). For the Bregman potential, we choose $\phi_\mu(\mathrm{T}) = \int \psi \circ \mathrm{T} \, d\mu$ for $\psi(x) = \frac{1}{2} x^T \Sigma x$. In this situation, $\mathcal{G}(\mu) = \int V \, d\mu$ is 1-smooth and 1-convex relative to $\phi$. Thus, we can apply Proposition 25. We refer to Appendix F for more details on the convexity of $\mathcal{H}$.

For Bregman potentials whose gradient is not affine, the distributions do not necessarily stay Gaussian along the flows. Thus, we work on the Bures-Wasserstein space and use the Bures-Wasserstein gradient, *i.e.* we project the gradient on the space of affine maps with symmetric linear term, *i.e.* of the form $\mathrm{T}(x) = b + S(x - m)$ with $S \in S_d(\mathbb{R})$ [43]. We refer to [43, 73] for more details on this submanifold. This can be seen as performing Variational Inference. We derive the closed-form of the different schemes in Appendix D.3.

Even though these procedures do not fit exactly the theory developed in this work, we show the relative smoothness of $\mathcal{F}$ relative to $\mathcal{H}$ along the curve $\mu_t = \big((1 - t)\mathrm{Id} + t\mathrm{T}_{k+1}\big)_\# \mu_k$ under the hypothesis that the covariances matrices have bounded eigenvalues. Moreover, since $d_{\tilde{\mathcal{F}}_\mu} = d_{\phi_\mu^V} + d_{\tilde{\mathcal{H}}_\mu} \geq d_{\tilde{\mathcal{H}}_\mu}$, $\mathcal{F}$ is also 1-convex relative to $\mathcal{H}$.

**Proposition 27.** *Let $\lambda > 0$, $\mathcal{F}(\mu) = \int V \, d\mu + \mathcal{H}(\mu)$ with $V(x) = \frac{1}{2} x^T \Sigma^{-1} x$ where $\Sigma \in S_d^{++}(\mathbb{R})$ and $\Sigma \preceq \lambda I_d$. Suppose that for all $k \geq 0$, $(1 - \tau)\Sigma_{k+1}\Sigma_k^{-1} + \tau \Sigma_{k+1}\Sigma^{-1} \succeq 0$. Then, $\mathcal{F}$ is smooth relative to $\mathcal{H}$ along $\mu_t = \big((1 - t)\mathrm{Id} + t\mathrm{T}_{k+1}\big)_\# \mu_k$ where $\mu_k = \mathcal{N}(0, \Sigma_k)$ with $\Sigma_k \in S_d^{++}(\mathbb{R})$, $\Sigma_k \preceq \lambda I_d$.*

*Proof.* See Appendix H.14. □

## G.4 Single-cell experiments

First, we provide more details on the experiment on single cells of Section 5. Then, we detail a second experiment comparing the method with using a static map.

**Details on the metrics.** We show the benefits of using the polynomial preconditioner over the single-cell datasets for different metrics.

- The first one considered is the Sliced-Wasserstein distance [16, 102], defined as

$$\forall \mu, \nu \in \mathcal{P}_2(\mathbb{R}^d), \ \mathrm{SW}_2^2(\mu, \nu) = \int_{S^{d-1}} \mathrm{W}_2^2(P_\#^\theta \mu, P_\#^\theta \nu) \, \mathrm{d}\lambda(\theta), \tag{128}$$

where $S^{d-1} = \{\theta \in \mathbb{R}^d, \ \|\theta\|_2 = 1\}$, $\lambda$ denotes the uniform distribution on $S^{d-1}$ and for all $\theta \in S^{d-1}$, $x \in \mathbb{R}^d$, $P^\theta(x) = \langle x, \theta \rangle$. For $\mathcal{F}(\mu) = \frac{1}{2}\mathrm{SW}_2^2(\mu, \nu)$, the Wasserstein gradient can be computed as [18]

$$\nabla_{\mathrm{W}_2} \mathcal{F}(\mu) = \int_{S^{d-1}} \psi_\theta'\big(P^\theta(x)\big)\theta \, \mathrm{d}\lambda(\theta), \tag{129}$$

where, for $t \in \mathbb{R}$, $\psi_\theta'(t) = t - F_{P_\#^\theta \nu}^{-1}\big(F_{P_\#^\theta \mu}(t)\big)$ with $F_{P_\#^\theta \mu}$ the cumulative distribution function of $P_\#^\theta \mu$. In practice, we compute SW and its gradient using a Monte-Carlo approximation by first drawing $L$ uniform random directions $\theta_1, \ldots, \theta_L$.

- The second one considered is the Sinkhorn divergence [50] defined as

$$\forall \mu, \nu \in \mathcal{P}_2(\mathbb{R}^d), \ \mathrm{S}_{\varepsilon,2}^2(\mu, \nu) = \mathrm{OT}_\varepsilon(\mu, \nu) - \frac{1}{2}\mathrm{OT}_\varepsilon(\mu, \mu) - \frac{1}{2}\mathrm{OT}_\varepsilon(\nu, \nu), \tag{130}$$

with

$$\mathrm{OT}_\varepsilon(\mu, \nu) = \inf_{\gamma \in \Pi(\mu, \nu)} \int \|x - y\|_2^2 \, \mathrm{d}\gamma(x, y) + \varepsilon \mathrm{KL}(\gamma \| \mu \otimes \nu), \tag{131}$$

the entropic regularized OT. The Wasserstein gradient of $\mathrm{S}_{\varepsilon,2}^2$ is simply obtained as the potential [50].

- Finally, we also consider the energy distance, defined as

$$\forall \mu, \nu \in \mathcal{P}_2(\mathbb{R}^d), \ \mathrm{ED}(\mu, \nu) = -\iint \|x - y\|_2 \, \mathrm{d}(\mu - \nu)(x)\mathrm{d}(\mu - \nu)(y). \tag{132}$$

To compute its Wasserstein gradient, we use the sliced procedure of [60].

**Parameters chosen.** For all the metrics, we fixed the step size at $\tau = 1$. To choose the parameter $a$ of the preconditioner $h^*(x) = (\|x\|_2^a + 1)^{1/a} - 1$, we ran a grid search over $a \in \{1.25, 1.5, 1.75\}$ for a random treatment, and used it for all the others. In particular, we used for the dataset 4i $a = 1.5$ for the Sinkhorn divergence and for SW, and $a = 1.75$ for the energy distance. For the scRNAseq dataset, we used $a = 1.25$ for the Sinkhorn divergence and SW, and $a = 1.5$ for the energy distance. We note that for the dataset 4i, the data lie in dimension $d = 48$ and $d = 50$ for scRNAseq. For all the metrics, we first sampled 4096 particles from the source (untreated) dataset, and used in average between 2000 and 3000 samples from the target dataset. For the test value, we also added 40% of unseen cells following [21]. Note that we reported the results in Figure 3 for 3 different initializations for each treatment, and reported these results with their mean. We report the results using a fixed relative tolerance tol $= 10^{-3}$, *i.e.* at the first iteration where $|\mathcal{F}(\mu_k) - \mathcal{F}(\mu_{k-1})|/\mathcal{F}(\mu_{k-1}) \le$ tol, with a maximum value of iterations of $10^4$. For the Sinkhorn divergence, we chose $\varepsilon$ as $10^{-1}$ time the variance of the target. Finally, for SW and the computation of the gradient of the energy distance, we used a Monte-Carlo approximation with $L = 1024$ projections.

**Comparison to an OT static map.** We now compare the prediction of the response of cells to a perturbation using Wasserstein gradient descent, with and without preconditioning, to the one provided by a static estimator, the entropic map $T_\varepsilon$ [101]. This experiment motivates the use of a dynamic procedure, iterating multiple steps to map the unperturbed population $\mu$ to the perturbed population $\nu$, instead of a unique static step. We use the proteomic dataset [21] as the one considered in 3. We use the default OTT-JAX [38] of $T_\varepsilon$. The results are shown in Figure 7.

**Runtime.** For this experiment, we used a GPU Tesla P100-PCIE-16GB. Depending on the convergence and on the metric considered, each run took in between 30s and 10mn. So in total, it took a few hundred of hours of computation time.

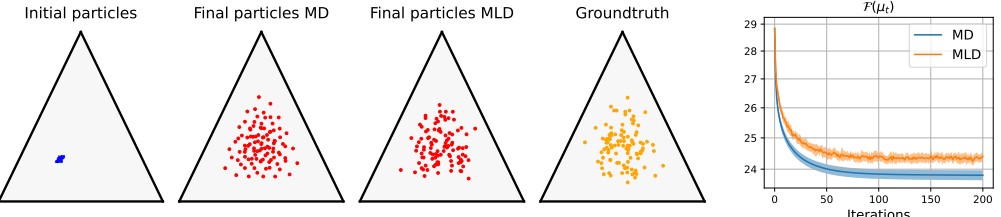

Figure 8: **(Left)** Samples from a Dirichlet posterior distribution for Mirror Descent (MD) and Mirror Langevin (MLD). **(Right)** Evolution of the objective averaged over 20 different initialisations.

### G.5 Mirror descent on the simplex

We can also leverage the mirror map to perform sampling in constrained spaces. This has received a lot of attention recently either through mirror Langevin methods [3, 32, 116], diffusion methods [51, 82], mirror SVGD [113, 114] or other MCMC algorithms [52, 94].

The goal here is to sample from a Dirichlet distribution, *i.e.* from a distribution $\nu \propto e^{-V}$ where $V(x) = -\sum_{i=1}^d a_i \log(x_i) - a_{d+1} \log \left(1 - \sum_{i=1}^d x_i\right)$. To sample from such a distribution, we minimize the Kullback-Leibler divergence, *i.e.* $\mathcal{F}(\mu) = \mathrm{KL}(\mu\|\nu) = \int V \, \mathrm{d}\mu + \mathcal{H}(\mu)$. To stay on the (open) simplex $\Delta_d = \{x \in \mathbb{R}^{d+1}, x_i > 0, \sum_{i=1}^{d+1} x_i < 1\}$, we use the mirror map $\phi(\mu) = \int \psi \mathrm{d}\mu$ with $\psi(x) = \sum_{i=1}^d x_i \log(x_i) + (1 - \sum_i x_i) \log(1 - \sum_i x_i)$ for which

$$\nabla\psi(x) = \left(\log x_i - \log\left(1 - \sum_j x_j\right)\right)_i, \quad \nabla\psi^*(y) = \left(\frac{e^{y_i}}{1 + \sum_j e^{y_j}}\right)_i. \tag{133}$$

The scheme here is given by $\mathrm{T}_{k+1} = \nabla\psi^* \circ (\nabla\psi - \gamma \nabla_{\mathrm{W}_2}\mathcal{F}(\mu_k))$, where $\nabla_{\mathrm{W}_2}\mathcal{F}(\mu_k) = \nabla V + \nabla \log \mu_k$, with the density of $\mu_k$ estimated through a Kernel Density Estimator (KDE). We plot on Figure 8a the results obtained for $d = 2$, $a_1 = a_2 = a_3 = 6$ and 100 samples. We also report the results for the Mirror Langevin Dynamic (MLD) algorithm, which provide iid samples, which are thus less ordered. We plot the evolution of the KL over iterations on Figure 8b (where the entropy is estimated using the Kozachenko-Leonenko estimator [42]).

The KDE used here will not scale well with the dimension, however, different methods have been recently propose to overcome this issue, such as using projection on lower dimensional subspaces [127], or using neural networks to learn ratio density estimators [6, 49, 128].

## H  Proofs

### H.1  Proof of Proposition 1

Let $\mu \in \mathcal{P}_2(\mathbb{R}^d)$, $\mathrm{S}, \mathrm{T} \in D(\tilde{\mathcal{F}}_\mu)$, $\epsilon > 0$. Since $\mathcal{F}$ is Wasserstein differentiable at $\mathrm{S}_{\#}\mu$, applying Proposition 9 at $\mathrm{S}_{\#}\mu$ with $\nu = (\mathrm{S} + \epsilon(\mathrm{T} - \mathrm{S}))_{\#}\mu$ and $\gamma = (\mathrm{S}, \mathrm{S} + \epsilon(\mathrm{T} - \mathrm{S}))_{\#}\mu \in \Pi(\mathrm{S}_{\#}\mu, \nu)$, we

obtain,

$$
\begin{aligned}
\tilde{\mathcal{F}}_\mu\big(\mathrm{S} + \epsilon(\mathrm{T}-\mathrm{S})\big) &= \mathcal{F}\big((\mathrm{S}+\epsilon(\mathrm{T}-\mathrm{S}))_{\#}\mu\big) \\
&= \mathcal{F}(\mathrm{S}_{\#}\mu) + \int \langle \nabla_{\mathrm{W}_2}\mathcal{F}(\mathrm{S}_{\#}\mu)(x), y - x \rangle \, \mathrm{d}\gamma(x,y) \\
&\quad + o\left( \sqrt{\int \|x-y\|_2^2 \, \mathrm{d}\gamma(x,y)} \right) \\
&= \tilde{\mathcal{F}}_\mu(\mathrm{S}) + \epsilon \int \langle \nabla_{\mathrm{W}_2}\mathcal{F}(\mathrm{S}_{\#}\mu)\big(\mathrm{S}(x)\big), \mathrm{T}(x) - \mathrm{S}(x) \rangle \, \mathrm{d}\mu(x) \\
&\quad + o\left( \epsilon\sqrt{\int \|\mathrm{T}(x) - \mathrm{S}(x)\|_2^2 \, \mathrm{d}\mu(x)} \right) \\
&= \tilde{\mathcal{F}}_\mu(\mathrm{S}) + \epsilon \langle \nabla_{\mathrm{W}_2}\mathcal{F}(\mathrm{S}_{\#}\mu) \circ \mathrm{S}, \mathrm{T} - \mathrm{S}\rangle_{L^2(\mu)} + \epsilon o(\|\mathrm{T}-\mathrm{S}\|_{L^2(\mu)}). \quad (134)
\end{aligned}
$$

Thus, $\delta\tilde{\mathcal{F}}_\mu(\mathrm{S}, \mathrm{T}-\mathrm{S}) = \langle \nabla_{\mathrm{W}_2}\mathcal{F}(\mathrm{S}_{\#}\mu) \circ \mathrm{S}, \mathrm{T}-\mathrm{S} \rangle_{L^2(\mu)}$. Note that in the third equality we used that $\nabla_{\mathrm{W}_2}\mathcal{F}(\mu) \in L^2(\mu)$. $\qquad\square$

## H.2  Proof of Proposition 2

Let $\mu, \rho \in \mathcal{P}_{2,\mathrm{ac}}(\mathbb{R}^d)$ and $\nu \in \mathcal{P}_2(\mathbb{R}^d)$. Define $\mathrm{T}_{\phi_\mu}^{\mu,\nu} = \mathrm{argmin}_{\mathrm{T}_{\#}\mu=\nu} \, \mathrm{d}_{\phi_\mu}(\mathrm{T}, \mathrm{Id})$, $\mathrm{U}_{\phi_\rho}^{\rho,\nu} = \mathrm{argmin}_{\mathrm{U}_{\#}\rho=\nu} \, \mathrm{d}_{\phi_\rho}(\mathrm{U}, \mathrm{Id})$ and let $\mathrm{S} \in L^2(\mu)$ such that $\mathrm{S}_{\#}\mu = \rho$. Then, noticing that $\gamma = (\mathrm{T}_{\phi_\mu}^{\mu,\nu}, \mathrm{S})_{\#}\mu \in \Pi(\nu, \rho)$, we have

$$
\begin{aligned}
\mathrm{d}_{\phi_\mu}(\mathrm{T}_{\phi_\mu}^{\mu,\nu}, \mathrm{S}) &= \phi\big((\mathrm{T}_{\phi_\mu}^{\mu,\nu})_{\#}\mu\big) - \phi(\mathrm{S}_{\#}\nu) - \int \langle \nabla_{\mathrm{W}_2}\phi(\mathrm{S}_{\#}\mu)(y), x - y \rangle \, \mathrm{d}(\mathrm{T}_{\phi_\mu}^{\mu,\nu}, \mathrm{S})_{\#}\mu(x,y) \\
&= \phi(\nu) - \phi(\rho) - \int \langle \nabla_{\mathrm{W}_2}\phi(\rho)(y), x - y\rangle \, \mathrm{d}\gamma(x,y) \\
&\geq \mathrm{W}_\phi(\nu, \rho) = \mathrm{d}_{\phi_\rho}(\mathrm{U}_{\phi_\rho}^{\rho,\nu}, \mathrm{Id}). \quad (135)
\end{aligned}
$$

In the last line, we used Proposition 15, *i.e.* that the optimal coupling is of the form $(\mathrm{U}_{\phi_\rho}^{\rho,\nu}, \mathrm{Id})_{\#}\rho$. $\qquad\square$

## H.3  Proof of Proposition 3

Let $\mathrm{T}_{k+1} = \mathrm{argmin}_{\mathrm{T} \in L^2(\mu_k)} \, \tau \langle \nabla_{\mathrm{W}_2}\mathcal{F}(\mu_k), \mathrm{T}-\mathrm{Id}\rangle_{L^2(\mu_k)} + \mathrm{d}_{\phi_{\mu_k}}(\mathrm{T}, \mathrm{Id})$. Applying the three-point inequality (Lemma 29) with $\psi(\mathrm{T}) = \tau \langle \nabla_{\mathrm{W}_2}\mathcal{F}(\mu_k), \mathrm{T} - \mathrm{Id}\rangle_{L^2(\mu_k)}$ which is convex, $\mathrm{T}_0 = \mathrm{Id}$ and $\mathrm{T}^* = \mathrm{T}_{k+1}$, we get for all $\mathrm{T} \in L^2(\mu_k)$,

$$
\begin{aligned}
\tau \langle &\nabla_{\mathrm{W}_2}\mathcal{F}(\mu_k), \mathrm{T}-\mathrm{Id}\rangle_{L^2(\mu_k)} + \mathrm{d}_{\phi_{\mu_k}}(\mathrm{T}, \mathrm{Id}) \\
&\geq \tau \langle \nabla_{\mathrm{W}_2}\mathcal{F}(\mu_k), \mathrm{T}_{k+1}-\mathrm{Id}\rangle_{L^2(\mu_k)} + \mathrm{d}_{\phi_{\mu_k}}(\mathrm{T}_{k+1}, \mathrm{Id}) + \mathrm{d}_{\phi_{\mu_k}}(\mathrm{T}, \mathrm{T}_{k+1}),
\end{aligned} \quad (136)
$$

which is equivalent to

$$
\begin{aligned}
\langle &\nabla_{\mathrm{W}_2}\mathcal{F}(\mu_k), \mathrm{T}_{k+1}-\mathrm{Id}\rangle_{L^2(\mu_k)} + \frac{1}{\tau}\mathrm{d}_{\phi_{\mu_k}}(\mathrm{T}_{k+1}, \mathrm{Id}) \\
&\leq \langle \nabla_{\mathrm{W}_2}\mathcal{F}(\mu_k), \mathrm{T}-\mathrm{Id}\rangle_{L^2(\mu_k)} + \frac{1}{\tau}\mathrm{d}_{\phi_{\mu_k}}(\mathrm{T}, \mathrm{Id}) - \frac{1}{\tau}\mathrm{d}_{\phi_{\mu_k}}(\mathrm{T}, \mathrm{T}_{k+1}).
\end{aligned} \quad (137)
$$

By the $\beta$-smoothness of $\tilde{\mathcal{F}}_{\mu_k}$ relative to $\phi_{\mu_k}$, we also have

$$
\mathrm{d}_{\tilde{\mathcal{F}}_{\mu_k}}(\mathrm{T}_{k+1}, \mathrm{Id}) = \tilde{\mathcal{F}}_{\mu_k}(\mathrm{T}_{k+1}) - \tilde{\mathcal{F}}_{\mu_k}(\mathrm{Id}) - \langle \nabla_{\mathrm{W}_2}\mathcal{F}(\mu_k), \mathrm{T}_{k+1}-\mathrm{Id}\rangle_{L^2(\mu_k)} \leq \beta\mathrm{d}_{\phi_{\mu_k}}(\mathrm{T}_{k+1}, \mathrm{Id})
$$

$$
\iff \tilde{\mathcal{F}}_{\mu_k}(\mathrm{T}_{k+1}) \leq \tilde{\mathcal{F}}_{\mu_k}(\mathrm{Id}) + \langle \nabla_{\mathrm{W}_2}\mathcal{F}(\mu_k), \mathrm{T}_{k+1}-\mathrm{Id}\rangle_{L^2(\mu_k)} + \beta\mathrm{d}_{\phi_{\mu_k}}(\mathrm{T}_{k+1}, \mathrm{Id}). \quad (138)
$$

Moreover, since $\beta \leq \frac{1}{\tau}$, this inequality implies (by non-negativity of $\mathrm{d}_{\phi_{\mu_k}}$),

$$
\tilde{\mathcal{F}}_{\mu_k}(\mathrm{T}_{k+1}) \leq \tilde{\mathcal{F}}_{\mu_k}(\mathrm{Id}) + \langle \nabla_{\mathrm{W}_2}\mathcal{F}(\mu_k), \mathrm{T}_{k+1}-\mathrm{Id}\rangle_{L^2(\mu_k)} + \frac{1}{\tau}\mathrm{d}_{\phi_{\mu_k}}(\mathrm{T}_{k+1}, \mathrm{Id}). \quad (139)
$$

Then, using the inequality (137), we obtain for all $T \in L^2(\mu_k)$,

$$\tilde{\mathcal{F}}_{\mu_k}(T_{k+1}) \leq \tilde{\mathcal{F}}_{\mu_k}(\mathrm{Id}) + \langle \nabla_{W_2}\mathcal{F}(\mu_k), T - \mathrm{Id}\rangle_{L^2(\mu_k)} + \frac{1}{\tau}d_{\phi_{\mu_k}}(T, \mathrm{Id}) - \frac{1}{\tau}d_{\phi_{\mu_k}}(T, T_{k+1}). \tag{140}$$

Observing that $\tilde{\mathcal{F}}_{\mu_k}(T_{k+1}) = \mathcal{F}(\mu_{k+1})$ and $\tilde{\mathcal{F}}_{\mu_k}(\mathrm{Id}) = \mathcal{F}(\mu_k)$, we get

$$\mathcal{F}(\mu_{k+1}) \leq \mathcal{F}(\mu_k) + \langle \nabla_{W_2}\mathcal{F}(\mu_k), T - \mathrm{Id}\rangle_{L^2(\mu_k)} + \frac{1}{\tau}d_{\phi_{\mu_k}}(T, \mathrm{Id}) - \frac{1}{\tau}d_{\phi_{\mu_k}}(T, T_{k+1}). \tag{141}$$

Finally, setting $T = \mathrm{Id}$, we obtain the result:

$$\mathcal{F}(\mu_{k+1}) \leq \mathcal{F}(\mu_k) - \frac{1}{\tau}d_{\phi_{\mu_k}}(\mathrm{Id}, T_{k+1}). \tag{142}$$

$\square$

## H.4  Proof of Proposition 4

Let $\nu \in \mathcal{P}_2(\mathbb{R}^d)$, and $T = \mathrm{argmin}_{T, T_{\#}\mu_k = \nu} \ d_{\phi_{\mu_k}}(T, \mathrm{Id})$. From the relative convexity hypothesis, we have

$$\begin{aligned}
d_{\tilde{\mathcal{F}}_{\mu_k}}(T, \mathrm{Id}) &\geq \alpha d_{\phi_{\mu_k}}(T, \mathrm{Id}) \\
&\iff \tilde{\mathcal{F}}_{\mu_k}(T) - \tilde{\mathcal{F}}_{\mu_k}(\mathrm{Id}) - \langle \nabla_{W_2}\mathcal{F}(\mu_k), T - \mathrm{Id}\rangle_{L^2(\mu_k)} \geq \alpha d_{\phi_{\mu_k}}(T, \mathrm{Id}) \\
&\iff \tilde{\mathcal{F}}_{\mu_k}(T) - \alpha d_{\phi_{\mu_k}}(T, \mathrm{Id}) \geq \tilde{\mathcal{F}}_{\mu_k}(\mathrm{Id}) + \langle \nabla_{W_2}\mathcal{F}(\mu_k), T - \mathrm{Id}\rangle_{L^2(\mu_k)} \\
&\iff \mathcal{F}(\nu) - \alpha d_{\phi_{\mu_k}}(T, \mathrm{Id}) \geq \mathcal{F}(\mu_k) + \langle \nabla_{W_2}\mathcal{F}(\mu_k), T - \mathrm{Id}\rangle_{L^2(\mu_k)}.
\end{aligned} \tag{143}$$

Plugging this into (140), we get

$$\mathcal{F}(\mu_{k+1}) \leq \mathcal{F}(\nu) + \frac{1}{\tau}\big(d_{\phi_{\mu_k}}(T, \mathrm{Id}) - d_{\phi_{\mu_k}}(T, T_{k+1})\big) - \alpha d_{\phi_{\mu_k}}(T, \mathrm{Id}). \tag{144}$$

Then, by definition of $T$, note that $d_{\phi_{\mu_k}}(T, \mathrm{Id}) = W_\phi(\nu, \mu_k)$, and by Assumption 1, we have $d_{\phi_{\mu_k}}(T, T_{k+1}) \geq W_\phi(\nu, \mu_{k+1})$, since $T_{\#}\mu_k = \nu$ and $(T_{k+1})_{\#}\mu_k = \mu_{k+1}$. Thus,

$$\mathcal{F}(\mu_{k+1}) - \mathcal{F}(\nu) \leq \left(\frac{1}{\tau} - \alpha\right) W_\phi(\nu, \mu_k) - \frac{1}{\tau}W_\phi(\nu, \mu_{k+1}). \tag{145}$$

Observing that $\mathcal{F}(\mu_k) \leq \mathcal{F}(\mu_\ell)$ for all $\ell \leq k$ (by Proposition 3 and non-negativity of $d_\phi$ for $\phi$ convex) and that $W_\phi(\nu, \mu) \geq 0$, we can apply Lemma 30 with $f = \mathcal{F}$, $c = \mathcal{F}(\nu)$ and $g = W_\phi(\nu, \cdot)$, and we obtain

$$\forall k \geq 1, \ \mathcal{F}(\mu_k) - \mathcal{F}(\nu) \leq \frac{\alpha}{\left(\frac{\frac{1}{\tau}}{\frac{1}{\tau}-\alpha}\right)^k - 1}W_\phi(\nu, \mu_0) \leq \frac{\frac{1}{\tau} - \alpha}{k}W_\phi(\nu, \mu_0). \tag{146}$$

For the second result, from (145), we get for $\nu = \mu^*$ the minimizer of $\mathcal{F}$, since $\mathcal{F}(\mu_{k+1}) - \mathcal{F}(\mu^*) \geq 0$,

$$W_\phi(\mu^*, \mu_{k+1}) \leq (1 - \alpha\tau) W_\phi(\mu^*, \mu_k) \leq (1 - \alpha\tau)^{k+1} W_\phi(\mu^*, \mu_0). \tag{147}$$

$\square$

## H.5 Proof of Proposition 5

Let $k \geq 0$, by the definition of $\mathrm{d}_{\phi_{\mu_k}^{h^*}}$ and the hypothesis $\mathrm{d}_{\phi_{\mu_k}^{h^*}}\left(\nabla_{\mathrm{W}_2}\mathcal{F}(\mu_{k+1}) \circ \mathrm{T}_{k+1}, \nabla_{\mathrm{W}_2}\mathcal{F}(\mu_k)\right) \leq \beta \mathrm{d}_{\tilde{\mathcal{F}}_{\mu_k}}\left(\mathrm{Id}, \mathrm{T}_{k+1}\right)$, we have

$$
\begin{aligned}
\phi_{\mu_{k+1}}^{h^*}\left(\nabla_{\mathrm{W}_2}\mathcal{F}(\mu_{k+1})\right) &= \phi_{\mu_k}^{h^*}\left(\nabla_{\mathrm{W}_2}\mathcal{F}(\mu_k)\right) \\
&\quad + \langle \nabla h^* \circ \nabla_{\mathrm{W}_2}\mathcal{F}(\mu_k), \nabla_{\mathrm{W}_2}\mathcal{F}(\mu_{k+1}) \circ \mathrm{T}_{k+1} - \nabla_{\mathrm{W}_2}\mathcal{F}(\mu_k)\rangle_{L^2(\mu_k)} \\
&\quad + \mathrm{d}_{\phi_{\mu_k}^{h^*}}\left(\nabla_{\mathrm{W}_2}\mathcal{F}((\mathrm{T}_{k+1})_{\#}\mu_k) \circ \mathrm{T}_{k+1}, \nabla_{\mathrm{W}_2}\mathcal{F}(\mu_k)\right) \\
&\leq \phi_{\mu_k}^{h^*}\left(\nabla_{\mathrm{W}_2}\mathcal{F}(\mu_k)\right) \\
&\quad + \langle \nabla h^* \circ \nabla_{\mathrm{W}_2}\mathcal{F}(\mu_k), \nabla_{\mathrm{W}_2}\mathcal{F}(\mu_{k+1}) \circ \mathrm{T}_{k+1} - \nabla_{\mathrm{W}_2}\mathcal{F}(\mu_k)\rangle_{L^2(\mu_k)} \\
&\quad + \beta \mathrm{d}_{\tilde{\mathcal{F}}_{\mu_k}}\left(\mathrm{Id}, \mathrm{T}_{k+1}\right) \\
&\leq \phi_{\mu_k}^{h^*}\left(\nabla_{\mathrm{W}_2}\mathcal{F}(\mu_k)\right) \\
&\quad + \langle \nabla h^* \circ \nabla_{\mathrm{W}_2}\mathcal{F}(\mu_k), \nabla_{\mathrm{W}_2}\mathcal{F}(\mu_{k+1}) \circ \mathrm{T}_{k+1} - \nabla_{\mathrm{W}_2}\mathcal{F}(\mu_k)\rangle_{L^2(\mu_k)} \\
&\quad + \frac{1}{\tau}\mathrm{d}_{\tilde{\mathcal{F}}_{\mu_k}}\left(\mathrm{Id}, \mathrm{T}_{k+1}\right),
\end{aligned}
\tag{148}
$$

where we used in the last line that $\tau \leq \frac{1}{\beta}$ and the non-negativity of the Bregman divergence since $\mathcal{F}$ is convex along $t \mapsto \left((1-t)\mathrm{T}_{k+1} + t\mathrm{Id}\right)_{\#}\mu_k$ and thus by Proposition 13, $\mathrm{d}_{\tilde{\mathcal{F}}_{\mu_k}}\left(\mathrm{Id}, \mathrm{T}_{k+1}\right) \geq 0$.

Let $\mathrm{T} \in L^2(\mu_k)$. Then, using the three-point identity (Lemma 28) (with $\mathrm{S} = \mathrm{Id}$, $\mathrm{U} = \mathrm{T}$ and $\mathrm{T} = \mathrm{T}_{k+1}$), and remembering that $\mathrm{T}_{k+1} = \mathrm{Id} - \tau \nabla h^* \circ \nabla_{\mathrm{W}_2}\mathcal{F}(\mu_k)$, we get

$$
\begin{aligned}
\mathrm{d}_{\tilde{\mathcal{F}}_{\mu_k}}\left(\mathrm{Id}, \mathrm{T}_{k+1}\right) &= \mathrm{d}_{\tilde{\mathcal{F}}_{\mu_k}}\left(\mathrm{Id}, \mathrm{T}\right) - \mathrm{d}_{\tilde{\mathcal{F}}_{\mu_k}}\left(\mathrm{T}_{k+1}, \mathrm{T}\right) \\
&\quad - \langle \nabla_{\mathrm{W}_2}\mathcal{F}\left((\mathrm{T}_{k+1})_{\#}\mu_k\right) \circ \mathrm{T}_{k+1}, \mathrm{Id} - \mathrm{T}_{k+1}\rangle_{L^2(\mu_k)} \\
&\quad + \langle \nabla_{\mathrm{W}_2}\mathcal{F}(\mathrm{T}_{\#}\mu_k) \circ \mathrm{T}, \mathrm{Id} - \mathrm{T}_{k+1}\rangle_{L^2(\mu_k)} \\
&= \mathrm{d}_{\tilde{\mathcal{F}}_{\mu_k}}\left(\mathrm{Id}, \mathrm{T}\right) - \mathrm{d}_{\tilde{\mathcal{F}}_{\mu_k}}\left(\mathrm{T}_{k+1}, \mathrm{T}\right) \\
&\quad + \langle \nabla_{\mathrm{W}_2}\mathcal{F}(\mathrm{T}_{\#}\mu_k) \circ \mathrm{T} - \nabla_{\mathrm{W}_2}\mathcal{F}(\mu_{k+1}) \circ \mathrm{T}_{k+1}, \mathrm{Id} - \mathrm{T}_{k+1}\rangle_{L^2(\mu_k)} \\
&= \mathrm{d}_{\tilde{\mathcal{F}}_{\mu_k}}\left(\mathrm{Id}, \mathrm{T}\right) - \mathrm{d}_{\tilde{\mathcal{F}}_{\mu_k}}\left(\mathrm{T}_{k+1}, \mathrm{T}\right) \\
&\quad + \tau\langle \nabla_{\mathrm{W}_2}\mathcal{F}(\mathrm{T}_{\#}\mu_k) \circ \mathrm{T} - \nabla_{\mathrm{W}_2}\mathcal{F}(\mu_{k+1}) \circ \mathrm{T}_{k+1}, \nabla h^* \circ \nabla_{\mathrm{W}_2}\mathcal{F}(\mu_k)\rangle_{L^2(\mu_k)}.
\end{aligned}
\tag{149}
$$

This is equivalent to

$$
\begin{aligned}
\langle \nabla h^* &\circ \nabla_{\mathrm{W}_2}\mathcal{F}(\mu_k), \nabla_{\mathrm{W}_2}\mathcal{F}(\mu_{k+1}) \circ \mathrm{T}_{k+1} - \nabla_{\mathrm{W}_2}\mathcal{F}(\mu_k)\rangle_{L^2(\mu_k)} + \frac{1}{\tau}\mathrm{d}_{\tilde{\mathcal{F}}_{\mu_k}}\left(\mathrm{Id}, \mathrm{T}_{k+1}\right) \\
&= \frac{1}{\tau}\mathrm{d}_{\tilde{\mathcal{F}}_{\mu_k}}\left(\mathrm{Id}, \mathrm{T}\right) - \frac{1}{\tau}\mathrm{d}_{\tilde{\mathcal{F}}_{\mu_k}}\left(\mathrm{T}_{k+1}, \mathrm{T}\right) \\
&\quad + \langle \nabla_{\mathrm{W}_2}\mathcal{F}(\mathrm{T}_{\#}\mu_k) \circ \mathrm{T} - \nabla_{\mathrm{W}_2}\mathcal{F}(\mu_k), \nabla h^* \circ \nabla_{\mathrm{W}_2}\mathcal{F}(\mu_k)\rangle_{L^2(\mu_k)}.
\end{aligned}
\tag{150}
$$

Then, using the definition of $\mathrm{d}_{\phi_{\mu_k}^{h^*}}\left(\nabla_{\mathrm{W}_2}\mathcal{F}(\mathrm{T}_{\#}\mu_k) \circ \mathrm{T}, \nabla_{\mathrm{W}_2}\mathcal{F}(\mu_k)\right)$, we obtain

$$
\begin{aligned}
\langle \nabla h^* &\circ \nabla_{\mathrm{W}_2}\mathcal{F}(\mu_k), \nabla_{\mathrm{W}_2}\mathcal{F}(\mu_{k+1}) \circ \mathrm{T}_{k+1} - \nabla_{\mathrm{W}_2}\mathcal{F}(\mu_k)\rangle_{L^2(\mu_k)} + \frac{1}{\tau}\mathrm{d}_{\tilde{\mathcal{F}}_{\mu_k}}\left(\mathrm{Id}, \mathrm{T}_{k+1}\right) \\
&= \frac{1}{\tau}\mathrm{d}_{\tilde{\mathcal{F}}_{\mu_k}}\left(\mathrm{Id}, \mathrm{T}\right) - \frac{1}{\tau}\mathrm{d}_{\tilde{\mathcal{F}}_{\mu_k}}\left(\mathrm{T}_{k+1}, \mathrm{T}\right) \\
&\quad - \mathrm{d}_{\phi_{\mu_k}^{h^*}}\left(\nabla_{\mathrm{W}_2}\mathcal{F}(\mathrm{T}_{\#}\mu_k) \circ \mathrm{T}, \nabla_{\mathrm{W}_2}\mathcal{F}(\mu_k)\right) + \phi_{\mu_k}^{h^*}\left(\nabla_{\mathrm{W}_2}\mathcal{F}(\mathrm{T}_{\#}\mu_k) \circ \mathrm{T}\right) - \phi_{\mu_k}^{h^*}\left(\nabla_{\mathrm{W}_2}\mathcal{F}(\mu_k)\right).
\end{aligned}
\tag{151}
$$

Plugging this into (148), we get

$$
\begin{aligned}
\phi_{\mu_{k+1}}^{h^*}\left(\nabla_{\mathrm{W}_2}\mathcal{F}(\mu_{k+1})\right) &\leq \phi_{\mu_k}^{h^*}\left(\nabla_{\mathrm{W}_2}\mathcal{F}(\mathrm{T}_{\#}\mu_k) \circ \mathrm{T}\right) + \frac{1}{\tau}\mathrm{d}_{\tilde{\mathcal{F}}_{\mu_k}}\left(\mathrm{Id}, \mathrm{T}\right) - \frac{1}{\tau}\mathrm{d}_{\tilde{\mathcal{F}}_{\mu_k}}\left(\mathrm{T}_{k+1}, \mathrm{T}\right) \\
&\quad - \mathrm{d}_{\phi_{\mu_k}^{h^*}}\left(\nabla_{\mathrm{W}_2}\mathcal{F}(\mathrm{T}_{\#}\mu_k) \circ \mathrm{T}, \nabla_{\mathrm{W}_2}\mathcal{F}(\mu_k)\right).
\end{aligned}
\tag{152}
$$

For $T = Id$, we get

$$\phi_{\mu_{k+1}}^{h^*}\big(\nabla_{W_2}\mathcal{F}(\mu_{k+1})\big) \leq \phi_{\mu_k}^{h^*}\big(\nabla_{W_2}\mathcal{F}(\mu_k)\big) - \frac{1}{\tau}d_{\tilde{\mathcal{F}}_{\mu_k}}(T_{k+1}, Id). \tag{153}$$

$\square$

## H.6 Proof of Proposition 6

Let $\mu^* \in \mathcal{P}_2(\mathbb{R}^d)$ be the minimizer of $\mathcal{F}$, $k \geq 0$ and $T = \operatorname{argmin}_{T \in L^2(\mu_k), T_\# \mu_k = \mu^*} d_{\tilde{\mathcal{F}}_{\mu_k}}(Id, T)$. First, observe that since $\mu^*$ is the minimizer of $\mathcal{F}$, then $\nabla_{W_2}\mathcal{F}(\mu^*) = 0$ (see *e.g.* [74, Theorem 3.1]), and thus $\phi_{\mu_k}^{h^*}(0) = h^*(0)$. Moreover, it induces that $d_{\tilde{\mathcal{F}}_{\mu_k}}(Id, T) = \mathcal{F}(\mu_k) - \mathcal{F}(\mu^*)$ and $d_{\tilde{\mathcal{F}}_{\mu_k}}(T_{k+1}, T) = \mathcal{F}(\mu_{k+1}) - \mathcal{F}(\mu^*)$.

Therefore, using (152) and the hypothesis $\alpha d_{\tilde{\mathcal{F}}_{\mu_k}}(Id, T) \leq d_{\phi_{\mu_k}^{h^*}}\big(0, \nabla_{W_2}\mathcal{F}(\mu_k)\big)$, we get

$$\begin{aligned}
\phi_{\mu_{k+1}}^{h^*}\big(\nabla_{W_2}\mathcal{F}(\mu_{k+1})\big) - h^*(0) &\leq \frac{1}{\tau}d_{\tilde{\mathcal{F}}_{\mu_k}}(Id, T) - \frac{1}{\tau}d_{\tilde{\mathcal{F}}_{\mu_k}}(T_{k+1}, T) - d_{\phi_{\mu_k}^{h^*}}\big(0, \nabla_{W_2}\mathcal{F}(\mu_k)\big) \\
&\leq \frac{1}{\tau}d_{\tilde{\mathcal{F}}_{\mu_k}}(Id, T) - \frac{1}{\tau}d_{\tilde{\mathcal{F}}_{\mu_k}}(T_{k+1}, T) - \alpha d_{\tilde{\mathcal{F}}_{\mu_k}}(Id, T) \\
&= \left(\frac{1}{\tau} - \alpha\right)d_{\tilde{\mathcal{F}}_{\mu_k}}(Id, T) - \frac{1}{\tau}d_{\tilde{\mathcal{F}}_{\mu_k}}(T_{k+1}, T) \\
&= \left(\frac{1}{\tau} - \alpha\right)\big(\mathcal{F}(\mu_k) - \mathcal{F}(\mu^*)\big) - \frac{1}{\tau}\big(\mathcal{F}(\mu_{k+1}) - \mathcal{F}(\mu^*)\big).
\end{aligned} \tag{154}$$

Then, applying Lemma 30 with $f = \phi_.^{h^*} \circ \nabla_{W_2}\mathcal{F}$ (which satisfies $\phi_{\mu_{k+1}}^{h^*}\big(\nabla_{W_2}\mathcal{F}(\mu_{k+1})\big) \leq \phi_{\mu_k}^{h^*}\big(\nabla_{W_2}\mathcal{F}(\mu_k)\big)$ by Proposition 5), $c = h^*(0)$ and $g = \mathcal{F}(\cdot) - \mathcal{F}(\mu^*) \geq 0$, we get

$$\phi_{\mu_k}^{h^*}\big(\nabla_{W_2}\mathcal{F}(\mu_k)\big) - h^*(0) \leq \frac{\alpha}{\left(\frac{\frac{1}{\tau}}{\frac{1}{\tau}-\alpha}\right)^k - 1}\big(\mathcal{F}(\mu_0) - \mathcal{F}(\mu^*)\big) \leq \frac{\frac{1}{\tau}-\alpha}{k}\big(\mathcal{F}(\mu_0) - \mathcal{F}(\mu^*)\big). \tag{155}$$

Concerning the convergence of $\mathcal{F}(\mu_k)$, if $\alpha > 0$ and $h^*$ attains its minimum in 0, then necessarily $\phi_\mu^{h^*}(T) \geq h^*(0)$ for all $\mu \in \mathcal{P}_2(\mathbb{R}^d)$ and $T \in L^2(\mu)$. Thus, using (152), we get

$$\begin{aligned}
0 \leq \phi_{\mu_{k+1}}^{h^*}\big(\nabla_{W_2}\mathcal{F}(\mu_{k+1})\big) - h^*(0) &\leq \frac{1}{\tau}d_{\tilde{\mathcal{F}}_{\mu_k}}(Id, T) - \frac{1}{\tau}d_{\tilde{\mathcal{F}}_{\mu_k}}(T_{k+1}, T) - d_{\phi_{\mu_k}^{h^*}}\big(0, \nabla_{W_2}\mathcal{F}(\mu_k)\big) \\
&\leq \frac{1}{\tau}\big(\mathcal{F}(\mu_k) - \mathcal{F}(\mu^*)\big) - \frac{1}{\tau}\big(\mathcal{F}(\mu_{k+1}) - \mathcal{F}(\mu^*)\big) \\
&\quad - \alpha d_{\tilde{\mathcal{F}}_{\mu_k}}(Id, T) \\
&= \left(\frac{1}{\tau} - \alpha\right)\big(\mathcal{F}(\mu_k) - \mathcal{F}(\mu^*)\big) - \frac{1}{\tau}\big(\mathcal{F}(\mu_{k+1}) - \mathcal{F}(\mu^*)\big).
\end{aligned} \tag{156}$$

Thus, for all $k \geq 0$,

$$\begin{aligned}
\mathcal{F}(\mu_{k+1}) - \mathcal{F}(\mu^*) &= (1 - \tau\alpha)\big(\mathcal{F}(\mu_k) - \mathcal{F}(\mu^*)\big) \\
&\leq (1 - \tau\alpha)^{k+1}\big(\mathcal{F}(\mu_0) - \mathcal{F}(\mu^*)\big).
\end{aligned} \tag{157}$$

$\square$

## H.7 Proof of Proposition 7

Let $\mu \in \mathcal{P}_2(\mathbb{R}^d)$. Since $\tilde{\mathcal{F}}_\mu^*$ is Gâteaux differentiable, we can define its Bregman divergence.

For the first point, $\phi^{h^*}$ is $\beta$-smooth relative to $\mathcal{F}_\mu^*$ along $t \mapsto \big((1-t)\nabla_{W_2}\mathcal{F}(\mu) + t\nabla_{W_2}\mathcal{F}(T_\#\mu) \circ T\big)_\# \mu$. Thus, by applying Definition 3 for $s = 1$ and $t = 0$, we have

$$d_{\phi_\mu^{h^*}}\big(\nabla_{W_2}\mathcal{F}(T_\#\mu) \circ T, \nabla_{W_2}\mathcal{F}(\mu)\big) \leq \beta d_{\tilde{\mathcal{F}}_\mu^*}\big(\nabla_{W_2}\mathcal{F}(T_\#\mu) \circ T, \nabla_{W_2}\mathcal{F}(\mu)\big). \tag{158}$$

Using Lemma 19, we finally obtain

$$d_{\phi_\mu^{h^*}}\big(\nabla_{W_2}\mathcal{F}(T_{\#}\mu)\circ T, \nabla_{W_2}\mathcal{F}(\mu_k)\big) \le \beta d_{\tilde{\mathcal{F}}_\mu^*}\big(\nabla_{W_2}\mathcal{F}(T_{\#}\mu)\circ T, \nabla_{W_2}\mathcal{F}(\mu)\big)$$
$$= \beta d_{\tilde{\mathcal{F}}_\mu}\big(\mathrm{Id}, T\big), \tag{159}$$

which is the desired inequality.

The second point follows similarly. $\qquad\square$

## H.8  Proof of Lemma 11

Let us define $\tilde{\mathcal{G}}_\mu : L^2(\mu) \times \mathbb{R}^d \to \mathbb{R}^d$ as for all $T \in L^2(\mu)$, $x \in \mathbb{R}^d$,

$$\tilde{\mathcal{G}}_\mu(T, x) = \nabla_{W_2}\mathcal{F}(T_{\#}\mu)(x) = \begin{pmatrix} \frac{\partial}{\partial x_1}\frac{\delta\mathcal{F}}{\delta\mu}(T_{\#}\mu)(x) \\ \vdots \\ \frac{\partial}{\partial x_d}\frac{\delta\mathcal{F}}{\delta\mu}(T_{\#}\mu)(x) \end{pmatrix} = \begin{pmatrix} \tilde{G}_\mu^1(T, x) \\ \vdots \\ \tilde{G}_\mu^d(T, x) \end{pmatrix}, \tag{160}$$

with for all $i$, $\tilde{G}_\mu^i : L^2(\mu) \times \mathbb{R}^d \to \mathbb{R}$, $\tilde{G}_\mu^i(T, x) = \frac{\partial}{\partial x_i}\frac{\delta\mathcal{F}}{\delta\mu}(T_{\#}\mu)(x)$. Using the chain rule, for all $x \in \mathbb{R}^d$,

$$\frac{d\tilde{G}_\mu^i}{ds}\big(T_s, T_s(x)\big) = \left\langle \nabla_1 \tilde{G}_\mu^i\big(T_s, T_s(x)\big), \frac{dT_s}{ds} \right\rangle_{L^2(\mu)} + \left\langle \nabla_2 \tilde{G}_\mu^i\big(T_s, T_s(x)\big), \frac{dT_s}{ds}(x) \right\rangle. \tag{161}$$

On one hand, we have $\nabla_2 \tilde{G}_\mu^i\big(T_s, T_s(x)\big) = \nabla\frac{\partial}{\partial x_i}\frac{\delta\mathcal{F}}{\delta\mu}\big((T_s)_{\#}\mu\big)\big(T_s(x)\big)$. On the other hand, let us compute $\nabla_1 \tilde{G}_\mu^i(T, x)$. First, we define the shorthands $\tilde{g}_\mu^{x,i}(T) = \tilde{G}_\mu^i(T, x) = \frac{\partial}{\partial x_i}\frac{\delta\mathcal{F}}{\delta\mu}(T_{\#}\mu)(x)$ and $g^{x,i}(\nu) = \frac{\partial}{\partial x_i}\frac{\delta\mathcal{F}}{\delta\mu}(\nu)(x)$. Since $\tilde{g}_\mu^{x,i}(T) = g^{x,i}(T_{\#}\mu)$, applying Proposition 1, we know that $\nabla_1 \tilde{G}_\mu(T, x) = \nabla \tilde{g}_\mu^{x,i}(T) = \nabla_{W_2} g^{x,i}(T_{\#}\mu)\circ T$.

Now, let us compute $\nabla_{W_2} g^{x,i}(\nu) = \nabla\frac{\delta g^{x,i}}{\delta\mu}(\nu)$. Let $\chi$ be such that $\int d\chi = 0$, then using the hypothesis that $\frac{\delta}{\delta\mu}\nabla\frac{\delta\mathcal{F}}{\delta\mu} = \nabla\frac{\delta^2\mathcal{F}}{\delta\mu^2}$ and the definition of $g^{x,i}$,

$$\int \frac{\delta g^{x,i}}{\delta\mu}(\nu)\, d\chi = \int \frac{\partial}{\partial x_i}\frac{\delta^2\mathcal{F}}{\delta\mu^2}(\nu)(x, y)\, d\chi(y). \tag{162}$$

Thus, $\nabla_{W_2} g^{x,i}(\nu) = \nabla_y \frac{\partial}{\partial x_i}\frac{\delta^2\mathcal{F}}{\delta\mu^2}(\nu)(x, y)$.

Putting everything together, we obtain

$$\frac{d\tilde{G}_\mu^i}{ds}\big(T_s, T_s(x)\big) = \left\langle \nabla_y \frac{\partial}{\partial x_i}\frac{\delta^2\mathcal{F}}{\delta\mu^2}\big((T_s)_{\#}\mu\big)\big(T_s(x), T_s(\cdot)\big), \frac{dT_s}{ds} \right\rangle_{L^2(\mu)}$$
$$+ \left\langle \nabla\frac{\partial}{\partial x_i}\frac{\delta\mathcal{F}}{\delta\mu}\big((T_s)_{\#}\mu\big)\big(T_s(x)\big), \frac{dT_s}{ds}(x) \right\rangle$$
$$= \int \left\langle \nabla_y \frac{\partial}{\partial x_i}\frac{\delta^2\mathcal{F}}{\delta\mu^2}\big((T_s)_{\#}\mu\big)\big(T_s(x), T_s(y)\big), \frac{dT_s}{ds}(y) \right\rangle d\mu(y) \tag{163}$$
$$+ \left\langle \nabla\frac{\partial}{\partial x_i}\frac{\delta\mathcal{F}}{\delta\mu}\big((T_s)_{\#}\mu\big)\big(T_s(x)\big), \frac{dT_s}{ds}(x) \right\rangle,$$

and thus

$$\frac{d}{ds}\tilde{\mathcal{G}}_\mu\big(T_s, T_s(x)\big) = \int \nabla_y\nabla_x \frac{\delta^2\mathcal{F}}{\delta\mu^2}\big((T_s)_{\#}\mu\big)\big(T_s(x), T_s(y)\big)\frac{dT_s}{ds}(y)\, d\mu(y)$$
$$+ \nabla^2\frac{\delta\mathcal{F}}{\delta\mu}\big((T_s)_{\#}\mu\big)\big(T_s(x)\big)\frac{dT_s}{ds}(x). \tag{164}$$

$\square$

## H.9  Proof of Proposition 12

First, recall that by using the chain rule and Proposition 1, $\frac{\mathrm{d}}{\mathrm{d}t}\mathcal{F}(\mu_t) = \langle \nabla_{W_2}\mathcal{F}(\mu_t) \circ T_t, \frac{\mathrm{d}T_t}{\mathrm{d}t}\rangle_{L^2(\mu)}$. Thus, since $\frac{\mathrm{d}^2 T_t}{\mathrm{d}t^2} = 0$,

$$
\begin{aligned}
\frac{\mathrm{d}^2}{\mathrm{d}t^2}\mathcal{F}(\mu_t) &= \frac{\mathrm{d}}{\mathrm{d}t}\left\langle \nabla_{W_2}\mathcal{F}(\mu_t) \circ T_t, \frac{\mathrm{d}T_t}{\mathrm{d}t}\right\rangle_{L^2(\mu)} \\
&= \left\langle \frac{\mathrm{d}}{\mathrm{d}t}\left(\nabla_{W_2}\mathcal{F}(\mu_t) \circ T_t\right), \frac{\mathrm{d}T_t}{\mathrm{d}t}\right\rangle_{L^2(\mu)}.
\end{aligned}
\tag{165}
$$

By Lemma 11,

$$
\begin{aligned}
\frac{\mathrm{d}^2}{\mathrm{d}t^2}\mathcal{F}(\mu_t) &= \iint \left\langle \nabla_y \nabla_x \frac{\delta^2 \mathcal{F}}{\delta \mu^2}\big((T_t)_{\#}\mu\big)\big(T_t(x), T_t(y)\big)\frac{\mathrm{d}T_t}{\mathrm{d}t}(y), \frac{\mathrm{d}T_t}{\mathrm{d}t}(x)\right\rangle \, \mathrm{d}\mu(y)\mathrm{d}\mu(x) \\
&\quad + \int \left\langle \nabla^2 \frac{\delta \mathcal{F}}{\delta \mu}\big((T_t)_{\#}\mu\big)\big(T_t(x)\big)\frac{\mathrm{d}T_t}{\mathrm{d}t}(x), \frac{\mathrm{d}T_t}{\mathrm{d}t}(x)\right\rangle \, \mathrm{d}\mu(x) \\
&= \iint \left\langle \nabla_y \nabla_x \frac{\delta^2 \mathcal{F}}{\delta \mu^2}\big((T_t)_{\#}\mu\big)\big(T_t(x), T_t(y)\big)v(y), v(x)\right\rangle \, \mathrm{d}\mu(y)\mathrm{d}\mu(x) \\
&\quad + \int \left\langle \nabla^2 \frac{\delta \mathcal{F}}{\delta \mu}\big((T_t)_{\#}\mu\big)\big(T_t(x)\big)v(x), v(x)\right\rangle \, \mathrm{d}\mu(x) \\
&= \int \left\langle \int \nabla_y \nabla_x \frac{\delta^2 \mathcal{F}}{\delta \mu^2}\big((T_t)_{\#}\mu\big)\big(T_t(x), T_t(y)\big)v(y) \, \mathrm{d}\mu(y)\right. \\
&\quad \left. + \nabla^2 \frac{\delta \mathcal{F}}{\delta \mu}\big((T_t)_{\#}\mu\big)\big(T_t(x)\big)v(x), v(x)\right\rangle \, \mathrm{d}\mu(x).
\end{aligned}
\tag{166}
$$

$\square$

## H.10  Proof of Proposition 13

1. **(c1)** $\implies$ **(c2)**. Let $t > 0$, $t_1, t_2 \in [0,1]$,

$$
\begin{aligned}
\mathcal{F}(\tilde{\mu}_t^{t_1 \to t_2}) &\leq (1-t)\mathcal{F}\big((T_{t_1})_{\#}\mu\big) + t\mathcal{F}\big((T_{t_2})_{\#}\mu\big) \\
&\Longleftrightarrow \frac{\mathcal{F}(\tilde{\mu}_t^{t_1 \to t_2}) - \mathcal{F}\big((T_{t_1})_{\#}\mu\big)}{t} \leq \mathcal{F}\big((T_{t_2})_{\#}\mu\big) - \mathcal{F}\big((T_{t_1})_{\#}\mu\big). \quad \text{(167)}
\end{aligned}
$$

   Passing to the limit $t \to 0$ and using Proposition 1, we get $\langle \nabla_{W_2}\mathcal{F}\big((T_{t_1})_{\#}\mu\big) \circ T_{t_1}, T_{t_2} - T_{t_1}\rangle_{L^2(\mu)} \leq \mathcal{F}\big((T_{t_2})_{\#}\mu\big) - \mathcal{F}\big((T_{t_1})_{\#}\mu\big)$.

2. **(c2)** $\implies$ **(c3)**. Let $t_1, t_2 \in [0,1]$, then by hypothesis,

$$
\begin{cases}
\langle \nabla_{W_2}\mathcal{F}\big((T_{t_1})_{\#}\mu\big) \circ T_{t_1}, T_{t_2} - T_{t_1}\rangle_{L^2(\mu)} \leq \mathcal{F}\big((T_{t_2})_{\#}\mu\big) - \mathcal{F}\big((T_{t_1})_{\#}\mu\big) \\
\langle \nabla_{W_2}\mathcal{F}\big((T_{t_2})_{\#}\mu\big) \circ T_{t_2}, T_{t_1} - T_{t_2}\rangle_{L^2(\mu)} \leq \mathcal{F}\big((T_{t_1})_{\#}\mu\big) - \mathcal{F}\big((T_{t_2})_{\#}\mu\big).
\end{cases}
\tag{168}
$$

   Summing the two inequalities, we get

$$
\langle \nabla_{W_2}\mathcal{F}\big((T_{t_2})_{\#}\mu\big) \circ T_{t_2} - \nabla_{W_2}\mathcal{F}\big((T_{t_1})_{\#}\mu\big) \circ T_{t_1}, T_{t_2} - T_{t_1}\rangle_{L^2(\mu)} \geq 0.
\tag{169}
$$

3. **(c3)** $\implies$ **(c4)**. Let $t_1, t_2 \in [0,1]$. First, we have,

$$
\begin{aligned}
\int_0^1 \frac{\mathrm{d}^2}{\mathrm{d}t^2}\mathcal{F}(\tilde{\mu}_t^{t_1 \to t_2}) \, \mathrm{d}t &= \frac{\mathrm{d}}{\mathrm{d}t}\mathcal{F}(\tilde{\mu}_t^{t_1 \to t_2})\Big|_{t=1} - \frac{\mathrm{d}}{\mathrm{d}t}\mathcal{F}(\tilde{\mu}_t^{t_1 \to t_2})\Big|_{t=0} \\
&= \langle \nabla_{W_2}\mathcal{F}\big((T_{t_2})_{\#}\mu\big) \circ T_{t_2} - \nabla_{W_2}\mathcal{F}\big((T_{t_1})_{\#}\mu\big) \circ T_{t_1}, \quad \text{(170)} \\
&\quad\quad T_{t_2} - T_{t_1}\rangle_{L^2(\mu)} \\
&\geq 0.
\end{aligned}
$$

   Let $\epsilon \in (0,1)$ and define $t \mapsto \nu_t^\epsilon = \tilde{\mu}_{\epsilon t}^{t_1 \to 1}$ the interpolation curve between $(T_{t_1})_{\#}\mu$ and $\big(T_{t_1} + \epsilon(T - T_{t_1})\big)_{\#}\mu$. Then, noting that $T_{t_1} + \epsilon(T - T_{t_1}) = T_{t_1 + \epsilon(1-t_1)}$, so

$\nu_t^\epsilon = \tilde{\mu}_{\epsilon t}^{t_1 \to 1} = \tilde{\mu}_t^{t_1 \to t_1 + \epsilon(1-t_1)}$ and we have that

$$\int_0^1 \frac{\mathrm{d}^2}{\mathrm{d}t^2} \mathcal{F}(\nu_t^\epsilon)\, \mathrm{d}t \geq 0. \tag{171}$$

Moreover, by continuity, $\frac{\mathrm{d}^2}{\mathrm{d}t^2}\mathcal{F}(\nu_t^\epsilon) \xrightarrow[\epsilon \to 0]{} \frac{\mathrm{d}^2}{\mathrm{d}t^2}\mathcal{F}\big((\mathrm{T}_{t_1})_\# \mu\big) = \frac{\mathrm{d}^2}{\mathrm{d}t^2}\mathcal{F}(\mu_{t_1})$. Then, since $t \mapsto \frac{\mathrm{d}^2}{\mathrm{d}t^2}\mathcal{F}(\nu_t^\epsilon)$ is continuous on $[0,1]$, it is bounded, and we can apply the dominated convergence theorem. This implies that for all $t_1 \in [0,1]$,

$$\mathrm{Hess}_{\mu_{t_1}} \mathcal{F} = \frac{\mathrm{d}^2}{\mathrm{d}t^2}\mathcal{F}(\mu_t)\Big|_{t=t_1} = \lim_{\epsilon \to 0} \int_0^1 \frac{\mathrm{d}^2}{\mathrm{d}t^2}\mathcal{F}(\nu_t^\epsilon)\, \mathrm{d}t \geq 0. \tag{172}$$

4. **(c4)** $\implies$ **(c1)**. Let $t_1, t_2 \in [0,1]$ and $\varphi(t) = \mathcal{F}(\tilde{\mu}_t^{t_1 \to t_2})$ for all $t \in [0,1]$. From [125, Equation 16.5],

$$\forall t \in [0,1], \ \varphi(t) = (1-t)\varphi(0) + t\varphi(1) - \int_0^1 \frac{\mathrm{d}^2}{\mathrm{d}t^2}\varphi(s)G(s,t)\,\mathrm{d}s, \tag{173}$$

where $G$ is the Green function defined as $G(s,t) = s(1-t)\mathbb{1}_{\{s \leq t\}} + t(1-s)\mathbb{1}_{\{t \leq s\}} \geq 0$ [125, Equation 16.6]. Then, $\frac{\mathrm{d}^2}{\mathrm{d}t^2}\mathcal{F}(\mu_t) \geq 0$ implies that $\int_0^1 \frac{\mathrm{d}^2}{\mathrm{d}t^2}\varphi(s)G(s,t)\,\mathrm{d}s \geq 0$, and thus

$$\varphi(t) = \mathcal{F}(\tilde{\mu}_t^{t_1 \to t_2}) \leq (1-t)\varphi(0) + t\varphi(1) = (1-t)\mathcal{F}\big((\mathrm{T}_{t_1})_\#\mu\big) + t\mathcal{F}\big((\mathrm{T}_{t_2})_\#\mu\big). \tag{174}$$

$\square$

## H.11  Proof of Proposition 24

Let $\mathrm{J}(\mathrm{T}) = \mathrm{d}_{\phi_{\mu_k}}(\mathrm{T}, \mathrm{Id}) + \tau\big(\langle \nabla_{\mathrm{W}_2}\mathcal{G}(\mu_k), \mathrm{T} - \mathrm{Id}\rangle_{L^2(\mu_k)} + \mathcal{H}(\mathrm{T}_\#\mu_k)\big)$. Taking the first variation, we get

$$\begin{aligned}
\nabla \mathrm{J}(\tilde{\mathrm{T}}_{k+1}) &= \nabla\phi_{\mu_k}(\tilde{\mathrm{T}}_{k+1}) - \nabla\phi_{\mu_k}(\mathrm{Id}) + \tau\big(\nabla_{\mathrm{W}_2}\mathcal{G}(\mu_k) + \nabla_{\mathrm{W}_2}\mathcal{H}\big((\tilde{\mathrm{T}}_{k+1})_\#\mu_k\big) \circ \tilde{\mathrm{T}}_{k+1}\big) \\
&= \nabla\phi_{\mu_k}(\tilde{\mathrm{T}}_{k+1}) + \tau\nabla_{\mathrm{W}_2}\mathcal{H}\big((\tilde{\mathrm{T}}_{k+1})_\#\mu_k\big) \circ \tilde{\mathrm{T}}_{k+1} - \big(\nabla\phi_{\mu_k}(\mathrm{Id}) - \tau\nabla_{\mathrm{W}_2}\mathcal{G}(\mu_k)\big) \\
&= \nabla\phi_{\mu_k}(\tilde{\mathrm{T}}_{k+1}) + \tau\nabla_{\mathrm{W}_2}\mathcal{H}\big((\tilde{\mathrm{T}}_{k+1})_\#\mu_k\big) \circ \tilde{\mathrm{T}}_{k+1} - \nabla\phi_{\mu_k}(\mathrm{S}_{k+1}). \tag{175}
\end{aligned}$$

Thus,

$$\nabla \mathrm{J}(\tilde{\mathrm{T}}_{k+1}) = 0 \iff \tilde{\mathrm{T}}_{k+1} \in \underset{\mathrm{T} \in L^2(\mu_k)}{\mathrm{argmin}} \ \mathrm{d}_{\phi_{\mu_k}}(\mathrm{T}, \mathrm{S}_{k+1}) + \tau\mathcal{H}(\mathrm{T}_\#\mu_k). \tag{176}$$

Now, we aim at showing that $\tilde{\mathrm{T}}_{k+1} = \mathrm{T}_{k+1} \circ \mathrm{S}_{k+1}$ or

$$\min_{\mathrm{T} \in L^2(\mu_k)} \ \mathrm{d}_{\phi_{\mu_k}}(\mathrm{T}, \mathrm{S}_{k+1}) + \tau\mathcal{H}(\mathrm{T}_\#\mu_k) = \min_{\mathrm{T} \in L^2(\nu_{k+1})} \ \mathrm{d}_{\phi_{\nu_{k+1}}}(\mathrm{T}, \mathrm{Id}) + \tau\mathcal{H}(\mathrm{T}_\#\nu_{k+1}). \tag{177}$$

First, by the change of variable formula, since $\phi_\mu$ is pushforward compatible, observe that for $\mathrm{T} \in L^2(\nu_{k+1})$, $\mathrm{d}_{\phi_{\nu_{k+1}}}(\mathrm{T}, \mathrm{Id}) + \tau\mathcal{H}(\mathrm{T}_\#\nu_{k+1}) = \mathrm{d}_{\phi_{\mu_k}}(\mathrm{T} \circ \mathrm{S}_{k+1}, \mathrm{S}_{k+1}) + \tau\mathcal{H}\big((\mathrm{T} \circ \mathrm{S}_{k+1})_\#\mu_k\big)$.

Since $\{\mathrm{T} \circ \mathrm{S}_{k+1} \mid \mathrm{T} \in L^2(\nu_{k+1})\} \subset L^2(\mu_k)$, we have

$$\min_{\mathrm{T} \in L^2(\nu_{k+1})} \ \mathrm{d}_{\phi_{\nu_{k+1}}}(\mathrm{T}, \mathrm{Id}) + \tau\mathcal{H}(\mathrm{T}_\#\nu_{k+1}) \geq \min_{\mathrm{T} \in L^2(\mu_k)} \ \mathrm{d}_{\phi_{\mu_k}}(\mathrm{T}, \mathrm{S}_{k+1}) + \tau\mathcal{H}(\mathrm{T}_\#\mu_k). \tag{178}$$

By assumption, $\nu_{k+1} \in \mathcal{P}_{2,\mathrm{ac}}(\mathbb{R}^d)$. Thus, applying Proposition 15, there exists $\mathrm{T}_{\phi_{\nu_{k+1}}}^{\nu_{k+1},\mu_{k+1}}$ such that $(\mathrm{T}_{\phi_{\nu_{k+1}}}^{\nu_{k+1},\mu_{k+1}})_\#\nu_{k+1} = \mu_{k+1}$ and $\mathrm{T}_{\phi_{\nu_{k+1}}}^{\nu_{k+1},\mu_{k+1}} = \mathrm{argmin}_{\mathrm{T}, \mathrm{T}_\#\nu_{k+1}=\mu_{k+1}} \ \mathrm{d}_{\phi_{\nu_{k+1}}}(\mathrm{T}, \mathrm{Id})$, and thus $\mathrm{d}_{\phi_{\nu_{k+1}}}(\mathrm{T}_{\phi_{\nu_{k+1}}}^{\nu_{k+1},\mu_{k+1}}, \mathrm{Id}) = \mathrm{W}_\phi(\mu_{k+1}, \nu_{k+1})$.

By contradiction, we suppose that

$$\min_{\mathrm{T} \in L^2(\nu_{k+1})} \ \mathrm{d}_{\phi_{\nu_{k+1}}}(\mathrm{T}, \mathrm{Id}) + \tau\mathcal{H}(\mathrm{T}_\#\nu_{k+1}) > \mathrm{d}_{\phi_{\mu_k}}(\tilde{\mathrm{T}}_{k+1}, \mathrm{S}_{k+1}) + \tau\mathcal{H}\big((\tilde{\mathrm{T}}_{k+1})_\#\mu_k\big). \tag{179}$$

On one hand, we have $(\mathrm{T}_{\phi_{\nu_{k+1}}}^{\nu_{k+1},\mu_{k+1}} \circ \mathrm{S}_{k+1})_\#\mu_k = (\mathrm{T}_{\phi_{\nu_{k+1}}}^{\nu_{k+1},\mu_{k+1}})_\#\nu_{k+1} = \mu_{k+1}$, and therefore $\mathcal{H}\big((\mathrm{T}_{\phi_{\nu_{k+1}}}^{\nu_{k+1},\mu_{k+1}} \circ \mathrm{S}_{k+1})_\#\mu_k\big) = \mathcal{H}(\mu_{k+1}) = \mathcal{H}\big((\tilde{\mathrm{T}}_{k+1})_\#\mu_k\big)$. On the other hand, $(\tilde{\mathrm{T}}_{k+1}, \mathrm{S}_{k+1})_\#\mu_k \in \Pi(\mu_{k+1}, \nu_{k+1})$, and thus

$$\mathrm{d}_{\phi_{\mu_k}}(\tilde{\mathrm{T}}_{k+1}, \mathrm{S}_{k+1}) \geq \mathrm{W}_\phi(\mu_{k+1}, \nu_{k+1}) = \mathrm{d}_{\phi_{\nu_{k+1}}}(\mathrm{T}_{\phi_{\nu_{k+1}}}^{\nu_{k+1},\mu_{k+1}}, \mathrm{Id}). \tag{180}$$

Thus,

$$\min_{T \in L^2(\nu_{k+1})} d_{\phi_{\nu_{k+1}}}(T, \mathrm{Id}) + \tau \mathcal{H}(T_{\#}\nu_{k+1}) > d_{\phi_{\mu_k}}(\tilde{T}_{k+1}, S_{k+1}) + \tau \mathcal{H}\big((\tilde{T}_{k+1})_{\#}\mu_k\big) \tag{181}$$

$$\geq d_{\phi_{\nu_{k+1}}}(T^{\nu_{k+1}, \mu_{k+1}}_{\phi_{\nu_{k+1}}}, \mathrm{Id}) + \tau \mathcal{H}\big((T^{\nu_{k+1}, \mu_{k+1}}_{\phi_{\nu_{k+1}}})_{\#}\nu_{k+1}\big).$$

But $T^{\nu_{k+1}, \mu_{k+1}}_{\phi_{\nu_{k+1}}} \in L^2(\nu_{k+1})$, so this is a contradiction. So, we can conclude that the two schemes are equivalent, and moreover, $\tilde{T}_{k+1} = T^{\nu_{k+1}, \mu_{k+1}}_{\phi_{\nu_{k+1}}} \circ S_{k+1}$. $\qquad \square$

## H.12 Proof of Proposition 25

Let $\psi(T) = \tau \big( \langle \nabla_{W_2} \mathcal{G}(\mu_k), T - \mathrm{Id} \rangle_{L^2(\mu_k)} + \mathcal{H}(T_{\#}\mu_k) \big)$. Since $\tilde{\mathcal{H}}_{\mu_k}$ is convex on $L^2(\mu_k)$, $\psi$ is convex, and we can apply the three-point inequality (Lemma 29) and for all $T \in L^2(\mu_k)$,

$$\tau \big( \mathcal{H}(T_{\#}\mu_k) + \langle \nabla_{W_2} \mathcal{G}(\mu_k), T - \mathrm{Id} \rangle_{L^2(\mu_k)} \big) + d_{\phi_{\mu_k}}(T, \mathrm{Id})$$
$$\geq \tau \big( \mathcal{H}(\mu_{k+1}) + \langle \nabla_{W_2} \mathcal{G}(\mu_k), \tilde{T}_{k+1} - \mathrm{Id} \rangle_{L^2(\mu_k)} \big) + d_{\phi_{\mu_k}}(\tilde{T}_{k+1}, \mathrm{Id}) + d_{\phi_{\mu_k}}(T, \tilde{T}_{k+1}), \tag{182}$$

which is equivalent to

$$\mathcal{H}(\mu_{k+1}) + \langle \nabla_{W_2} \mathcal{G}(\mu_k), \tilde{T}_{k+1} - \mathrm{Id} \rangle_{L^2(\mu_k)} + \frac{1}{\tau} d_{\phi_\mu}(\tilde{T}_{k+1}, \mathrm{Id})$$
$$\leq \mathcal{H}(T_{\#}\mu_k) + \langle \nabla_{W_2} \mathcal{G}(\mu_k), T - \mathrm{Id} \rangle_{L^2(\mu_k)} + \frac{1}{\tau} d_{\phi_{\mu_k}}(T, \mathrm{Id}) - \frac{1}{\tau} d_{\phi_{\mu_k}}(T, \tilde{T}_{k+1}). \tag{183}$$

Since $\tilde{\mathcal{G}}_{\mu_k}$ is $\beta$-smooth relatively to $\phi_{\mu_k}$ along $t \mapsto \big((1-t)\mathrm{Id} + t\tilde{T}_{k+1}\big)_{\#}\mu_k$, and $\tau \leq \frac{1}{\beta}$, we also have

$$\mathcal{G}(\mu_{k+1}) \leq \mathcal{G}(\mu_k) + \langle \nabla_{W_2} \mathcal{G}(\mu_k), \tilde{T}_{k+1} - \mathrm{Id} \rangle_{L^2(\mu_k)} + \beta d_{\phi_{\mu_k}}(\tilde{T}_{k+1}, \mathrm{Id})$$
$$\leq \mathcal{G}(\mu_k) + \langle \nabla_{W_2} \mathcal{G}(\mu_k), \tilde{T}_{k+1} - \mathrm{Id} \rangle_{L^2(\mu_k)} + \frac{1}{\tau} d_{\phi_{\mu_k}}(\tilde{T}_{k+1}, \mathrm{Id}). \tag{184}$$

Thus, applying first the smoothness of $\mathcal{G}$ and then the three-point inequality, we get for all $T \in L^2(\mu_k)$,

$$\mathcal{H}(\mu_{k+1}) + \mathcal{G}(\mu_{k+1}) \leq \mathcal{H}(\mu_{k+1}) + \mathcal{G}(\mu_k) + \langle \nabla_{W_2} \mathcal{G}(\mu_k), \tilde{T}_{k+1} - \mathrm{Id} \rangle_{L^2(\mu_k)} + \frac{1}{\tau} d_{\phi_{\mu_k}}(\tilde{T}_{k+1}, \mathrm{Id})$$
$$\leq \mathcal{H}(T_{\#}\mu_k) + \mathcal{G}(\mu_k) + \langle \nabla_{W_2} \mathcal{G}(\mu_k), T - \mathrm{Id} \rangle_{L^2(\mu_k)} + \frac{1}{\tau} d_{\phi_{\mu_k}}(T, \mathrm{Id})$$
$$- \frac{1}{\tau} d_{\phi_{\mu_k}}(T, \tilde{T}_{k+1}). \tag{185}$$

Now, let $\nu \in \mathcal{P}_2(\mathbb{R}^d)$ and $T^{\mu_k, \nu}_{\phi_{\mu_k}} = \mathrm{argmin}_{T, T_{\#}\mu_k = \nu} \, d_{\phi_{\mu_k}}(T, \mathrm{Id})$, and suppose that $\tilde{\mathcal{G}}_{\mu_k}$ is $\alpha$-convex relative to $\phi_{\mu_k}$ along $t \mapsto \big((1-t)\mathrm{Id} + tT^{\mu_k, \nu}_{\phi_{\mu_k}}\big)_{\#}\mu_k$. Thus,

$$d_{\tilde{\mathcal{G}}_{\mu_k}}(T^{\mu_k, \nu}_{\phi_{\mu_k}}, \mathrm{Id}) \geq \alpha d_{\phi_{\mu_k}}(T^{\mu_k, \nu}_{\phi_{\mu_k}}, \mathrm{Id})$$
$$\iff \mathcal{G}(\nu) - \alpha d_{\phi_{\mu_k}}(T^{\mu_k, \nu}_{\phi_{\mu_k}}, \mathrm{Id}) \geq \mathcal{G}(\mu_k) + \langle \nabla_{W_2} \mathcal{G}(\mu_k), T^{\mu_k, \nu}_{\phi_{\mu_k}} - \mathrm{Id} \rangle_{L^2(\mu_k)}. \tag{186}$$

Plugging this into (185), we get

$$\mathcal{F}(\mu_{k+1}) \leq \mathcal{H}(\nu) + \mathcal{G}(\nu) - \alpha d_{\phi_{\mu_k}}(T^{\mu_k, \nu}_{\phi_{\mu_k}}, \mathrm{Id}) + \frac{1}{\tau} d_{\phi_{\mu_k}}(T^{\mu_k, \nu}_{\phi_{\mu_k}}, \mathrm{Id}) - \frac{1}{\tau} d_{\phi_{\mu_k}}(T^{\mu_k, \nu}_{\phi_{\mu_k}}, \tilde{T}_{k+1}). \tag{187}$$

Now, note that $d_{\phi_{\mu_k}}(T^{\mu_k, \nu}_{\phi_{\mu_k}}, \mathrm{Id}) = W_\phi(\nu, \mu_k)$ and by Assumption 1, $d_{\phi_{\mu_k}}(T^{\mu_k, \nu}_{\phi_{\mu_k}}, \tilde{T}_{k+1}) \geq W_\phi(\nu, \mu_{k+1})$. Thus,

$$\mathcal{F}(\mu_{k+1}) - \mathcal{F}(\nu) \leq \left( \frac{1}{\tau} - \alpha \right) W_\phi(\nu, \mu_k) - \frac{1}{\tau} W_\phi(\nu, \mu_{k+1}). \tag{188}$$

Using $T = \mathrm{Id}$ in (185), we observe that $\mathcal{F}(\mu_k) \leq \mathcal{F}(\mu_\ell)$ for all $\ell \leq k$. Moreover, $W_\phi(\nu, \mu_k) \geq 0$. Thus, applying Lemma 30 with $f = \mathcal{F}$, $c = \mathcal{F}(\nu)$ and $g = W_\phi(\nu, \cdot)$, we obtain

$$\forall k \geq 1, \ \mathcal{F}(\mu_k) - \mathcal{F}(\nu) \leq \frac{\alpha}{\left( \frac{\frac{1}{\tau}}{\frac{1}{\tau} - \alpha} \right)^k - 1} W_\phi(\nu, \mu_0) \leq \frac{\frac{1}{\tau} - \alpha}{k} W_\phi(\nu, \mu_0). \tag{189}$$
$\qquad \square$

## H.13 Proof of Lemma 26

First, $\nabla V^*$ is bijective. Thus, we only need to show that $h = \nabla V - \tau \nabla U$ is injective. Take $u = V - \tau U$.

Since $U$ is $\beta$-smooth relative to $V$, we have for all $x, y$,

$$U(x) \leq U(y) + \langle \nabla U(y), x - y \rangle + \beta d_V(x, y), \tag{190}$$

which is equivalent to

$$-U(y) \leq -U(x) + \langle \nabla U(y), x - y \rangle + \beta d_V(x, y). \tag{191}$$

Moreover, by definition of $d_V$,

$$V(y) = V(x) - \langle \nabla V(y), x - y \rangle - d_V(x, y). \tag{192}$$

Summing the two inequalities, we get

$$\begin{aligned} V(y) - \tau U(y) &\leq V(x) - \langle \nabla V(y), x - y \rangle - d_V(x, y) - \tau U(x) + \tau \langle \nabla U(y), x - y \rangle \\ &\quad + \tau \beta d_V(x, y) \\ &= V(x) - \tau U(x) - \langle \nabla V(y) - \tau \nabla U(y), x - y \rangle - (1 - \tau\beta) d_V(x, y). \end{aligned} \tag{193}$$

This is equivalent to

$$u(y) \leq u(x) - \langle \nabla u(y), x - y \rangle - (1 - \tau\beta) d_V(x, y), \tag{194}$$

and thus with $u$ being $(1 - \tau\beta)$-convex relative to $V$ (for $\tau\beta \leq 1$). For $\tau\beta < 1$, it is equivalent to $u - (1 - \tau\beta)V$ convex, *i.e.* $\langle \nabla u(x) - \nabla u(y), x - y \rangle \geq (1 - \tau\beta)\langle \nabla V(x) - \nabla V(y), x - y \rangle \geq 0$. Since $V$ is strictly convex, $\nabla u$ is injective.

Moreover, $|\det \nabla T| = |\det (\nabla^2 V^* \circ (\nabla V - \tau \nabla U)) \det \nabla^2 u| > 0$ because on one hand $u$ is $(1 - \beta\tau)$-convex relative to $V$ which is strictly convex, and on the other hand, $V^*$ is also strictly convex.

To conclude, applying [5, Lemma 5.5.3], $T_{\#}\mu$ is absolutely continuous with respect to the Lebesgue measure. $\qquad\square$

## H.14 Proof of Proposition 27

On one hand, $\mathcal{H}$ is 1-smooth relative to $\mathcal{H}$, thus we only need to show that $\mu \mapsto \int V d\mu$ is smooth relative to $\mathcal{H}$. Using Proposition 13, we need to show that

$$\frac{d^2}{dt^2}\mathcal{V}(\mu_t) = \frac{1}{2}\int (T_{k+1} - \mathrm{Id})^T \nabla^2 V (T_{k+1} - \mathrm{Id})\, d\mu_k \leq \beta \frac{d^2}{dt^2}\mathcal{H}(\mu_t). \tag{195}$$

Recall from (61) that $T_{k+1}(x) = ((1-\tau)\Sigma_{k+1}\Sigma_k^{-1} + \tau\Sigma_{k+1}\Sigma^{-1})x + cst$, thus $\nabla T_{k+1}$ is a constant. Using the computations of [43, Appendix B.2],

$$\frac{d^2}{dt^2}\mathcal{H}(\mu_t) = \langle [\nabla T_t]^{-2}, \nabla T_{k+1} - I_d \rangle. \tag{196}$$

Assuming $(1-\tau)\Sigma_{k+1}\Sigma_k^{-1} + \tau\Sigma_{k+1}\Sigma^{-1} \succeq 0$, $T_{k+1}$ is the gradient of a convex function and $\mu_t$ is a Wasserstein geodesic. Thus, by [43],

$$\frac{d^2}{dt^2}\mathcal{H}(\mu_t) \geq \frac{1}{\|\Sigma_{\mu_t}\|_{\mathrm{op}}}\|T_{k+1} - \mathrm{Id}\|^2_{L^2(\mu_k)}. \tag{197}$$

Moreover, by [33, Lemma 10], $\mu \mapsto \|\Sigma_\mu\|_{\mathrm{op}}$ is convex along generalized geodesics, and thus $\Sigma_{\mu_t} \preceq \lambda I_d$ and $\|\Sigma_{\mu_t}\|_{\mathrm{op}} \leq \lambda$ [43]. Hence, noting $\sigma_{\max}(M)$ the largest eigenvalue of some matrix $M$,

$$\begin{aligned} \frac{d^2}{dt^2}\mathcal{H}(\mu_t) \geq \frac{1}{\lambda}\|T_{k+1} - \mathrm{Id}\|^2_{L^2(\mu_k)} &\geq \frac{1}{\lambda\sigma_{\max}(\nabla^2 V)}\int (T_{k+1} - \mathrm{Id})^T\nabla^2 V(T_{k+1} - \mathrm{Id})d\mu_k \\ &= \frac{2}{\lambda\sigma_{\max}(\nabla^2 V)}\frac{d^2}{dt^2}\mathcal{V}(\mu_t). \end{aligned} \tag{198}$$

From this inequality, we deduce that

$$\frac{\lambda\sigma_{\max}(\nabla^2 V)}{2}d_{\tilde{\mathcal{H}}_{\mu_k}}(T_{k+1},\mathrm{Id}) = \frac{\lambda\sigma_{\max}(\nabla^2 V)}{2}\big(\mathcal{H}(\mu_{k+1}) - \mathcal{H}(\mu_k)$$
$$- \langle\nabla_{W_2}\mathcal{H}(\mu_k), T_{k+1} - \mathrm{Id}\rangle_{L^2(\mu_k)}\big)$$
$$= \frac{\lambda\sigma_{\max}(\nabla^2 V)}{2}\int(1-t)\frac{\mathrm{d}^2}{\mathrm{d}t^2}\mathcal{H}(\mu_t)\,\mathrm{d}t \qquad (199)$$
$$\geq \int\frac{\mathrm{d}^2}{\mathrm{d}t^2}\mathcal{V}(\mu_t)(1-t)\,\mathrm{d}t$$
$$= d_{\tilde{\mathcal{V}}_{\mu_k}}(T_{k+1},\mathrm{Id}).$$

So,

$$d_{\tilde{\mathcal{F}}_{\mu_k}}(T_{k+1},\mathrm{Id}) = d_{\tilde{\mathcal{V}}_{\mu_k}}(T_{k+1},\mathrm{Id}) + d_{\tilde{\mathcal{H}}_{\mu_k}}(T_{k+1},\mathrm{Id})$$
$$\leq \left(1 + \frac{\lambda\sigma_{\max}(\nabla^2 V)}{2}\right)d_{\tilde{\mathcal{H}}_{\mu_k}}(T_{k+1},\mathrm{Id}). \qquad (200)$$
$$\square$$

# I  Additional results

## I.1  Three-point identity and inequality

In this Section, we derive results which are useful to show the convergence of mirror descent or preconditioned schemes. Namely, we first derive the three-point identity which we use to show the convergence of the preconditioned scheme in Proposition 5 as well as the three-point inequality, which we use for the convergence of the mirror descent scheme in Proposition 3.

**Lemma 28** (Three-Point Identity). *Let $\phi : L^2(\mu) \to \mathbb{R}$ be Gâteaux differentiable. For all $S, T, U \in L^2(\mu)$, we have*

$$d_\phi(S, U) = d_\phi(S, T) + d_\phi(T, U) + \langle\nabla\phi(T), S - T\rangle_{L^2(\mu)} - \langle\nabla\phi(U), S - T\rangle_{L^2(\mu)}. \qquad (201)$$

*Proof.* Let $S, T, U \in L^2(\mu)$, then using the linearity of the Gâteaux differential,

$$d_\phi(S, U) - d_\phi(S, T) - d_\phi(T, U) = \phi(S) - \phi(U) - \langle\nabla\phi(U), S - U\rangle_{L^2(\mu)}$$
$$- \phi(S) + \phi(T) + \langle\nabla\phi(T), S - T\rangle_{L^2(\mu)}$$
$$- \phi(T) + \phi(U) + \langle\nabla\phi(U), T - U\rangle_{L^2(\mu)}$$
$$= -\langle\nabla\phi(U), S - U\rangle_{L^2(\mu)} + \langle\nabla\phi(T), S - T\rangle_{L^2(\mu)} \qquad (202)$$
$$+ \langle\nabla\phi(U), T - U\rangle_{L^2(\mu)}$$
$$= \langle\nabla\phi(T), S - T\rangle_{L^2(\mu)} - \langle\nabla\phi(U), S - T\rangle_{L^2(\mu)}.$$
$$\square$$

**Lemma 29** (Three-Point Inequality). *Let $\mu \in \mathcal{P}_2(\mathbb{R}^d)$, $T_0 \in L^2(\mu)$ and $\phi_\mu : L^2(\mu) \to \mathbb{R}$ convex, and Gâteaux differentiable. Let $\psi : L^2(\mu) \to \mathbb{R}$ be convex, proper and lower semicontinuous. Assume there exists $T^* = \mathrm{argmin}_{T\in L^2(\mu)}\, d_{\phi_\mu}(T, T_0) + \psi(T)$. Then, for all $T \in L^2(\mu)$,*

$$\psi(T) + d_{\phi_\mu}(T, T_0) \geq \psi(T^*) + d_{\phi_\mu}(T^*, T_0) + d_{\phi_\mu}(T, T^*). \qquad (203)$$

*Proof.* Denote $J(T) = d_{\phi_\mu}(T, T_0) + \psi(T)$. Let $T^* = \mathrm{argmin}_{T\in L^2(\mu)}\, J(T)$, hence $0 \in \partial J(T^*)$.

Since $\phi$ and $\psi$ are proper, convex and lower semicontinuous, and $T \mapsto d_{\phi_\mu}(T, T_0)$ is continuous (since $\phi_\mu$ is continuous), thus by [99, Theorem 3.30], $\partial J(T^*) = \partial\psi(T^*) + \partial d_{\phi_\mu}(\cdot, T_0)(T^*)$.

Moreover, since $\phi_\mu$ is differentiable, $\partial d_{\phi_\mu}(\cdot, T_0)(T^*) = \{\nabla_T d_{\phi_\mu}(T^*, T_0)\} = \{\nabla\phi_\mu(T^*) - \nabla\phi_\mu(T_0)\}$, and thus $\nabla\phi_\mu(T_0) - \nabla\phi_\mu(T^*) \in \partial\psi(T^*)$

Finally, by definition of subgradients and by applying Lemma 28, we get for all $T \in L^2(\mu)$,

$$\psi(T) \geq \psi(T^*) - \big(\langle\nabla\phi_\mu(T^*), T - T^*\rangle_{L^2(\mu)} - \langle\nabla\phi_\mu(T_0), T - T^*\rangle_{L^2(\mu)}\big)$$
$$= \psi(T^*) - d_{\phi_\mu}(T, T_0) + d_{\phi_\mu}(T, T^*) + d_{\phi_\mu}(T^*, T_0). \qquad (204)$$
$$\square$$

**Remark 3.** *Actually we can restrict $\psi$ to be convex along $\left((1-t)\mathrm{T}^* + t\mathrm{T}\right)_{\#}\mu$. In that case, $\mathrm{d}_\psi(\mathrm{T},\mathrm{T}^*) = \psi(\mathrm{T}) - \psi(\mathrm{T}^*) - \langle\varphi,\mathrm{T} - \mathrm{T}^*\rangle_{L^2(\mu)} \geq 0$ for $\varphi \in \partial\psi(\mathrm{T}^*)$ (by Proposition 13) and we still have $\partial\psi(\mathrm{T}^*) + \partial\mathrm{d}_{\phi_\mu}(\cdot,\mathrm{T}_0)(\mathrm{T}^*) \subset \partial\mathrm{J}(\mathrm{T}^*)$ (see [99, Theorem 3.30]) so that we can conclude.*

## I.2   Convergence lemma

We first provide a Lemma which follows from [88, Theorem 3.1], and which is useful for the proofs of Propositions 4, 6 and 25.

**Lemma 30.** *Let $f : X \to \mathbb{R}$, $g : X \to \mathbb{R}_+$ and $(x_k)_{k\in\mathbb{N}}$ a sequence in $X$ such that for all $k \geq 1$, $f(x_k) \leq f(x_{k-1})$. Assume that there exists $\beta > \alpha \geq 0$, $c \in \mathbb{R}$ such that for all $k \geq 0$, $f(x_{k+1}) - c \leq (\beta - \alpha)g(x_k) - \beta g(x_{k+1})$, then*

$$\forall k \geq 1, \ f(x_k) - c \leq \frac{\alpha}{\left(\frac{\beta}{\beta-\alpha}\right)^k - 1}g(x_0) \leq \frac{\beta - \alpha}{k}g(x_0). \tag{205}$$

*Proof.* First, observe the $f(x_k) \leq f(x_\ell)$ for all $\ell \leq k$. Thus, for all $k \geq 1$,

$$
\begin{aligned}
\sum_{\ell=1}^k \left(\frac{\beta}{\beta-\alpha}\right)^\ell \cdot \left(f(x_k) - c\right) &\leq \sum_{\ell=1}^k \left(\frac{\beta}{\beta-\alpha}\right)^\ell \left(f(x_\ell) - c)\right) \\
&\leq \sum_{\ell=1}^k \left(\frac{\beta}{\beta-\alpha}\right)^\ell \left((\beta-\alpha)g(x_{\ell-1}) - \beta g(x_\ell)\right) \\
&= \beta\sum_{\ell=0}^{k-1} \left(\frac{\beta}{\beta-\alpha}\right)^\ell g(x_\ell) - \beta\sum_{\ell=1}^k \left(\frac{\beta}{\beta-\alpha}\right)^\ell g(x_\ell) \\
&= \beta g(x_0) - \beta\left(\frac{\beta}{\beta-\alpha}\right)^k g(x_k) \\
&\leq \beta g(x_0) \quad \text{since } g \geq 0.
\end{aligned}
\tag{206}
$$

Now, note that $\frac{\beta}{\sum_{\ell=1}^k\left(\frac{\beta}{\beta-\alpha}\right)^\ell} = \frac{\alpha}{\left(\frac{\beta}{\beta-\alpha}\right)^k-1} = \frac{\alpha}{\left(1+\frac{\alpha}{\beta-\alpha}\right)^k-1} \leq \frac{\beta-\alpha}{k}$ since $\left(1+\frac{\alpha}{\beta-\alpha}\right)^k \geq 1 + k\frac{\alpha}{\beta-\alpha}$ (by convexity on $\mathbb{R}_+$ of $x \mapsto (1+x)^k$). Thus,

$$f(x_k) - c \leq \frac{\beta}{\sum_{\ell=1}^k\left(\frac{\beta}{\beta-\alpha}\right)^\ell}g(x_0) = \frac{\alpha}{\left(\frac{\beta}{\beta-\alpha}\right)^k - 1}g(x_0) \leq \frac{\beta-\alpha}{k}g(x_0). \tag{207}$$
$\qquad\square$

## I.3   Some properties of Bregman divergences

We provide in this Section additional results on the Bregman divergences introduced in Section 3. First, we focus on $\phi_\mu(\mathrm{T}) = \int V \circ \mathrm{T}\, \mathrm{d}\mu$. The following Lemma is akin to [80, Proposition 4] which shows it only for OT maps.

**Lemma 31.** *Let $V : \mathbb{R}^d \to \mathbb{R}$ convex and $\phi_\mu(\mathrm{T}) = \int V \circ \mathrm{T}\, \mathrm{d}\mu$. Then,*

$$\forall \mathrm{T},\mathrm{S} \in L^2(\mu), \ \mathrm{d}_{\phi_\mu}(\mathrm{T},\mathrm{S}) = \int \mathrm{d}_V\left(\mathrm{T}(x),\mathrm{S}(x)\right) \mathrm{d}\mu(x). \tag{208}$$

*Proof.* Let $\mathrm{T},\mathrm{S} \in L^2(\mu)$, then remembering that $\nabla_{\mathrm{W}_2}\mathcal{V}(\mu) = \nabla V$, we have

$$
\begin{aligned}
\mathrm{d}_{\phi_\mu}(\mathrm{T},\mathrm{S}) &= \phi_\mu(\mathrm{T}) - \phi_\mu(\mathrm{S}) - \langle\nabla V \circ \mathrm{S}, \mathrm{T} - \mathrm{S}\rangle_{L^2(\mu)} \\
&= \int V \circ \mathrm{T} - V \circ \mathrm{S} - \langle\nabla V \circ \mathrm{S}, \mathrm{T} - \mathrm{S}\rangle\, \mathrm{d}\mu \\
&= \int \mathrm{d}_V\left(\mathrm{T}(x),\mathrm{S}(x)\right) \mathrm{d}\mu(x).
\end{aligned}
\tag{209}
$$
$\qquad\square$

Next, we focus on $\phi_\mu(\mathrm{T}) = \frac{1}{2} \iint W\big(\mathrm{T}(x) - \mathrm{T}(x')\big) \, \mathrm{d}\mu(x)\mathrm{d}\mu(x')$, and we generalize the result from [80, Proposition 4].

**Lemma 32.** *Let $W : \mathbb{R}^d \to \mathbb{R}$ even ($W(x) = W(-x)$), convex and differentiable. Let $\phi_\mu(\mathrm{T}) = \frac{1}{2} \iint W\big(\mathrm{T}(x) - \mathrm{T}(x')\big) \, \mathrm{d}\mu(x)\mathrm{d}\mu(x')$. Then,*

$$\forall \mathrm{T}, \mathrm{S} \in L^2(\mu), \ \mathrm{d}_{\phi_\mu}(\mathrm{T}, \mathrm{S}) = \frac{1}{2} \iint \mathrm{d}_W\big(\mathrm{T}(x) - \mathrm{T}(x'), \mathrm{S}(x) - \mathrm{S}(x')\big) \, \mathrm{d}\mu(x)\mathrm{d}\mu(x'). \quad (210)$$

*Proof.* Let $\mathrm{T}, \mathrm{S} \in L^2(\mu)$, remember that $\nabla_{\mathrm{W}_2} \mathcal{W}(\mu) = \nabla W \star \mu$, and thus $\nabla_{\mathrm{W}_2} \mathcal{W}(\mathrm{S}_{\#}\mu) \circ \mathrm{S} = (\nabla W \star \mathrm{S}_{\#}\mu) \circ \mathrm{S}$. Thus,

$$\begin{aligned}
\mathrm{d}_{\phi_\mu}(\mathrm{T}, \mathrm{S}) &= \phi_\mu(\mathrm{T}) - \phi_\mu(\mathrm{S}) - \langle (\nabla W \star \mathrm{S}_{\#}\mu) \circ \mathrm{S}, \mathrm{T} - \mathrm{S} \rangle_{L^2(\mu)} \\
&= \frac{1}{2} \iint W\big(\mathrm{T}(x) - \mathrm{T}(x')\big) \, \mathrm{d}\mu(x)\mathrm{d}\mu(x') - \frac{1}{2} \iint W\big(\mathrm{S}(x) - \mathrm{S}(x')\big) \, \mathrm{d}\mu(x)\mathrm{d}\mu(x') \\
&\quad - \int \langle (\nabla W \star \mathrm{S}_{\#}\mu)(\mathrm{S}(x)), \mathrm{T}(x) - \mathrm{S}(x) \rangle \, \mathrm{d}\mu(x).
\end{aligned} \quad (211)$$

Then, note that $\nabla W(-x) = -\nabla W(x)$ and thus the last term can be written as:

$$\begin{aligned}
&\int \langle (\nabla W \star \mathrm{S}_{\#}\mu)(\mathrm{S}(x)), \mathrm{T}(x) - \mathrm{S}(x) \rangle \, \mathrm{d}\mu(x) \\
&= \iint \langle \nabla W\big(\mathrm{S}(x) - \mathrm{S}(x')\big), \mathrm{T}(x) - \mathrm{S}(x) \rangle \, \mathrm{d}\mu(x)\mathrm{d}\mu(x') \\
&= \frac{1}{2} \iint \langle \nabla W\big(\mathrm{S}(x) - \mathrm{S}(x')\big), \mathrm{T}(x) - \mathrm{S}(x) \rangle \, \mathrm{d}\mu(x)\mathrm{d}\mu(x') \\
&\quad + \frac{1}{2} \langle \nabla W\big(\mathrm{S}(x') - \mathrm{S}(x)\big), \mathrm{T}(y) - \mathrm{S}(y) \rangle \, \mathrm{d}\mu(x)\mathrm{d}\mu(x') \\
&= \frac{1}{2} \iint \langle \nabla W\big(\mathrm{S}(x) - \mathrm{S}(x')\big), \mathrm{T}(x) - \mathrm{S}(x) \rangle \, \mathrm{d}\mu(x)\mathrm{d}\mu(x') \\
&\quad - \frac{1}{2} \langle \nabla W\big(\mathrm{S}(x) - \mathrm{S}(x')\big), \mathrm{T}(x') - \mathrm{S}(x') \rangle \, \mathrm{d}\mu(x)\mathrm{d}\mu(x') \\
&= \frac{1}{2} \iint \langle \nabla W\big(\mathrm{S}(x) - \mathrm{S}(x')\big), \mathrm{T}(x) - \mathrm{T}(x') - \big(\mathrm{S}(x) - \mathrm{S}(x')\big) \rangle \, \mathrm{d}\mu(x)\mathrm{d}\mu(x').
\end{aligned} \quad (212)$$

Finally, we get

$$\begin{aligned}
\mathrm{d}_{\phi_\mu}(\mathrm{T}, \mathrm{S}) &= \frac{1}{2} \iint \Big( W\big(\mathrm{T}(x) - \mathrm{T}(x')\big) - W\big(\mathrm{S}(x) - \mathrm{S}(x')\big) \\
&\quad - \langle \nabla W\big(\mathrm{S}(x) - \mathrm{S}(x')\big), \mathrm{T}(x) - \mathrm{T}(x') - \big(\mathrm{S}(x) - \mathrm{S}(x')\big) \rangle \Big) \, \mathrm{d}\mu(x)\mathrm{d}\mu(x') \\
&= \frac{1}{2} \iint \mathrm{d}_W\big(\mathrm{T}(x) - \mathrm{T}(x'), \mathrm{S}(x) - \mathrm{S}(x')\big) \, \mathrm{d}\mu(x)\mathrm{d}\mu(x'). \qquad \square
\end{aligned} \quad (213)$$

Now, we make the connection with the mirror map used by Deb et al. [41] and derive the related Bregman divergence.

**Lemma 33.** *Let $\phi_\mu(\mathrm{T}) = \frac{1}{2} \mathrm{W}_2^2(\mathrm{T}_{\#}\mu, \rho)$ for $\mu, \rho \in \mathcal{P}_{2,\mathrm{ac}}(\mathbb{R}^d)$. Then, for all $\mathrm{T}, \mathrm{S} \in L^2(\mu)$, such that $\mathrm{T}_{\#}\mu, \mathrm{S}_{\#}\mu \in \mathcal{P}_{2,\mathrm{ac}}(\mathbb{R}^d)$,*

$$\mathrm{d}_{\phi_\mu}(\mathrm{T}, \mathrm{S}) = \frac{1}{2} \| \mathrm{T}^\rho_{\mathrm{T}_{\#}\mu} \circ \mathrm{T} - \mathrm{T}^\rho_{\mathrm{S}_{\#}\mu} \circ \mathrm{S} - (\mathrm{T} - \mathrm{S}) \|^2_{L^2(\mu)} + \langle \mathrm{T}^\rho_{\mathrm{S}_{\#}\mu} \circ \mathrm{S} - \mathrm{S}, \mathrm{T}^\rho_{\mathrm{T}_{\#}\mu} \circ \mathrm{T} - \mathrm{T}^\rho_{\mathrm{S}_{\#}\mu} \circ \mathrm{S} \rangle_{L^2(\mu)}, \quad (214)$$

*where $\mathrm{T}^\rho_{\mathrm{T}_{\#}\mu}$ denotes the OT map between $\mathrm{T}_{\#}\mu$ and $\rho$.*

*Proof.* Let $T, S \in L^2(\mu)$ such that $T_{\#}\mu, S_{\#}\mu \in \mathcal{P}_{2,\mathrm{ac}}(\mathbb{R}^d)$. Remember that $\nabla_{W_2} W_2^2(\cdot, \rho) = \mathrm{Id} - T^\rho_\cdot$, then

$$
\begin{aligned}
d_{\phi_\mu}(T, S) &= \phi_\mu(T) - \phi_\mu(S) - \langle \nabla_{W_2}\phi(S_{\#}\mu) \circ S, T - S \rangle_{L^2(\mu)} \\
&= \frac{1}{2} W_2^2(T_{\#}\mu, \rho) - \frac{1}{2} W_2^2(S_{\#}\mu, \rho) - \langle (\mathrm{Id} - T^\rho_{S_{\#}\mu}) \circ S, T - S \rangle_{L^2(\mu)} \\
&= \frac{1}{2} \| T^\rho_{T_{\#}\mu} \circ T - T \|^2_{L^2(\mu)} - \frac{1}{2} \| T^\rho_{S_{\#}\mu} \circ S - S \|^2_{L^2(\mu)} + \langle T^\rho_{S_{\#}\mu} \circ S - S, T - S \rangle_{L^2(\mu)} \\
&= \frac{1}{2} \| T^\rho_{T_{\#}\mu} \circ T - T \|^2_{L^2(\mu)} - \frac{1}{2} \| T^\rho_{S_{\#}\mu} \circ S - S \|^2_{L^2(\mu)} \\
&\quad + \langle T^\rho_{S_{\#}\mu} \circ S - S, T - T^\rho_{S_{\#}\mu} \circ S \rangle_{L^2(\mu)} + \langle T^\rho_{S_{\#}\mu} \circ S - S, T^\rho_{S_{\#}\mu} \circ S - S \rangle_{L^2(\mu)} \\
&= \frac{1}{2} \| T^\rho_{T_{\#}\mu} \circ T - T \|^2_{L^2(\mu)} + \frac{1}{2} \| T^\rho_{S_{\#}\mu} \circ S - S \|^2_{L^2(\mu)} \\
&\quad - \langle T^\rho_{S_{\#}\mu} \circ S - S, T^\rho_{S_{\#}\mu} \circ S - T \rangle_{L^2(\mu)} \\
&= \frac{1}{2} \| T^\rho_{T_{\#}\mu} \circ T - T \|^2_{L^2(\mu)} + \frac{1}{2} \| T^\rho_{S_{\#}\mu} \circ S - S \|^2_{L^2(\mu)} \\
&\quad - \langle T^\rho_{S_{\#}\mu} \circ S - S, T^\rho_{S_{\#}\mu} \circ S - T^\rho_{T_{\#}\mu} \circ T \rangle_{L^2(\mu)} \\
&\quad - \langle T^\rho_{S_{\#}\mu} \circ S - S, T^\rho_{T_{\#}\mu} \circ T - T \rangle_{L^2(\mu)} \\
&= \frac{1}{2} \| T^\rho_{T_{\#}\mu} \circ T - T^\rho_{S_{\#}\mu} \circ S - (T - S) \|^2_{L^2(\mu)} \\
&\quad + \langle T^\rho_{S_{\#}\mu} \circ S - S, T^\rho_{T_{\#}\mu} \circ T - T^\rho_{S_{\#}\mu} \circ S \rangle_{L^2(\mu)}.
\end{aligned}
\tag{215}
$$
$\qquad\square$

