# OpenReview forum: "Mirror and Preconditioned Gradient Descent in Wasserstein Space"
_NeurIPS.cc/2024/Conference — NeurIPS 2024 spotlight_

### Official Review · Reviewer_P753 · 2024-07-08

**Soundness:** 4
**Presentation:** 4
**Contribution:** 4
**Rating:** 9
**Confidence:** 5

**Summary:**

The authors study the mirror descent method in Wasserstein-2 spaces. This is based on constructing the Bregman divergence functionals in Wasserstein-2 spaces. One can use it to define a mirror descent direction in the sense of pushforward mapping functions, which generalizes the classical Wasserstein gradient descent direction. The authors then studied the convergence analysis of the Wasserstein mirror descent algorithm. Several numerical examples are used to demonstrate the effectiveness of the proposed method.

**Strengths:**

The authors define and apply the Wasserstein mirror descent directions to conduct sampling algorithms.

The mirror descent method in Wasserstein space is a very natural and novel approach. Congratulations to the authors for using the Wasserstein-Bregman divergences in sampling algorithms. This paper's proof is very clear in the mirror descent method from optimization and optimal transport.

**Weaknesses:**

On page 8, the authors don't study the mirror descent of KL divergences for non-Gaussian target distributions. This could be a very intriguing example. The authors may think about how to implement this case. This could be compared with the Wasserstein gradient flow of KL divergence.

**Questions:**

The authors also propose the Wassrsetin-Bregman proximal gradient method. Can authors illustrate the main difficulty of implementing it in practice or some potential computational schemes? How does it compare with the classical sampling schemes if the objective functional is chosen as the KL divergence?

I am willing to improve my score after addressing these questions.

**Limitations:**

There is no limitation on this paper.

---

> ### Author Rebuttal · Authors · 2024-08-05
>
> Thank you for your appraisal and positive comments on our paper.
>
> **On page 8, the authors don't study the mirror descent of KL divergences for non-Gaussian target distributions. This could be a very intriguing example. The authors may think about how to implement this case. This could be compared with the Wasserstein gradient flow of KL divergence.**
>
> The difficulty of optimizing the KL divergence with a non-Gaussian target is that we cannot use closed-forms for the mean and covariance anymore (as done in the experiment of Figure 2 for Gaussian, where closed-forms were obtained by projecting the distributions at each step on the space of Gaussian).
>
> A possible solution is to use a particle approximation, which we considered in Appendix I.2, with the goal to sample from a distribution on the simplex. Thus, the Bregman potential was chosen here as a potential energy, with potential a barrier allowing to keep the samples onto the simplex. The density of the distribution at each step was approximated with a Kernel Density Estimator to approximate the Wasserstein gradient of the KL.
>
>
> **The authors also propose the Wasserstein-Bregman proximal gradient method. Can authors illustrate the main difficulty of implementing it in practice or some potential computational schemes? How does it compare with the classical sampling schemes if the objective functional is chosen as the KL divergence?**
>
> The main difficulty of implementing the Wasserstein-Bregman proximal gradient method is that it is required to solve at each step a backward step (see equation (128)), which does not have a closed-form in general [1]. Thus, at each step, it is required to solve another optimization problem. For the classical JKO scheme, it has been proposed to solve it using approximations with e.g. entropic approximation [2] or neural networks [3,4], but these methods are computationally costly. Nonetheless, note that there is a closed-form when the target and initial distribution are Gaussian [5,6].
>
> In Figure 2, we considered the Wasserstein-Bregman proximal gradient method with the KL divergence as objective functional, and a Gaussian $\mathcal{N}(0,\Sigma)$ as target distribution. For a Bregman potential of the form $\phi(\mu) = \int V\mathrm{d}\mu$, with $V(x)=x^T\Lambda^{-1} x$, we derived the closed-form in Appendix H.4 leveraging the fact that the distributions are always Gaussian. Using $\Lambda=\Sigma$, we showed on Figure 2 that it converges much faster than the regular Wasserstein proximal method studied in [1,5] (which corresponds to $\Lambda=I_d$).
>
>
> [1] Salim, A., Korba, A., & Luise, G. (2020). The Wasserstein proximal gradient algorithm. Advances in Neural Information Processing Systems, 33, 12356-12366.
>
> [2] Peyré, G. (2015). Entropic approximation of Wasserstein gradient flows. SIAM Journal on Imaging Sciences, 8(4), 2323-2351.
>
> [3] Mokrov, P., Korotin, A., Li, L., Genevay, A., Solomon, J. M., & Burnaev, E. (2021). Large-scale wasserstein gradient flows. Advances in Neural Information Processing Systems, 34, 15243-15256.
>
> [4] Alvarez-Melis, D., Schiff, Y., & Mroueh, Y. (2021). Optimizing functionals on the space of probabilities with input convex neural networks. arXiv preprint arXiv:2106.00774.
>
> [5] Diao, M. Z., Balasubramanian, K., Chewi, S., & Salim, A. (2023). Forward-backward Gaussian variational inference via JKO in the Bures-Wasserstein space. In International Conference on Machine Learning (pp. 7960-7991). PMLR.
>
> [6] Wibisono, A. (2018). Sampling as optimization in the space of measures: The Langevin dynamics as a composite optimization problem. In Conference on Learning Theory (pp. 2093-3027). PMLR.

---

> > ### Comment · Reviewer_P753 · 2024-08-08
> > **Reply to authors**
> >
> > The authors have addressed my questions. Congratulations to the authors for working in a very important direction. More serious computations and analysis can be conducted along this line.

---

> > > ### Author Response · Authors · 2024-08-11
> > >
> > > Thank you for your response and again for your positive comments!

---

### Official Review · Reviewer_PBU2 · 2024-07-08

**Soundness:** 4
**Presentation:** 3
**Contribution:** 4
**Rating:** 7
**Confidence:** 3

**Summary:**

In this paper, the authors endeavor to integrate the concepts of mirror descent (MD) and preconditioned gradient descent (PGD) within the framework of Wasserstein distances, from a convergence theory perspective. Initially, they provide a comprehensive background including, but not limited to, Wasserstein distances, Bregman divergence, convexity, and smoothness. With assumptions regarding $\beta$-smoothness and $\alpha$-convexity, they successfully demonstrate the iterative procedures for MD and PGD in the Wasserstein space. Subsequently, they conduct experiments focusing on sampling methodologies and single-cell dynamics.

**Strengths:**

1. **Solid Mathematical Foundations:** The authors offer a rigorous mathematical derivation concerning the convergence behaviors of MD and PGD, supplemented by a thorough analysis of the theoretical background.
2. **Relevance to Conference Themes:** The paper presents fundamental theoretical analysis that is highly pertinent to the themes of the conference.
3. **Comprehensive Background Introduction:** The supplementary materials provide an extensive overview necessary to understand the nuances of the paper. The foundational work done by the authors is commendable.

**Weaknesses:**

1. **Results Presentation:** The presentation of results could be enhanced by providing more detailed visualizations. For instance, it would be beneficial to include a snapshot illustrating particle evolution for Figure 1, right $K_2$. Additionally, if the authors have access to MATLAB, it is recommended to utilize the `surf` function for the contour of Dirichlet distribution's pdf in Figure 4 to better demonstrate the sampling tasks.
2. **Clarification of Definitions:** It is recommended that the authors include explicit definitions of convergence and global convergence to preclude any potential misunderstandings in the appendix.
3. **Visualization of Convergence Data:** I suggest employing seaborn's `sns.lineplot` with a shaded area to depict the error bars, which would support the assertions made in checklist item 7 more robustly.
4. **Overlooked References:** It appears that significant references relevant to discussions on the convergence of mirror descent may have been overlooked. I recommend incorporating additional reference [1] that provides pertinent insights into these discussions.


----
References:

[1]. Tzen, Belinda, et al. "Variational principles for mirror descent and mirror langevin dynamics." IEEE Control Systems Letters 7 (2023): 1542-1547.

**Questions:**

1. **About the Proof of Convergence:**
   From the perspective of gradient flow, assuming we have a velocity field $\dot{X} = v_t$, we can derive that $v_t = -\Vert \nabla\frac{\delta \mathcal{F}}{\mu} \Vert_2^2$. In this context, does assuming $\beta$-smoothness offer additional advantages beyond the conventional benefits?

2. **Alternative Proof Approaches:**
   Could the convergence location and rate be established from the perspective of a Lyapunov functional? This approach may provide a robust framework for proving stability and convergence.

3. **Riemannian-based Approaches:**
   Is it feasible to extend the proposed method to Riemannian-based approaches, such as Riemannian SVGD [1]?

4. **About Higher Order Systems and Extensions:**
   Is it possible to extend this approach to systems characterized by the Fisher-Rao metric [2]? Furthermore, considering that the continuity equation in Wasserstein space represents a first-order system, could this approach be adapted to Hamiltonian Flow [3], which is the second order system to my knowledge?
---
### References
[1]. Zhang, Ruqi, Qiang Liu, and Xin Tong. "Sampling in constrained domains with orthogonal-space variational gradient descent." Advances in Neural Information Processing Systems 35 (2022): 37108-37120.
[2]. Neklyudov, Kirill, et al. "Wasserstein Quantum Monte Carlo: A Novel Approach for Solving the Quantum Many-Body Schrödinger Equation." Advances in Neural Information Processing Systems 36 (2024).
[3]. Ambrosio, Luigi, and Wilfrid Gangbo. "Hamiltonian ODEs in the Wasserstein Space of Probability Measures." Communications on Pure and Applied Mathematics: A Journal Issued by the Courant Institute of Mathematical Sciences 61.1 (2008): 18-53.

**Limitations:**

I appreciate the efforts of the authors. Nevertheless, I recommend that the authors include a dedicated section in the appendix detailing the assumptions such as convexity and smoothness within functionals. This addition would significantly aid in understanding and following the methodologies employed in this work.

---

> ### Author Rebuttal · Authors · 2024-08-05
>
> Thank you for reading the paper and for your feedback. We answer your comments below. Please do not hesitate if you have other questions.
>
> **The presentation of results could be enhanced by providing more detailed visualizations.**
>
> Thank you for these suggestions, we will take it into account and add snapshots of particles as well as contours.
>
> **It appears that significant references relevant to discussions on the convergence of mirror descent may have been overlooked. I recommend incorporating additional reference [1] that provides pertinent insights into these discussions.**
>
> Thank you for this reference that we did not know. We will add it to the paper.
>
> **From the perspective of gradient flow, assuming we have a velocity field $\dot{X} = v_t$, we can derive that $v_t = -\Vert \nabla\frac{\delta \mathcal{F}}{\mu} \Vert_2^2$. In this context, does assuming $\beta$-smoothness offer additional advantages beyond the conventional benefits?**
>
> For Wasserstein gradient flows, the velocity field is $v_t=-\nabla_{W_2}\mathcal{F}(\mu_t)$, and we have $\frac{\mathrm{d}\mathcal{F}(\mu_t)}{\mathrm{d}t} = \langle \nabla_{W_2}\mathcal{F}(\mu_t), v_t\rangle_{L^2(\mu_t)} = - \|\nabla_{W_2}\mathcal{F}(\mu_t)\|_{L^2(\mu_t)}^2$. Thus, the objective of the continuous formulation is necessarily non-increasing without assuming any smoothness of the objective.
>
> In discrete time, the Wasserstein gradient descent (obtained as the explicit Euler discretization) $\mu_{k+1} = (\mathrm{Id}-\tau\nabla_{W_2}\mathcal{F}(\mu_k))_\\#\mu_k$ requires smoothness of the objective to obtain descent at each iteration (and eventually global convergence guarantees if we assume furthermore convexity). See for instance analysis of gradient descent [2]. If the step size is too big, it is not guaranteed that the scheme decreases at each iteration, and could actually diverge. Hence in practice, knowing/estimating the smoothness constant will guide the choice of the step size.
>
> **Could the convergence location and rate be established from the perspective of a Lyapunov functional?**
>
> Our rates can indeed be obtained by decreasing the following Lyapunov functional: $L_k = k\tau(\mathcal{F}(\mu_k) - \mathcal{F}(\nu)) + \tau(\frac{1}{\tau}-\alpha) W_\phi(\nu,\mu_k)$. Indeed, $L_{k+1}-L_k \le 0$ by using the inequalities from equations (85) and (7). Thus $L_k$ is a Lyapunov function [3]. By using a telescopic sum, we recover the inequality $\mathcal{F}(\mu_k) - \mathcal{F}(\nu) \le \frac{1-\alpha\tau}{k\tau} W_\phi(\nu,\mu_0)$ obtained in Proposition 3. We will make this clearer.
>
> **Is it feasible to extend the proposed method to Riemannian-based approaches, such as Riemannian SVGD?**
>
> Extending the proposed methods to Riemannian manifolds is an interesting future direction of works that we did not consider yet.
>
> For now, Wasserstein gradient flows on manifolds have been sparsely studied. For instance, [4] studied theoretically Wasserstein gradient flows of the negative entropy on manifolds. In a more applied setting, [5] computed Wasserstein gradient flows on Lie group using the JKO scheme and [6] performed Wasserstein gradient descent of the Sliced-Wasserstein distance on Hadamard manifolds. There, the schemes and analyis have been derived on a case-by-case basis.
>
> To the best of our knowledge, the Mirror Descent algorithm has not been yet extended to Riemannian manifolds. One difficulty might come from the current theoretical analysis which requires to split the inner product, which is not directly possible when using log maps. Moreover, the scheme might also come with computational difficulties, as there might not be closed-forms easy to compute in general.
>
> **Is it possible to extend this approach to systems characterized by the Fisher-Rao metric?**
>
> The gradient in the space of probability distributions endowed with the Fisher-Rao metric is the 1st variation [7]. Thus, we might define Bregman divergences and the associated Mirror-Descent scheme using this gradient. This was studied e.g. in [8] for Mirror Descent with applications on the minimization of the KL divergence.
>
>
> **Could this approach be adapted to Hamiltonian Flow, which is the second order system to my knowledge?**
>
> This is an interesting direction for future works which we did not yet consider and which goes beyond our knowledge.
>
>
> **I recommend that the authors include a dedicated section in the appendix detailing the assumptions such as convexity and smoothness within functionals.**
>
> Thank you for this suggestion. We will expand our existing sections B.2 and C.3 in the appendix in this direction.
>
> [1] Tzen, B., Raj, A., Raginsky, M., & Bach, F. (2023). Variational principles for mirror descent and mirror langevin dynamics. IEEE Control Systems Letters, 7, 1542-1547.
>
> [2] Garrigos, G., & Gower, R. M. (2023). Handbook of convergence theorems for (stochastic) gradient methods. arXiv preprint arXiv:2301.11235.
>
> [3] Wilson, A. (2018). Lyapunov arguments in optimization. University of California, Berkeley.
>
> [4] Erbar, M. (2010). The heat equation on manifolds as a gradient flow in the Wasserstein space. In Annales de l'IHP Probabilités et statistiques (Vol. 46, No. 1, pp. 1-23).
>
> [5] Bon, D., Pai, G., Bellaard, G., Mula, O., & Duits, R. (2024). Optimal Transport on the Lie Group of Roto-translations. arXiv preprint arXiv:2402.15322.
>
> [6] Bonet, C., Drumetz, L., & Courty, N. (2024). Sliced-Wasserstein Distances and Flows on Cartan-Hadamard Manifolds. arXiv preprint arXiv:2403.06560.
>
> [7] Gallouët, T. O., & Monsaingeon, L. (2017). A JKO splitting scheme for Kantorovich--Fisher--Rao gradient flows. SIAM Journal on Mathematical Analysis, 49(2), 1100-1130.
>
> [8] Aubin-Frankowski, P. C., Korba, A., & Léger, F. (2022). Mirror descent with relative smoothness in measure spaces, with application to sinkhorn and em. Advances in Neural Information Processing Systems, 35, 17263-17275.

---

> ### Comment · Reviewer_PBU2 · 2024-08-09
> **Comments on Authors from Reviewer [PBU2]**
>
> Thank you for your detailed response, and I appreciate to adjust the score. Additionally, I have a follow-up question regarding lines 1447 and 1448. Why is KDE necessary when conducting mirror descent on the simplex? To my understanding, since the density of the Dirichlet distribution is bounded, ensuring that the velocity field meets the boundary condition $\lim_{\Vert x \Vert\rightarrow\infty} v(x)=0$ should suffice. We could potentially use the equation $ \int{\nabla_{x} [p(x)v(x)]} = 0 = \int{\nabla_x p(x) v(x)} + \int{p(x)\nabla v(x)} $ to circumvent the need for explicit density estimation. Could you please clarify this approach?

---

> > ### Author Response · Authors · 2024-08-11
> >
> > Thank you for your response and again for your feedback. We address your question below.
> >
> > **Why is KDE necessary when conducting mirror descent on the simplex?**
> >
> > In the experiment of Mirror Descent on the simplex, we minimize the Kullback-Leibler divergence $\mathcal{F}(\mu) = \mathrm{KL}(\mu||\nu)$ for $\nu \propto e^{-V}$ restricted to the simplex. To enforce the distributions to stay on the simplex, we used as Bregman potential $\phi(\mu) = \int \psi \mathrm{d}\mu$ with $\psi(x) = \sum_{i=1}^d x_i \log(x_i) + (1-\sum_{i=1}^d)\log(1-\sum_{i=1}^d x_i)$, which acts as a barrier.
> >
> > In this situation, the Mirror Descent scheme translates as $\mu_{k+1} = \big(\nabla \psi^* \circ (\nabla \psi - \tau \nabla_{W_2}\mathcal{F(\mu_k)})\big)_\\#\mu_k$, with $\nabla\_{W_2}\mathcal{F}(\mu_k) = \nabla V + \nabla \log p_k$ and $p_k$ the density of $\mu_k$. To approximate the distributions, we use an approximation with particles $\hat{\mu}_k^n = \frac{1}{n}\sum\_{i=1}^n \delta\_{x_i^k}$. However, we need to approximate the density $p_k$ to be able to compute $\nabla\_{W_2}\mathcal{F}(\mu_k)$. Thus, we used a kernel density estimator to do this.
> >
> > The equation you mention might provide another way to enforce the particles to stay on the simplex, hence avoiding the use of a barrier for the Bregman potential. This is an interesting direction of work that we did not consider yet.

---

> > > ### Comment · Reviewer_PBU2 · 2024-08-13
> > >
> > > Dear Authors,
> > > Thank you for your response. I have no further comments.
> > >
> > > Sincerely,
> > > Reviewer PBU2

---

### Official Review · Reviewer_mqrt · 2024-07-09

**Soundness:** 4
**Presentation:** 3
**Contribution:** 3
**Rating:** 6
**Confidence:** 4

**Summary:**

This paper presented a unified view on the functional optimization on the Wasserstein space. Authors proposed a mirror descent algorithm and a preconditioned gradient descent algorithm, both are applied to minimize the objective functional over the Wasserstein space. Many existing algorithms in the Wasserstein space-related literature can be recovered or seen as a special case of the algorithms proposed by the authors.
Convergence guarantees were provided and backed with numerical results.

**Strengths:**

Mirror descent and preconditioned GD are closely related and have been seen as generalizations of GD. Though Wasserstein GD have been proposed, the extension to MD and preconditioned GD is a natural step.

The formulation in the paper is general and can be applicable to a wide range of studies in both machine learning application and theory.

The literature review provided by the authors seem to be thorough, and the position of the current manuscript in the literature is well explained. The relationships between the proposed algorithm and existing algorithm were also clear throughout the paper.


The reviewer did not read the full analysis in the appendix, but the technical aspects of this paper seems solid and intuitive.

**Weaknesses:**

The main concern I have is as presented in the MD optimization literature: what new tool does the MD formulation bring to the table? Previously, the convergence of MD is similar to or worse than GD. The literature on relative strong smoothness and convexity enabled MD to converge even when the objective function is not smooth/convex in the Euclidean sense (which was really mostly KL). The function properties in the Wasserstein space are more complicated and therefore difficult to align, this was briefly discussed by the authors at the start of section 5. But the application offered later is largely limited to existing applications KL or Sinkhorn, whereas these specific cases seem to have been already studied by the literature anyway.

Some sections are not well presented. For instance, the subsection starting at l.136, the statements on the potential energies are confusing and it is very difficult to associate on sentence with either the previous or the latter sentence (such as "... specific example. In particular, we have xxx. We will also consider...", is the "xxx" with the specific example of what the authors consider?) Similar confusions also exist in this paper, but to a lesser extent.

While the contribution is clear from reading the entire paper, the "Contributions" paragraph does not help reader and clarify what this paper offers.

Though discussed briefly, the computational complexity and real-life run-time of these type of methods should be better addressed both in theory and experiments, especially on the comparison between the proposed and benchmark methods.

**Questions:**

some minor grammatical errors such as l.158 ($T_\eta^\nu$ OT maps from $\eta$...)

**Limitations:**

The authors did address the limitations of this work.

---

> ### Author Rebuttal · Authors · 2024-08-05
>
> Thank you for your valuable feedback and time. We have addressed your comments point-by-point below. Please don't hesitate to let us know if you have any further questions.
>
> **What new tool does the MD formulation bring to the table?**
>
> First, the Mirror Descent algorithm on the Wasserstein space allows to perform optimization over probability distributions on constrained spaces (and in particular sampling on constrained spaces for specific objectives). This motivated the development of e.g. Mirror Langevin [1,2,3]. This lens on Mirror Descent on the Wasserstein space was also studied e.g. in [4] to optimize the KL, but without a theoretical analysis of the discrete algorithm.
> Here, we chose to study the convergence of these algorithms, and one of the main contributions of this work is to define appropriate notion of relative smoothness and convexity. This differs a lot from previous works, and is new on the Wasserstein space to the best of our knowledge.
>
> Moreover, we also proposed to use general Bregman potentials, instead of just considering potential energies as done in most previous works focusing on Bregman divergences and Mirror Descent on the Wasserstein space. We believe that this additional flexibility could allow to find new pairs of functionals/Bregman potentials, with theoretical guarantees for the convergence of the Mirror Descent scheme.
> For instance, a convex functional is 1-smooth and 1-convex relative to itself, and thus could be used as Bregman potential for its optimization, provided we can compute the scheme. We note that this point of view was also recently taken to optimize the MMD in [5], while the MMD is known to be non convex w.r.t the Wasserstein distance [6], which leads MMD flows to converge to spurious local minima in practice [5,6,7].
>
>
> **The function properties in the Wasserstein space are more complicated and therefore difficult to align, this was briefly discussed by the authors at the start of section 5.**
>
> Showing the relative smoothness and convexity of functionals is indeed not straightforward in general. In the first paragraph of Section 5, we described some cases where we can show it leveraging the relative smoothness and convexity on $\mathbb{R}^d$, and suggested in the general case to compare the Hessians.
>
>
> **But the application offered later is largely limited to existing applications KL or Sinkhorn.**
>
> We did not focus only on minimizing the KL or Sinkhorn. On Figure 1, we applied the Mirror Descent scheme on an interaction energy with an interaction energy as Bregman potential, in a case where it is well relatively smooth.
> We focused on the KL for the Gaussian example of Figure 2, but this specific example has not been yet studied in the literature as we use Mirror Descent with negative entropy as Bregman potential.
> Finally, on Figure 3, we applied the preconditioned scheme on different divergences such as the Sliced-Wasserstein distance, the Sinkhorn divergence and the Energy distance. While Wasserstein gradient flows of these divergences have already been proposed and studied, applying the preconditioned gradient descent to them is new to the best of our knowledge.
>
>
> **Some sections are not well presented.**
>
> We apologize for the confusions and will revise the paper to improve its clarity.
>
>
> **The computational complexity and real-life run-time of these type of methods should be better addressed both in theory and experiments, especially on the comparison between the proposed and benchmark methods.**
>
> Compared to the usual Wasserstein gradient scheme, the preconditioned Wasserstein gradient scheme only requires to evaluate additionaly the gradient of $h^*$, and thus has the same computational complexity as the Wasserstein gradient descent. Nonetheless, we noted in Figure 3 that it required less iterations to converge for a well-chosen preconditioner.
>
> For Mirror Descent, the complexity depends on the Bregman potential. If we use $\phi_\mu(T)=\int V\circ T\ \mathrm{d}\mu$ with $V$ a potential which we know in closed form, and from which we can compute $\nabla V$ and $\nabla V^*$, then the computational cost is the same as the Wasserstein gradient descent. In the more general case, where we do not know how to invert $\nabla\phi_\mu$, we need to use the Newton method, which is more costly as it requires to invert a Jacobian of size $nd\times nd$ as stated Section I.1.
>
> The runtimes are reported in Appendix I. For instance, as stated line 1490, for the experiment of Figure 1, the Mirror Descent used in dimension $d=2$ for $n=100$ particles and $120$ epochs with $K_2$ and $K_2^\Sigma$ took about 5 minutes while with $K_4$ and $K_4^\Sigma$, it took about 1h. We leave for future works the development of cheaper optimization methods.
>
>
> **some minor grammatical errors**
>
> Thank you, we will correct the grammatical errors.
>
>
> [1] Ahn, K., & Chewi, S. (2021). Efficient constrained sampling via the mirror-Langevin algorithm. Advances in Neural Information Processing Systems, 34, 28405-28418.
>
> [2] Chewi, S., Le Gouic, T., Lu, C., Maunu, T., Rigollet, P., & Stromme, A. (2020). Exponential ergodicity of mirror-Langevin diffusions. Advances in Neural Information Processing Systems, 33, 19573-19585.
>
> [3] Hsieh, Y. P., Kavis, A., Rolland, P., & Cevher, V. (2018). Mirrored langevin dynamics. Advances in Neural Information Processing Systems, 31.
>
> [4] Sharrock, L., Mackey, L., & Nemeth, C. (2023, January). Learning Rate Free Bayesian Inference in Constrained Domains. In NeurIPS.
>
> [5] Gladin, E., Dvurechensky, P., Mielke, A., & Zhu, J. J. (2024). Interaction-Force Transport Gradient Flows. arXiv preprint arXiv:2405.17075.
>
> [6] Arbel, M., Korba, A., Salim, A., & Gretton, A. (2019). Maximum mean discrepancy gradient flow. Advances in Neural Information Processing Systems, 32.
>
> [7] Galashov, A., de Bortoli, V., & Gretton, A. (2024). Deep MMD Gradient Flow without adversarial training. arXiv preprint arXiv:2405.06780.

---

### Official Review · Reviewer_3j2f · 2024-07-12

**Soundness:** 4
**Presentation:** 3
**Contribution:** 3
**Rating:** 7
**Confidence:** 3

**Summary:**

The paper generalises mirror descent and preconditioning approaches from optimisation over R^d to optimisation over Wasserstein space, which is applicable in some actual problems, and confirms by the corresponding numerical experiments the efficiency of these approaches.

**Strengths:**

The paper completely corresponds to its stated goal, uses the most actual theoretical frameworks of long known approaches from optimisation over R^d, and focuses indeed on the most fundamental methods, which will allow a follower to develop any desired extension based on idea from optimisation over R^d and the detailed examination of technical issues related to the transition to Wasserstein space made in this work. Practical contribution of this paper is also notable, as it demonstrates in numerical experiments, but I guess that thanks to the proposed theoretical basement, many improvements can be made on top of that, which is to be done in future works.

**Weaknesses:**

The preconditioning is considered in some particular yet practically important case, but the complete generalisation of preconditioning is still to be finished.

**Questions:**

There is an aspect which was not mentioned in this paper, but is interesting from the theoretical point of view: stochastic oracle. The generalisation of stochastic mirror descent to Wasserstein space seems to be the demanding yet possible task, when that of preconditioning is challenging even in R^d case, but it would be interesting to find the proper notions of stochasticity and flexible enough analysis to deal with it in case of Wassestein space. Is your current analysis generalisable to stochastic optimisation and to inexact model of function, more generally?

**Limitations:**

The limitations are clear from reading.

---

> ### Author Rebuttal · Authors · 2024-08-05
>
> Thank you for your appraisal and positive comments on our paper.
>
> **The preconditioning is considered in some particular yet practically important case, but the complete generalisation of preconditioning is still to be finished.**
>
> We acknowledge that the complete generalization and study of the preconditioned gradient descent on the Wasserstein space is left for future works. We note that the convergence results would still be valid for $\phi_\mu$ strictly convex, and differentiable as described in the first paragraph of Section 4.
>
>
> **Is your current analysis generalisable to stochastic optimisation and to inexact model of function, more generally?**
>
> We expect our analysis to be generalisable to the stochastic setting. While we leave it for future works, we note that [1, 2] studied stochastic Wasserstein gradient schemes, while [3] studied Mirror Descent on $\mathbb{R}^d$ in the stochastic setting. Thus, it could be a nice avenue of future works to extend these results to Mirror Descent on the Wasserstein space.
>
>
> [1] Diao, M. Z., Balasubramanian, K., Chewi, S., & Salim, A. (2023, July). Forward-backward Gaussian variational inference via JKO in the Bures-Wasserstein space. In International Conference on Machine Learning (pp. 7960-7991). PMLR.
>
> [2] Backhoff-Veraguas, J., Fontbona, J., Rios, G., & Tobar, F. (2022). Stochastic Gradient Descent for Barycenters in Wasserstein Space. arXiv preprint arXiv:2201.04232.
>
> [3] Hanzely, F., & Richtárik, P. (2021). Fastest rates for stochastic mirror descent methods. Computational Optimization and Applications, 79, 717-766.

---

### Official Review · Reviewer_KGf7 · 2024-07-15

**Soundness:** 4
**Presentation:** 2
**Contribution:** 3
**Rating:** 6
**Confidence:** 3

**Summary:**

The paper provides convergence analysis for Mirror descent (with and without additional preconditioning) in the Wasserstein space. The analysis is performed for relatively smooth and convex functionals. The use of the studied algorithms on computational biology problem is illustrated.

**Strengths:**

The theory in the paper seems sound and substantially extends the prior work overcoming some technical difficulties related to the discrete nature of the algorithms (explicit discretization) and non-Euclidean geometry in the Wasserstein space. Convergence rates are similar to those in finite dimensional space for MD.

**Weaknesses:**

The paper is very dense and seems like not well adapted to 9 page format. A lot of explanations and implementation details are referred to the appendix. It is very difficult to read the paper when it is written is such style. For example, although I have spent several hours only trying to understand how the algorithm is implemented, I did not manage to do it even with the help of the appendix. I would advise the authors to revise the paper to make it more accessible.


1. Notations section should be improved. Many things are not properly defined throughout the paper, e.g., the coupling between distributions, what "Id" is?

2. On line 138, what does it mean that the sets $V$ and $W$ are "differentiable and $L$-smooth with $W$ even?

3. On line 142, the calligraphic $\mathcal W$ is not defined.

4. OT map mentioned on line 151 is not defined.

**Questions:**

n/a

---

> ### Author Rebuttal · Authors · 2024-08-05
>
> Thank you for your valuable feedback and time. We have addressed your comments point-by-point below. Please don't hesitate to let us know if you have any further questions.
>
> **The paper is very dense. I would advise the authors to revise the paper to make it more accessible.**
>
> We understand your concerns about the paper's density and the difficulty in following the implementation details. Taking advantage of the extra page, we will work on revising the paper to make it more accessible and ensure that key explanations are clearer and more integrated into the main text.
>
> **How the algorithm is implemented?**
>
> We will describe more clearly how the schemes are implemented in practice in a dedicated section of the appendix.
>
> In practice, we use a particle approximation of the probability measures, i.e. we sample $x_1^{(0)},\dots,x_n^{(0)}\sim \mu_0$ and then we apply the schemes to $\hat{\mu}_k^n = \frac{1}{n} \sum\_{i=1}^n \delta\_{x\_i^{(k)}}$ for all $k\ge 0$.
>
> For Mirror Descent with a Bregman potential $\phi$ chosen as a potential energy $\phi(\mu) = \int V\ \mathrm{d}\mu$ with $\nabla V$ and $\nabla V^*$ known in closed-forms, we use equation (9) to implement it: for all $k\ge 0$ and $i\in \{1,\dots,n\}$, $x\_i^{(k+1)} = \nabla V^*\big(\nabla V(x\_i^{(k)}) - \tau \nabla\_{W\_2}\mathcal{F}(\hat{\mu}\_k^n)(x\_i^{(k)})\big)$. The Wasserstein gradient of $\mathcal{F}$ in most of the cases considered can be computed in closed-form, e.g. for $\mathcal{F}(\mu)=\int U\mathrm{d}\mu$, $\nabla_{W_2}\mathcal{F}(\mu) = \nabla U$, or for $\mathcal{F}(\mu)=\iint W(x-y)\ \mathrm{d}\mu(x)\mathrm{d}\mu(y)$, $\nabla_{W_2}\mathcal{F}(\mu) = \nabla W\star \mu$.
>
> In the more general case, where the gradient of the Bregman potential cannot be inverted in closed-form, we used Newton's algorithm at each step, which is detailed in Appendix I.1, and we use equation (191) to implement it.
>
> For the Gaussian experiment, we update directly the means and covariances. The closed-form of NEM is reported in equation (197), and the closed-form of the Forward-Backward scheme are in Appendix H.4 in equation (160).
>
> Finally, for the preconditioned gradient descent scheme, we implemented it using equation (11): for all $k\ge 0$ and $i\in\{1,\dots,n\}$, $x_i^{(k+1)} = x_i^{(k)} - \nabla h^*\big(\nabla_{W_2}\mathcal{F}(\hat{\mu}_k^n)(x_i^{(k)})\big)$. In the experiments of Figure 3, we considered $h^*(x)=(\|x\|_2^a + 1)^{\frac{1}{a}} - 1$ with $a>0$, and its gradient can be computed in closed-form or by backpropagation.
>
> **Notations section should be improved. Many things are not properly defined throughout the paper, e.g., the coupling between distributions, what "Id" is?**
>
> We will complete the notation section. A coupling between $\mu\in \mathcal{P}\_2(\mathbb{R}^d)$ and $\nu\in\mathcal{P}\_2(\mathbb{R}^d)$ is a probability distribution $\gamma\in\mathcal{P}(\mathbb{R}^d\times \mathbb{R}^d)$ such that $\pi^1_{\\#}\gamma=\mu$ and $\pi^2\_\\#\gamma=\nu$ with $\pi^1:(x,y)\mapsto x$ and $\pi^2:(x,y)\mapsto y$. The $\mathrm{Id}$ is the identity mapping $x\mapsto x$.
>
> **On line 138, what does it mean that the sets $V$ and $W$ are "differentiable and $L$-smooth with $W$ even?**
>
> $V$ and $W$ are functions from $\mathbb{R}^d$ to $\mathbb{R}$. $V$ (or $W$) $L$-smooth means that its gradient is $L$-Lipschitz. We will clarify this in the revision of the paper. To ask for $W$ even means that $W(-x)=W(x)$ for all $x\in \mathbb{R}^d$.
>
> **On line 142, the calligraphic $\mathcal W$ is not defined.**
>
> $\mathcal{W}$ is defined as $\mathcal{W}(\mu) = \iint W(x-y)\ \mathrm{d}\mu(x)\mathrm{d}\mu(y)$, see line 127.
>
> **OT map mentioned on line 151 is not defined.**
>
> An OT map between $\mu,\nu\in\mathcal{P}\_2(\mathbb{R}^d)$ is a map $T:\mathbb{R}^d\to\mathbb{R}^d$ satisfying $T\_\\#\mu=\nu$ and $W_2^2(\mu,\nu) = \|T-\mathrm{Id}\|_{L^2(\mu)}^2$. It exists e.g. provided $\mu\ll\mathrm{Leb}$ by Brenier's theorem. We will clarify this in the revised version of the paper.

---

> > ### Comment · Reviewer_KGf7 · 2024-08-13
> > **Rebuttal Acknowledgement**
> >
> > I thank the authors for answering my questions and I trust the authors can improve the presentation in the next revision. I have no further concerns.

---

### Author Rebuttal · Authors · 2024-08-05

We thank all the reviewers for their positive comments and common appraisal of the soundness of our approach to lift Euclidean optimization schemes to the Wasserstein space. Following the reviewers' comments, we will improve the clarity of the exposition for the revised version, taking advantage of the allowed extra page.

---

### Decision · Program_Chairs · 2024-09-25

**Decision:**

Accept (spotlight)

**Comment:**

This paper considers mirror descent and preconditioned gradient descent analogs in the  Wasserstein space of probability distributions. These algorithms better capture the geometry of the optimization has interesting properties explored in optimization literature. Authors explore those in Wasserstein space prove guarantees of convergence of Wasserstein-gradient-based discrete-time schemes.


This paper was reviewed by five reviewers with the following Scores/Confidence: 6/4, 6/3, 9/5, 7/3, 7/3. I think the paper is studying an interesting topic and the results are relevant to NeurIPS community. The following concerns were brought up by the reviewers:

- The paper is very dense and presentation can be friendly to a broader audiance.

- The advantages of MD framework should be emphasized.

- Definitions should be clarified. See reviewer PBU2's comments.


Authors should carefully go over reviewers' suggestions and address any remaining concerns in their final revision. Based on the reviewers' suggestion, as well as my own assessment of the paper, I recommend including this paper to the NeurIPS 2024 program.